# Catalytic activity tunable ceria nanoparticles prevent chemotherapy-induced acute kidney injury without interference with chemotherapeutics

Qinjie Weng [1,2,3,6], Heng Sun [2,4,6], Chunyan Fang[1,3,6], Fan Xia[2,4], Hongwei Liao[4], Jiyoung Lee[4], Jincheng Wang [1,3], An Xie[2,4], Jiafeng Ren[2,4], Xia Guo [2,4], Fangyuan Li[1,2,4], Bo Yang[1,2,3] & Daishun Ling [1,2,4,5,7 ✉]

Acute kidney injury (AKI) is a prevalent and lethal adverse event that severely affects cancer patients receiving chemotherapy. It is correlated with the collateral damage to renal cells caused by reactive oxygen species (ROS). Currently, ROS management is a practical strategy that can reduce the risk of chemotherapy-related AKI, but at the cost of chemotherapeutic efficacy. Herein, we report catalytic activity tunable ceria nanoparticles (CNPs) that can prevent chemotherapy-induced AKI without interference with chemotherapeutic agents. Specifically, in the renal cortex, CNPs exhibit catalytic activity that decomposes hydrogen peroxide, and subsequently regulate the ROS-involved genes by activating the *Nrf2/Keap1* signaling pathway. These restore the redox homeostasis for the protection of kidney tubules. Under an acidic tumor microenvironment, CNPs become inert due to the excessive $H^+$ that disrupts the re-exposure of active catalytic sites, allowing a buildup of chemotherapy-mediated ROS generation to kill cancer cells. As ROS-modulating agents, CNPs incorporated with context-dependent catalytic activity, hold a great potential for clinical prevention and treatment of AKI in cancer patients.

[1] Zhejiang Province Key Laboratory of Anti-Cancer Drug Research, College of Pharmaceutical Sciences, Zhejiang University, Hangzhou, China. [2] Hangzhou Institute of Innovative Medicine, College of Pharmaceutical Sciences, Zhejiang University, Hangzhou, China. [3] Center for Drug Safety Evaluation and Research, College of Pharmaceutical Sciences, Zhejiang University, Hangzhou, China. [4] Institute of Pharmaceutics, College of Pharmaceutical Sciences, Zhejiang University, Hangzhou, China. [5] School of Chemistry and Chemical Engineering, Frontiers Science Center for Transformative Molecules, Institute of Translational Medicine, Shanghai Jiao Tong University, Shanghai, China. [6] These authors contributed equally: Qinjie Weng, Heng Sun, Chunyan Fang. [7] The author supervised this work: Daishun Ling. ✉email: dsling@sjtu.edu.cn

Acute kidney injury (AKI) is defined as a sudden or rapid loss of renal function that leads to the drastic reduction in urine output and thus deleterious urea and creatinine retention, resulting in approximately 1.7 million deaths globally each year[1,2]. Chemotherapy is one of the prevalent causes of AKI[3], as various chemotherapeutic agents excreted by the kidney can damage the structure and functional constituents of renal cells through the generation of reactive oxygen species (ROS)[4–6]. More disastrously, the impaired renal excretion further elevates toxin buildup and damages the kidney, forming a vicious circle endangering cancer patients[7]. Chemotherapeutic agent dosage reduction can mitigate the progression of AKI[8]; however, it comes at the price of limited tumor suppression efficacy. Small molecular antioxidants, such as N-acetyl cysteine (NAC), can be utilized as an alternative to protect against chemotherapy-induced AKI by reducing oxidative damage[9]. Recently, nanomaterials with ROS-regulating capacities have been studied to broaden the therapeutic strategies for ROS-involved diseases, including AKI[10–13]. For example, molybdenum-based polyoxometalate nanoclusters were fabricated as nano-antioxidants that preferentially accumulated in kidneys for efficient renal protection[14]; DNA origami nanostructures had demonstrated a potent ROS scavenging capacity to protect kidney structures and ameliorate AKI[15]. However, small molecular antioxidants or nanomaterials that indiscriminately reduce oxidative stress in tumor tissues are argued to spur the growth and metastasis of tumors[16–20], hindering their potential application for chemotherapy-induced AKI prevention in cancer patients.

Herein, by exploiting the pathophysiological differences between kidney and tumor, especially the pH discrepancy[21,22], we report catalytic activity tunable ceria nanoparticles (CNPs) that can reconcile AKI prevention and potent chemotherapy (Fig. 1a). The systemically administrated CNPs exhibit enzyme-like activities and activate the *Nrf2/Keap1* signaling pathway for efficient ROS scavenging in the neutral microenvironment of kidneys. However, under an acidic tumor microenvironment, CNPs become inert due to the excessive $H^+$ that disrupts the re-exposure of active catalytic sites, thus do not interfere with chemotherapy-mediated ROS generation for cancer therapy. Consequently, CNPs incorporated with context-dependent ROS modulation capacity may open a novel avenue for the prevention of AKI, especially for patients receiving chemotherapy.

## Results

### Synthesis and characterization of CNPs with tunable catalytic activity.

The hydrophobic CNPs with a diameter of ~3 nm (Fig. 1b) were synthesized via a modified reverse micelle method[23], and 1,2-distearoyl-sn-glycero-3-phosphoethanolamine-N-[methoxy (polyethylene glycol)-2,000] (DSPE-PEG$_{2K}$) was successfully surface modified (Supplementary Fig. 1), imparting the colloidal stability of the CNPs in aqueous solution (with a hydrodynamic diameter of ~9.7 nm and a surface zeta potential of ~ −20.4 mV) for long blood circulation[24–26] (Fig. 1c and Supplementary Fig. 2). X-ray photoelectron spectroscopy (XPS) analysis indicates that $Ce^{3+}$ and $Ce^{4+}$ co-exist on the surface of CNPs, providing the chemical basis for the catalytic activities[27–30] (Supplementary Fig. 3). Next, we investigated the antioxidant activity of CNPs under different pH conditions. CNPs demonstrated superoxide dismutase (SOD)-like activity under neutral and acidic conditions (Supplementary Fig. 4), which can catalyze the dismutation of superoxide radicals ($\cdot O_2^-$) into $H_2O_2$[31]. Intriguingly, CNPs are highly active for decomposing $H_2O_2$ in a neutral environment but inert under acidic conditions, as evaluated by oxygen production (Fig. 1d). Moreover, the CNPs-induced $H_2O_2$ decomposition follows Michaelis–Menten kinetics. As shown in Supplementary Table 1, the $K_M$ of CNPs to $H_2O_2$ under neutral conditions is 1.4 times lower than that of CNPs under acidic conditions, while the $V_{max}$ of CNPs under neutral conditions shows a 2.1-fold increase relative to CNPs under acidic conditions. To uncover the underlying mechanism, we incubated the CNPs with $H_2O_2$ under different pH conditions, and then examined the (111) planes of CNPs, on which the highest catalase-like activity normally occurs[32]. The X-ray powder diffraction (XRD) patterns show that the planes of CNPs (JCPDS 034-0394), particularly (111) planes, do not change after incubation (Supplementary Fig. 5), indicating the tunable catalytic activity is not based on the crystal lattice changes. Also, there is no obvious change in the size of CNPs after the incubation with $H_2O_2$ under different pH conditions (Supplementary Fig. 6). Next, the absorption of $H_2O_2$ and desorption of its products on the surface of CNPs were analyzed by Raman Spectroscopy. We found that the major peak at 460 cm$^{-1}$, a symmetric breathing mode of oxygen atoms around Ce, shifted to a new position centered at 850/880 cm$^{-1}$. This is likely attributed to the generation of O–O stretching of the adsorbed peroxide species on the surface of CNPs[33]. After a cycle of reaction, the major peak gradually shifts back to 460 cm$^{-1}$ under neutral conditions but not under acidic conditions (Fig. 1e).

Based on these results, a chemical mechanism for the tunable catalytic activity of CNPs can be proposed (Fig. 1f). Under the neutral conditions, a redox reaction is initiated between CNPs and surface-absorbed $H_2O_2$ ($H_2O_2 + 2Ce^{4+} \rightarrow O_2 + 2H^+ + 2Ce^{3+} + Vo$)[34]. The decomposition products of $H_2O_2$ are released to re-expose the surface of CNPs, and the $Ce^{3+}$ is oxidized to $Ce^{4+}$ for the next antioxidant cycle[34]. While under acidic conditions, excessive $H^+$ can inhibit the conversion of $Ce^{4+}$ to $Ce^{3+}$ that catalyzes the decomposition of surface-absorbed $H_2O_2$, which in turn disrupts the re-exposure of active catalytic sites and blocks the antioxidant cycles of CNPs[35]. Notably, inactive CNPs (iCNPs, $H_2O_2$, and $H^+$ pretreated CNPs) lose their anti-oxidant activity, which is difficult to reverse (Supplementary Fig. 7).

### CNPs with tunable catalytic activity exhibit context-dependent cytoprotective effect.

Encouraged by the tunable catalytic activity of CNPs in cell-free assays, we next examined whether they could exert different effects on living cells cultured under different conditions. Cisplatin (DDP) has been commonly utilized as an anti-tumor agent for various solid tumors[36,37], especially as a first-line drug for treating ovarian cancer[38], while its usage is limited by the potential nephrotoxicity. Considering that the renal epithelial cells are typical targets of DDP-induced nephrotoxicity, we used HK-2 cells (Human Renal Tubular Epithelial Cells) and ES-2 cells (Human Ovarian Cells) to study the pH-dependent cytoprotective effects of CeNPs on renal cells and tumor cells, respectively. Interestingly, the DDP-induced cytotoxicity is significantly reduced by CNPs at pH 7.4 but not affected at pH 6.6 or pH 6.0 (Fig. 2a, b, Supplementary Fig. 8). In contrast to CNPs, small molecular antioxidant NAC with pH-independent anti-oxidant activity resists the DDP-induced cytotoxicity even at pH 6.6 (Supplementary Fig. 9). Furthermore, the iCNPs show no cytoprotective effects on HK-2 cells and ES-2 cells under neutral conditions since their anti-oxidant activity is irreversibly lost upon pre-treatment with $H_2O_2$ under acidic conditions (Supplementary Fig. 10).

To verify the generality of the context-dependent cytoprotective effects of CNPs, we investigated the effects of CNPs on several other cell lines treated with DDP under neutral or acidic conditions. We found that the CNPs' pH-dependent cytoprotective effect against DDP held true in other tumor and normal cell lines, including another ovarian carcinoma cell line OVCAR8, hepatocellular

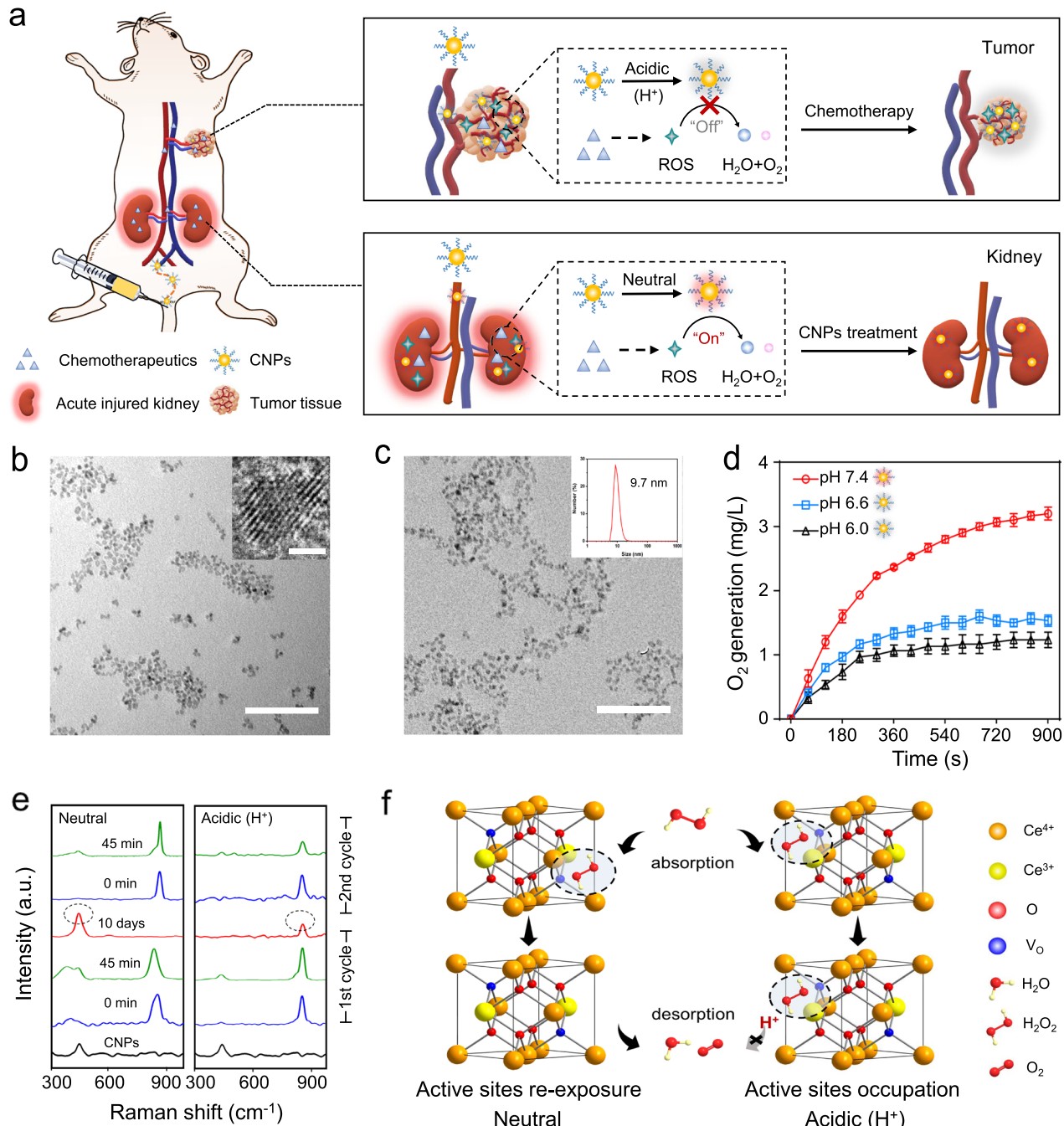

**Fig. 1 Design and characterization of catalytic activity tunable CNPs that protect against chemotherapy-induced AKI. a** Schematic illustration of catalytic activity tunable CNPs with context-dependent cytoprotective activities for AKI prevention during chemotherapy. High-concentration of chemotherapeutics in the kidneys induces AKI by producing excessive reactive oxygen species (ROS). In the renal cortex, the administered ultrafine CNPs are switched "on", and could counteract toxic ROS to prevent AKI. In the tumor acidic microenvironment, CNPs would be switched "off" by the high-level $H^+$, and exert no effect on intratumoral ROS, maintaining the efficacy of chemotherapy. **b** TEM image of ultrafine CNPs in chloroform, scale bar: 50 nm. Insert: high-resolution TEM image of CNPs, scale bar: 2 nm. **c** TEM image of ultrafine CNPs after DSPE-PEGlyation in water, scale bar: 50 nm. Insert: hydrodynamic diameter distribution of DSPE-PEGlyated CNPs. **d** The oxygen ($O_2$) production of CNPs under different pH conditions (pH 7.4, pH 6.6, and pH 6.0) during the reaction with $H_2O_2$. Data are presented as mean values ± SEM., $n = 3$ independent experiments. **e** Raman spectra of CNPs at different time points in each cycle of reaction with $H_2O_2$ under different pH conditions (pH 7.4 and pH 6.0). **f** Schematic illustration of the context-dependent catalase-like activity of CNPs under different pH conditions (pH 7.4 and pH 6.0). $V_O$, oxygen vacancy. In **b–c**, experiments were repeated three times independently. Source data are provided as a Source Data file.

carcinoma cell line HepG2, lung carcinoma cell line A549, and normal hepatocytes L02 (Fig. 2c, d, e, Supplementary Figs. 8 and 11). Moreover, CNPs can protect HK-2 cells and ES-2 cells from the cytotoxicity of paclitaxel (Taxol) under neutral conditions[39], which is another widely used oxidizing chemotherapeutic

agent, indicating the generic protective effects of CeNPs against ROS-related cellular damage (Fig. 2f, g).

**CNPs protect against chemotherapy-induced AKI in vivo.** We further evaluated the pharmacokinetics and biodistribution of CNPs

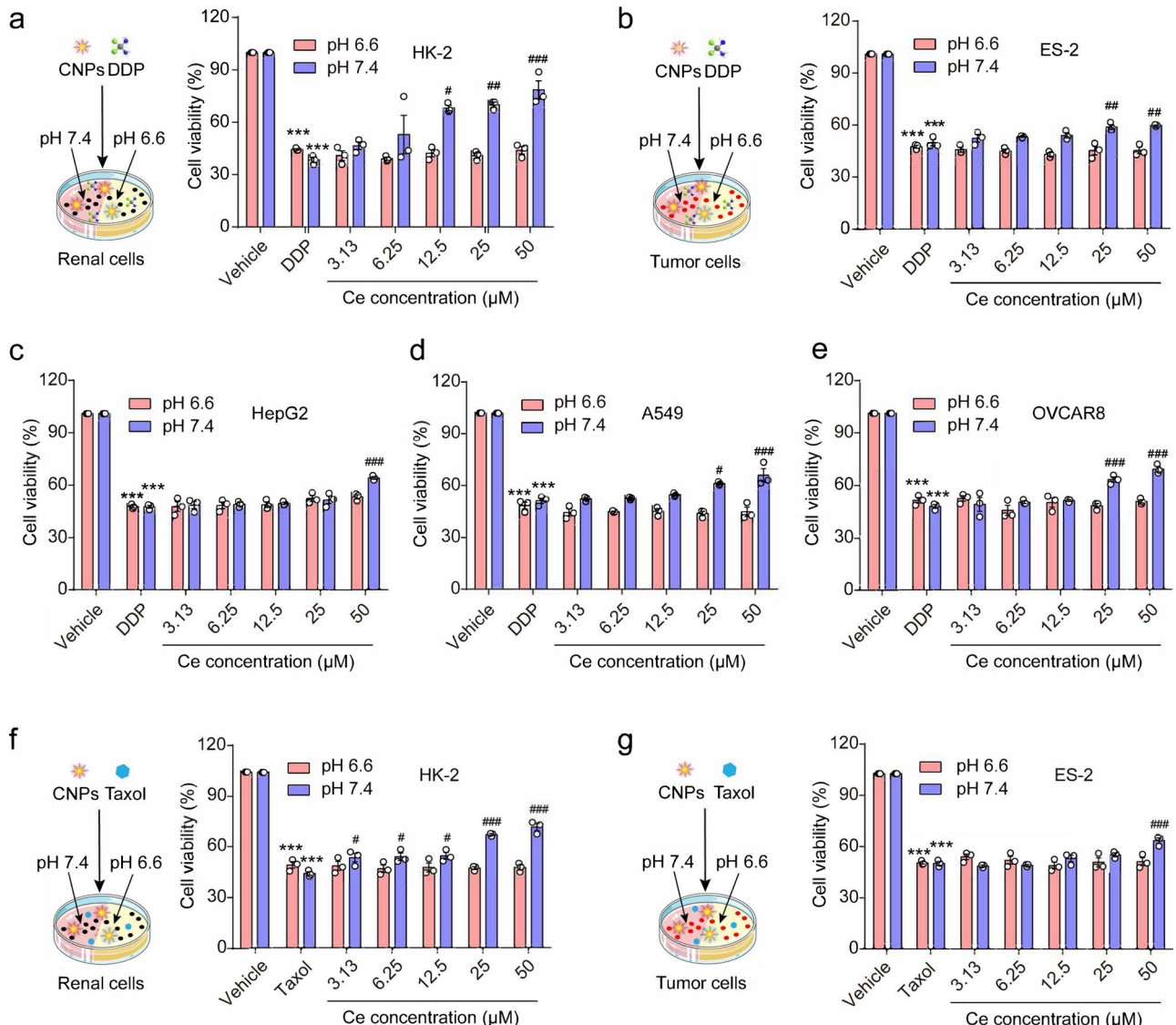

**Fig. 2 CNPs display context-dependent catalytic activity in vitro. a** The survival rate of HK-2 cells upon treatments with 10 μM DDP and different concentrations of CNPs at pH 7.4 and pH 6.6. $n = 3$ independent experiments, $P_{(DDP, 6.6)} = 3.3E-11$, $P_{(DDP, 7.4)} = 9.1E-06$, $P_{(12.5, 7.4)} = 0.0124$, $P_{(25, 7.4)} = 0.0078$, $P_{(50, 7.4)} = 0.00094$. **b** The survival rate of ES-2 cells upon treatments with 10 μM DDP and different concentrations of CNPs at pH 7.4 and pH 6.6. $n = 3$ independent experiments, $P_{(DDP, 6.6)} = 2.5E-11$, $P_{(DDP, 7.4)} = 3.7E-12$, $P_{(25, 7.4)} = 0.0046$, $P_{(50, 7.4)} = 0.0022$. **c** The survival rate of HepG2 cells upon treatments with 10 μM DDP and different concentrations of CNPs at pH 7.4 and pH 6.6. $n = 3$ independent experiments, $P_{(DDP, 6.6)} = 8.3E-11$, $P_{(DDP, 7.4)} = 2.4E-11$, $P_{(50, 7.4)} = 5.3E-05$. **d** The survival rate of A549 cells upon treatments with 10 μM DDP and different concentrations of CNPs at pH 7.4 and pH 6.6. $n = 3$ independent experiments, $P_{(DDP, 6.6)} = 8.3E-11$, $P_{(DDP, 7.4)} = 9.9E-11$, $P_{(25, 7.4)} = 0.01002$, $P_{(50, 7.4)} = 0.00024$. **e** The survival rate of OVCAR8 cells upon treatments with 10 μM DDP and different concentrations of CNPs at pH 7.4 and pH 6.6. $n = 3$ independent experiments, $P_{(DDP, 6.6)} = 6.9E-10$, $P_{(DDP, 7.4)} = 3.7E-10$, $P_{(25,7.4)} = 0.0009$, $P_{(50, 7.4)} = 2.8E-05$. **f** The survival rate of HK-2 cells upon treatments with 5 μM Taxol and different concentrations of CNPs at pH 7.4 and pH 6.6. $n = 3$ independent experiments, $P_{(DDP, 6.6)} = 7E-10$, $P_{(DDP, 7.4)} = 4.5E-11$, $P_{(3.13, 7.4)} = 0.0258$, $P_{(6.25,7.4)} = 0.0168$, $P_{(12.5, 7.4)} = 0.0116$, $P_{(25, 7.4)} = 3.7E-06$, $P_{(50, 7.4)} = 4.3E-07$. **g** The survival rate of ES-2 cells upon treatments with 5 μM Taxol and different concentrations of CNPs at pH 7.4 and pH 6.6. $n = 3$ independent experiments, $P_{(DDP, 6.6)} = 1.6E-09$, $P_{(DDP, 7.4)} = 6.8E-13$, $P_{(50, 7.4)} = 3.6E-05$. Data are presented as means ± SEM., \*\*\*$P < 0.001$ compared to vehicle, #$P < 0.05$, ##$P < 0.01$, ###$P < 0.001$ compared to DDP or Taxol; one-way ANOVA with multiple comparisons test. Source data are provided as a Source Data file.

in AKI mice. The result of the histological examination indicates that CNPs have no evident toxicity to major organs (Supplementary Fig. 12). As shown in Fig. 3a, ICR mice (8 male mice and 8 female mice) were intraperitoneally injected (i.p.) with a single dose of DPP (15 mg/kg) to induce AKI, accompanied by intravenous injection (i.v.) of saline, CNPs (0.5 or 1.5 mg/kg) or iCNPs (0.5 or 1.5 mg/kg), respectively. The blood circulation profiles of CNPs fit well with the classic two-compartment pharmacokinetic model, in which the

terminal elimination half-lives of the central component and peripheral component are 0.47 h and 72.2 h, respectively (Supplementary Fig. 13). The enhanced accumulation of CNPs in the kidneys of AKI mice is observed as compared to the normal mice (Supplementary Fig. 14), and CNPs exhibit long retention time in the renal cortex (Supplementary Fig. 15). Furthermore, CNPs are found in the renal tubules epithelial cell cilia, glomerular basement membrane (GBM), and renal tubules, as demonstrated by TEM

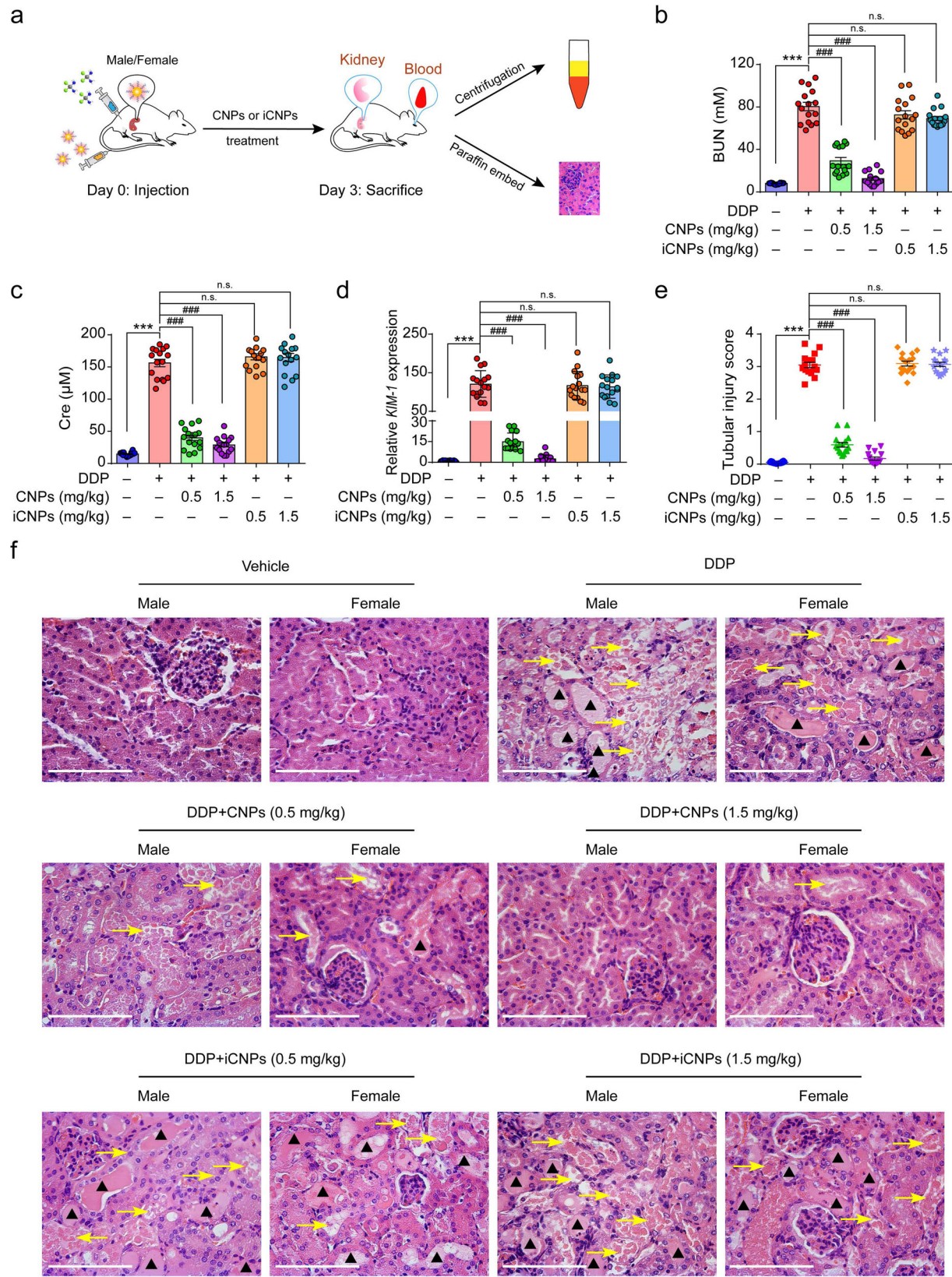

images of the renal cortex (Supplementary Fig. 16). The renal accumulation and retention of CNPs are likely due to the small core size (~3 nm), deformable PEG surface-modification, and negative surface charge of CNPs, as well as enhanced GBM permeability of injured kidney[40,41]. CNPs are also found in the urine of treated AKI mice, indicating that CNPs can filter through GBM to the renal tubules and be excreted through urine (Supplementary Fig. 17). Moreover, we studied the mechanism underlying the cellular endocytosis of CNPs in kidney cells. In consistent with previously reported PEGylated nanoparticles[42], the cellular uptake of CNPs is

**Fig. 3 CNPs protect against chemotherapy-induced AKI in vivo. a** Schematic representation of treatment schedule and therapy assessments for AKI mice. $n = 16$ independent animals, including 8 male mice and 8 female mice in each group. **b, c** Serum BUN (**b**) and Cre (**c**) levels of mice treated with vehicle, DDP, DDP plus CNPs or iCNPs (inactive CNPs), respectively. $n = 16$ independent animals. In **b**, $P_{(DDP)} = 4.4E{-}10$, $P_{(0.5\ CNPs)} = 4.4E{-}10$, $P_{(1.5\ CNPs)} = 4.4E{-}10$, $P_{(0.5\ iCNPs)} = 0.3664$, $P_{(1.5\ iCNPs)} = 0.0608$; in **c**, $P_{(DDP)} = 4.4E{-}10$, $P_{(0.5\ CNPs)} = 4.4E{-}10$, $P_{(1.5\ CNPs)} = 4.4E{-}10$, $P_{(0.5\ iCNPs)} = 0.6319$, $P_{(1.5\ iCNPs)} = 0.8457$. **d** Relative mRNA expression of *KIM-1* in the renal cortex from each group. $n = 16$ independent animals, $P_{(DDP)} = 4.4E{-}10$, $P_{(0.5\ CNPs)} = 4.4E{-}10$, $P_{(1.5\ CNPs)} = 4.4E{-}10$, $P_{(0.5\ iCNPs)} = 0.9985$, $P_{(1.5\ iCNPs)} = 0.9649$. **e** The tubular injury score was calculated according to the percentage of damaged tubules. A pathologist evaluated 5 randomly selected fields per section of the mouse kidneys at a magnification of ×400 in a blind manner. $n = 16$ independent animals, $P_{(DDP)} = 4.4E{-}10$, $P_{(0.5\ CNPs)} = 4.4E{-}10$, $P_{(1.5\ CNPs)} = 4.4E{-}10$, $P_{(0.5\ iCNPs)} = 0.9964$, $P_{(1.5\ iCNPs)} = 0.9999$. **f** Representative H&E sections of the kidney from male and female mice in each group. $n = 16$ independent animals. Arrows indicate tubules with necrosis, epithelial anoikis cavitation, or loss of brush border. Triangles denote the formation of casts in tubes. Scale bar: 100 μm. Data are presented as means ± SEM., \*\*\**P* < 0.001, ###*P* < 0.001, n.s., no significance; one-way ANOVA with multiple comparisons test. Source data are provided as a Source Data file.

reduced in the presence of MβCD, chlorpromazine, and amiloride, respectively, indicating that CNPs can be internalized into HK-2 cells via multiple pathways, among which the caveolin-mediated endocytosis plays a major role in the cellular uptake of CNPs[42] (Supplementary Fig. 18).

Then, the in vivo renal protective effect of CNPs was tested. The serum blood urea nitrogen (BUN) and creatinine (Cre), the classic indicators of clinical renal dysfunction[43], are drastically increased in mice treated with DDP alone, but prominently decreased in both CNPs co-treated male and female mice (Fig. 3b, c). Interestingly, the mRNA expression of *KIM-1*, a biomarker of kidney damage[44], is also downregulated in CNPs-treated groups (Fig. 3d), demonstrating the renal protective effect of CNPs in vivo. Furthermore, hematoxylin and eosin (H&E) staining of kidneys from DDP-treated mice exhibits severe renal tubular damage, as indicated by tubular necrosis, tubular dilatation, cast formation, loss of brush border, and epithelial degeneration. Notably, these structural damages are markedly ameliorated by CNPs treatment (Fig. 3e, f). In contrast, the iCNPs show no renal protection effects in DDP-treated mice. Moreover, the similar protective effect of CNPs was also observed in mice suffering cyclophosphamide (CP)-induced nephrotoxicity[45] (Supplementary Fig. 19), further demonstrating the generality of CNPs' in vivo therapeutic efficacy in chemotherapy-induced AKI.

**Anti-apoptosis effect of CNPs on AKI.** To elucidate the mechanism of AKI prevention, we investigated the impact of CNPs on cell apoptosis pathways. The apoptotic rate, as well as the levels of cleaved PARP and cleaved caspase-3 of HK-2 cells, are significantly increased by DDP but can be dramatically reduced via CNPs treatment, indicating the capability of CNPs in DDP-induced apoptosis prevention in vitro[46,47] (Fig. 4a–c). The anti-apoptotic effect of CNPs was further evaluated in vivo using TdT mediated dUTP Nick-End Labeling (TUNEL) assay, and we found that CNPs remarkably reduced the proportion of DDP-induced TUNEL positive cells in renal tissues according to the quantitative analysis (Fig. 4d, e). In contrast, iCNPs show no anti-apoptotic effect in vivo (Supplementary Fig. 20).

Considering ROS is a key factor in the induction of apoptosis[48,49], we investigated whether CNPs could exhibit their anti-apoptosis effect as a result of ROS scavenging. Firstly, CNPs show significant protective effects on HK-2 cells against $H_2O_2$, which is a salient member of the ROS family[50] (Supplementary Fig. 21). Next, consistent with the previous findings[51], DDP induced excessive ROS in HK-2 cells. Interestingly, the high ROS level is found significantly reduced after CNPs treatment (Fig. 4f, g, Supplementary Fig. 22), demonstrating that CNPs could scavenge DDP-induced intracellular ROS. Moreover, the activity of SOD is inhibited and the level of malondialdehyde (MDA) is elevated in HK-2 cells upon the DDP treatment, while the co-treatment with CNPs could reverse the imbalance of these ROS-related elements as induced by DDP (Fig. 4h, i). The relative expressions of antioxidant genes (*HO-1*) and

pro-oxidant genes (*NOX2*) in the renal cortex were further detected to confirm the protective effects of CNPs on the kidney. In accordance with the previous study[52], the qRT-PCR analysis demonstrates the upregulation of *HO-1* expression at 6 h and 12 h after the injection of DDP (15 mg/kg). However, the *HO-1* expression begins to decrease at 24 h and becomes even lower than the basal level at 72 h post-injection, which probably results from the oxidant injury induced by excessive ROS accumulation[53,54] (Supplementary Fig. 23). In addition, the *NOX2* expression is found to be upregulated at 72 h post-injection, which coincides with the high cellular ROS level[55]. Dramatically, the reduced *HO-1* and elevated *NOX2* expression are substantially reserved to basal level upon CNPs treatment (Fig. 4j, k).

**Molecular mechanism for the protective effects of CNPs on AKI.** To unravel the molecular mechanism underlying the capability of CNPs to restore redox homeostasis in AKI, we assessed the changes of key signaling molecules in the ROS regulatory network. It is known that *Nrf2* is a master regulator of the cellular antioxidant pathways, which would be decreased due to excessive ROS[51,56,57]. In this regard, western blot analysis was employed to measure the levels of Nrf2, as well as its related proteins, DJ-1 and Keap1 in vitro and in vivo. The results show that the expression of DJ-1, which is typically suppressed under oxidative stress[58], could be elevated by CNPs in vitro. Subsequently, Nrf2 is upregulated and Keap1 is downregulated[59,60]. There is no obvious change in the expressions of these proteins in HK-2 cells treated with CNPs alone (Fig. 5a). Consistently, the expressions of DJ-1 and Nrf2 are significantly augmented and Keap1 is decreased at the mRNA and protein levels in the renal cortex of CNPs-treated AKI mice (Fig. 5b–d and Supplementary Fig. 24).

To further characterize the role of *Nrf2* in CNPs-mediated antioxidant activity, we checked whether *Nrf2* silence would abrogate the protective effects of CNPs on DPP-poisoned HK-2 cells (Fig. 5e). Interestingly, once the *Nrf2* is depleted from HK-2 cells, CNPs would fail to suppress the expression of cleaved PARP or cleaved caspase-3 in DDP-treated HK-2 cells, indicating *Nrf2* is a critical factor for CNPs to prevent apoptosis (Fig. 5f). Moreover, the qRT-PCR analysis was conducted to profile the *DJ-1/Nrf2/Keap1* axis status. Consistent with the western blot analysis, the relative expressions of antioxidant genes and pro-oxidant genes in HK-2 cells with *Nrf2* knockdown are no longer sensitive to CNPs treatment (Fig. 5g, h). These results highlight the participation of CNPs in the activation of *Nrf2* to quench the ROS (Fig. 5i), and thus prevent apoptosis for renal cell protection.

**CNPs improve the overall chemotherapeutic outcome in vivo.** Encouraged by the tunable catalytic activity of CNPs for AKI prevention, we next assessed whether they could improve the overall therapeutic outcome in tumor-bearing mice receiving chemotherapy. DDP (3 mg/kg, twice a week) was intraperitoneally

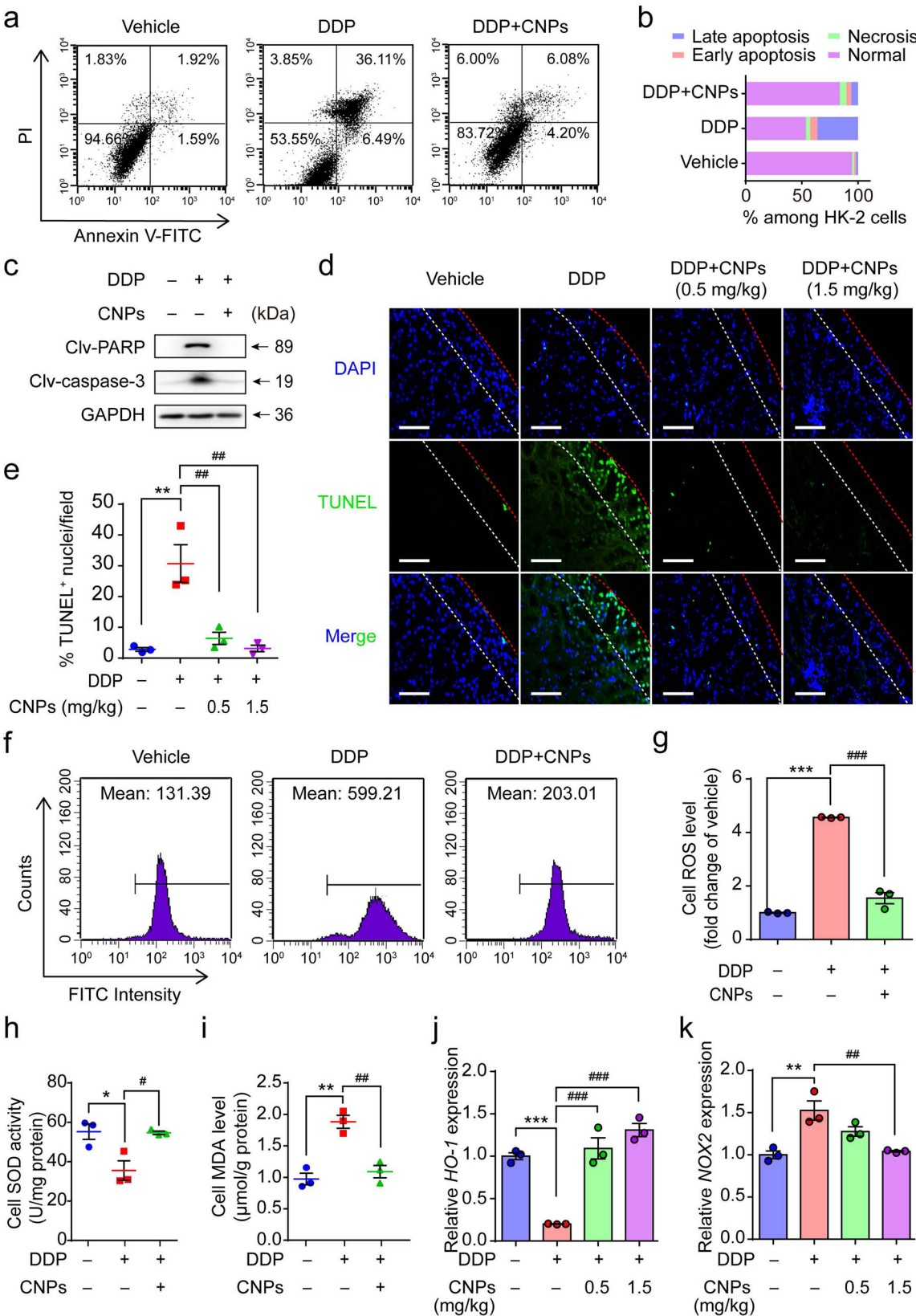

administered into ES-2 tumor xenograft nude mice, followed by CNPs or NAC treatment (Fig. 6a). The body weights of mice treated with DDP show negligible change as compared to that of the vehicle group (Supplementary Figs. 25 and 26), indicating that DDP at this dosage and administration frequency do not cause

severe toxicity that can threaten the survival of tumor-bearing mice. In accordance with in vitro results (Supplementary Figs. 9 and 27), the in vivo anti-tumor efficiency of DDP is found to be suppressed by NAC due to the quenched tumoricidal ROS, as confirmed by the enhanced SOD activity and reduced MDA level

**Fig. 4 CNPs protect against apoptosis by restoring the redox homeostasis. a** Flow cytometry results of HK-2 cells after treatment with vehicle and DDP with or without CNPs; cells were stained with Annexin V-FITC/PI. Representative images are shown. $n = 3$ independent experiments. **b** Proportion of normal, necrosis, early apoptosis, and late apoptosis in HK-2 cells. $n = 3$ independent experiments. **c** Western blot analysis of the cleaved PARP (clv-PARP) and cleaved caspase-3 (clv-caspase-3) levels in HK-2 cells after different treatments. GAPDH served as a loading control. $n = 2$ independent experiments. **d** Representative TUNEL staining images in kidneys. $n = 3$ independent mouse kidneys. Scale bar: 100 μm. The red dotted line indicates the boundary between the kidney edge and background, and the white dotted line represents the boundary between the renal cortex and the medulla. **e** Quantification of TUNEL positive cells in the respective groups. $n = 3$ independent mouse kidneys, $P_{(DDP)} = 0.0015$, $P_{(0.5\ CNPs)} = 0.0035$, $P_{(1.5\ CNPs)} = 0.0016$. **f** ROS content in HK-2 cells detected by flow cytometry analysis after different treatments. Representative images are shown. $n = 3$ independent experiments. **g** Statistical results of ROS level in HK-2 cells from each group. $n = 3$ independent experiments, $P_{(DDP)} = 1.8E{-}06$, $P_{(DDP+CNPs)} = 4.5E{-}06$. **h, i** Detection of SOD activities (**h**) and MDA levels (**i**) in HK-2 cells after different treatments. $n = 3$ independent experiments. In **h**, $P_{(DDP)} = 0.0206$, $P_{(DDP+CNPs)} = 0.0233$; in **i**, $P_{(DDP)} = 0.0015$, $P_{(DDP+CNPs)} = 0.00297$. **j, k** Relative mRNA expressions of antioxidant gene $HO{-}1$ (**j**) and pro-oxidant gene $NOX2$ (**i**) in the renal cortex of mice from each group. $n = 3$ independent experiments. In **j**, $P_{(DDP)} = 0.00034$, $P_{(0.5\ CNPs)} = 0.00016$, $P_{(1.5\ CNPs)} = 3.2E{-}05$; in **k**, $P_{(DDP)} = 0.0026$, $P_{(1.5\ CNPs)} = 0.0043$. Data are presented as means ± SEM., *$P < 0.05$, **$P < 0.01$, ***$P < 0.001$, #$P < 0.05$, ##$P < 0.01$, ###$P < 0.001$; one-way ANOVA with multiple comparisons test. Source data are provided as a Source Data file.

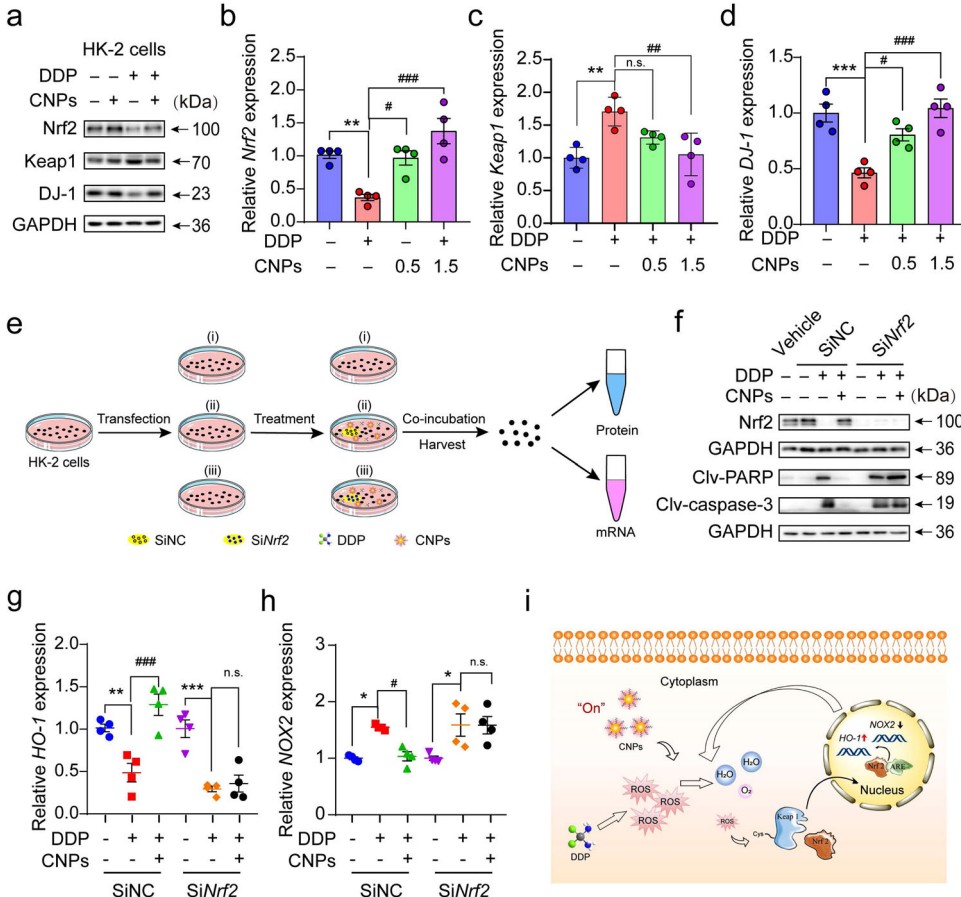

**Fig. 5 CNPs counteract DDP-induced oxidative stress via activating _Nrf2/Keap1_ signaling pathway. a** Western blot analysis of the Nrf2, Keap1, and DJ-1 levels in HK-2 cells after respective treatments with the vehicle, CNPs, DDP, and CNPs plus DDP. GAPDH served as a loading control. $n = 2$ independent experiments. **b, c, d** Relative mRNA expressions of _Nrf2_ (**b**), _Keap1_ (**c**), and _DJ-1_ (**d**) in the renal cortex from each group. $n = 4$ independent experiments. In **b**, $P_{(DDP)} = 0.0091$, $P_{(0.5\ CNPs)} = 0.015$, $P_{(1.5\ CNPs)} = 0.00026$; in **c**, $P_{(DDP)} = 0.003$, $P_{(0.5\ CNPs)} = 0.0977$, $P_{(1.5\ CNPs)} = 0.0054$; in **d**, $P_{(DDP)} = 0.0006$, $P_{(0.5\ CNPs)} = 0.0185$, $P_{(1.5\ CNPs)} = 0.00031$. **e** Schematic diagram of the experimental setup to evaluate the role of _Nrf2_ by small interfering RNA (SiRNA) in vitro using western blot and qRT-PCR analysis. **f** Western blot analysis of Nrf2, clv-PARP and clv-caspase-3 levels in SiNrf2-transfected HK-2 cells. GAPDH served as a loading control. $n = 2$ independent experiments. **g, h** Relative mRNA expressions of _HO-1_ (**g**) and _NOX2_ (**h**) in the respective groups. $n = 4$ independent experiments. In **g**, $P_{(siNC,\ DDP)} = 0.0092$, $P_{(siNC,\ DDP+CNPs)} = 0.0001$, $P_{(siNrf2,\ DDP)} = 0.00046$, $P_{(siNrf2,\ DDP+CNPs)} = 0.9958$; in **h**, $P_{(siNC,\ DDP)} = 0.0241$, $P_{(siNC,\ DDP+CNPs)} = 0.0382$, $P_{(siNrf2,\ DDP)} = 0.0139$, $P_{(siNrf2,\ DDP+CNPs)} = 1.000$. **i** Schematic representation of the mechanism of CNPs in reducing ROS level. CNPs decompose DDP-induced $H_2O_2$, a predominant cellular ROS, into $H_2O$ and $O_2$. Meanwhile, the minimally residual amounts of ROS activate the _Nrf2/Keap1_ signaling pathway. Specifically, Nrf2 moves into the nucleus and subsequently binds to the antioxidant response elements (ARE), leading to the upregulation of the antioxidant gene (_HO-1_) and downregulation of the pro-oxidant gene (_NOX2_). These gene regulations can further detoxify ROS. Each experiment was repeated three times independently. Data are presented as means ± SEM., *$P < 0.05$, **$P < 0.01$, ***$P < 0.001$, #$P < 0.05$, ##$P < 0.01$, ###$P < 0.001$, n.s., no significance; one-way ANOVA with multiple comparisons test. Source data are provided as a Source Data file.

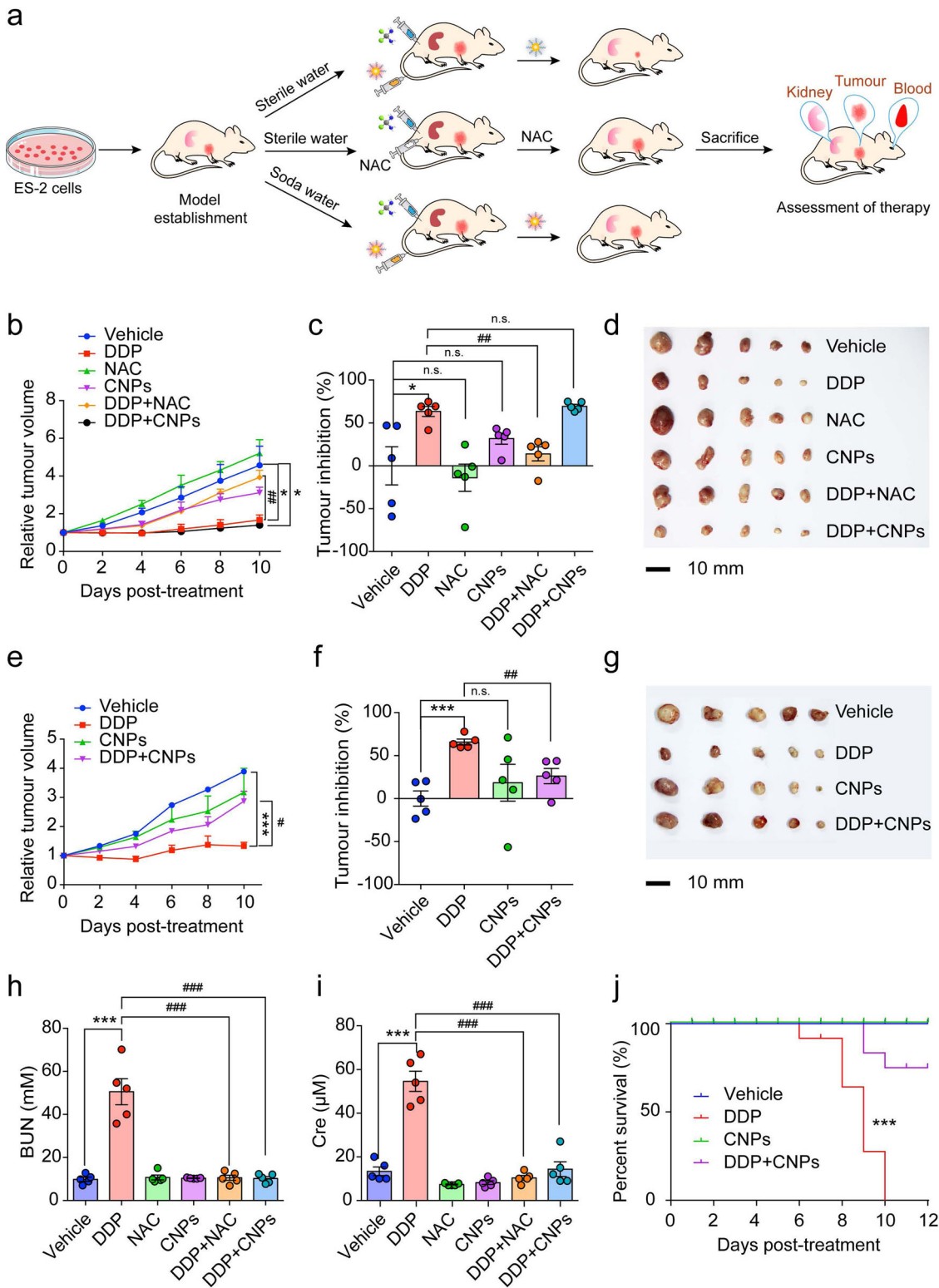

in the tumor tissues (Supplementary Fig. 28a,b). However, it is not affected by CNPs (Fig. 6b–d, Supplementary Fig. 29) whose ROS scavenging activities are turned off once exposed to the acidic tumor microenvironment, showing no obvious change in SOD activity and MDA level (Supplementary Fig. 28a,b). Intriguingly, CNPs can reverse the imbalance of ROS-related elements induced by DDP (Supplementary Fig. 30a,b), and thus weaken the anti-tumor efficiency of DPP when the intratumoral pH is artificially raised[61] (Fig. 6e, f, g). These in vivo results clearly show that, compared with NAC, CNPs can regulate the ROS scavenging activity in a context-dependent manner, thereby exerting no effect on the potency of chemotherapeutic agents in the acidic tumor microenvironment.

Both CNPs and NAC treatments remarkably reduce the serum BUN and Cre levels in tumor-bearing mice receiving DDP (Fig. 6h, i, Supplementary Fig. 31), indicating the successful

**Fig. 6 CNPs improve the overall chemotherapeutic outcome in vivo. a** Schematic diagram of ES-2 tumor xenograft establishment, treatment schedule, and therapy assessments. Groups of nude mice were treated with DDP (i.p., 3 mg/kg, twice one week, the total for twice) and CNPs (i.v., 1.5 mg/kg, twice one week, the total for twice) or NAC (oral administration, 400 mg/kg, daily), respectively. Saline was used as a negative control. Soda water was used to raise the intratumoral pH of mice. **b** Relative tumor volume of nude mice fed with sterile water in the respective groups. $n = 5$ independent animals, $P_{(DDP)} = 0.0245$, $P_{(DDP+NAC)} = 0.0012$, $P_{(DDP+CNPs)} = 0.0144$. **c** Tumor inhibition rate of nude mice fed with sterile water in each group. $n = 5$ independent animals, $P_{(DDP)} = 0.0246$, $P_{(NAC)} = 0.622$, $P_{(CNPs)} = 0.207$, $P_{(DDP+NAC)} = 0.0012$, $P_{(DDP+CNPs)} = 0.365$. **d** Images of the dissected tumors from groups of nude mice fed with sterile water after treatment. Scale bar: 10 mm. $n = 5$ independent animals. **e** Relative tumor volume of nude mice fed with soda water (200 mM, begun on the first day of dividing groups) in the respective groups. $n = 5$ independent animals, $P_{(DDP)} = 0.0001$, $P_{(DDP+CNPs)} = 0.044$. **f** Tumor inhibition rate of mice fed with soda water in each group. $n = 5$ independent animals, $P_{(DDP)} = 0.0001$, $P_{(CNPs)} = 0.4503$, $P_{(DDP+CNPs)} = 0.0031$. **g** Images of the dissected tumors from groups of nude mice fed with soda water after treatment. Scale bar: 10 mm. $n = 5$ independent animals. **h, i** Serum BUN (**h**) and Cre (**i**) levels of nude mice were measured on the last day. $n = 5$ independent animals. In **h**, $P_{(DDP)} = 1.1E{-}09$, $P_{(DDP+NAC)} = 1.6E{-}09$, $P_{(DDP+CNPs)} = 1.4E{-}09$; in **i**, $P_{(DDP)} = 4.7E{-}10$, $P_{(DDP+NAC)} = 1.1E{-}10$, $P_{(DDP+CNPs)} = 7.7E{-}10$. **j** The survival probability of tumor-bearing nude mice treated with DDP (i.p., 3 mg/kg, twice one week, the total for three times) and (or) CNPs (i.v., 1.5 mg/kg, twice one week, the total for three times) throughout the observation period. The green line overlaps the blue line. $n = 11$ independent animals. $P_{(DDP+CNPs)} = 5E{-}06$. The statistical significance was calculated by the log-rank (Mantel-Cox) test. Data are presented as means ± SEM., *$P < 0.05$, ***$P < 0.001$ compared to vehicle, #$P < 0.05$, ##$P < 0.01$, ###$P < 0.001$ compared to DDP; In **b–c**, **e–f**, statistical significance was calculated by two-tailed Student's t-test; **h–i**, one-way ANOVA with multiple comparisons test. Source data are provided as a Source Data file.

kidney protection. The H&E staining further demonstrates that the DDP-induced tubular damage is significantly rescued by CNPs treatment (Supplementary Fig. 32). Finally, we investigated the effect of CNPs on the survival of tumor-bearing mice receiving more frequent administration of DDP (3 mg/kg, three times a week). None of the mice with high-frequency DDP treatment can survive more than 10 days due to the severe side effects[62]. In stark contrast, there are much better outcomes (7 survived out of 11) for mice concurrently treated with CNPs (Fig. 6j).

## Discussion

Acute kidney injury is a ROS-related disease with high mortality, which frequently occurred in clinical cancer patients receiving chemotherapy[63]. However, no suitable therapeutic option has been established to protect against AKI, which hinders cancer patients from reaping the benefits of chemotherapy[1,3]. Although both AKI and cancer are ROS-associated diseases, completely opposite treatment strategies in the aspect of ROS level regulation are needed. Small molecular antioxidants and ROS scavenging nanomedicines have been reported to exhibit positive effects on relieving clinical symptoms of AKI[14,15], but at the risk of causing interference with chemotherapeutics agents whose treatment efficacies rely on intracellular ROS upregulation.

Recently, ceria nanoparticles with antioxidant activities have been widely studied in biomedical applications[64,65], such as pro-angiogenesis[66] and anti-inflammatory[67] therapies. In this study, CNPs with tunable catalytic activity were found to display context-dependent antioxidant efficiency to reconcile the divergent needs in tumor therapy and renal protection (Fig. 7). CNPs exhibit SOD-like activity under both neutral and acidic conditions to catalyze the dismutation of $\cdot O_2{}^-$ into $H_2O_2$, which is another kind of toxic ROS. Notably, CNPs are highly active for decomposing $H_2O_2$ under neutral conditions while become inert under acidic conditions. The in vitro results show that CNPs can exert a generic cytoprotective effect on diverse cells challenged by different oxidizing chemotherapeutics under neutral conditions, but without compromising their anti-tumor potency under acidic conditions commonly present in tumor microenvironments (Fig. 2). After systemic administration, the CNPs possess anti-oxidant activity for kidney protection in chemotherapeutic agents-induced AKI (Fig. 3). Moreover, the protective effect of CNPs is also observed in ischemia-reperfusion (IR)-induced AKI, which involves excessive ROS generation (Supplementary Fig. 33). In contrast, iCNPs with deficient ROS scavenging ability have no protective effects against DDP-induced AKI in vitro and

in vivo (Fig. 3, Supplementary Figs. 10 and 20), indicating the indispensable role of CNPs' antioxidant activity for AKI protection. However, such an antioxidant activity of CNPs is selectively suppressed in the acidic tumor microenvironment, where the high concentration of tumor extracellular $H^+$ disrupts the re-exposure of active catalytic sites and blocks the antioxidant cycles[34,35]. Consequently, CNPs also show no interference with the potency of chemotherapeutic agents in vivo (Fig. 6). These in vitro and in vivo results demonstrate the generality of the context-dependent ROS modulation activity of CNPs and their capability to protect against chemotherapeutic agents-induced AKI without compromising the potency of chemotherapy. Further mechanism studies demonstrate that CNPs prevent chemotherapeutic agents induced apoptosis of renal cells in AKI by detoxing $H_2O_2$, as well as activating the *Nrf2/Keap1* signaling pathway to regulate ROS-related genes for the restoration of redox homeostasis in the renal cells[68].

Finally, our results show that CNPs could significantly enhance the survival rate of tumor-bearing mice treated with a high dose of DDP, indicating their great potential to broaden the chemotherapeutic window, and thus allow cancer patients to reap the desired benefits of chemotherapy. Considering the facile preparation and context-dependent catalytic activity, we anticipate the ultrafine CNPs would be promising candidates for the prevention and treatment of clinical AKI, as well as other ROS-related diseases in patients receiving chemotherapy, thus improving the overall therapeutic efficacy.

## Methods

**Materials.** Cerium(III) acetate (99.99% metals basis), Rhodamine B Isothiocyanate (RITC), chlorpromazine, amiloride, and methyl-β-cyclodextrin (MβCD) were purchased from Aladdin. Oleylamine (technical grade, 70%), xylene (98.5%), and N-acetyl cysteine (NAC) were purchased from Sigma-Aldrich. Cisplatin injection (5 mg/mL) and cisplatin powder were purchased from Jiangsu Haosen Pharmaceutical Co., Ltd. 1,2-distearoyl-sn-glycero-3-phosphoethanolamine-N-[methoxy (polyethylene glycol)-2,000] (DSPE-PEG$_{2K}$) and DSPE-PEG$_{2K}$-NH$_2$ were purchased from Shanghai AVT Pharmaceutical Technology Co., Ltd. Hydrogen peroxide ($H_2O_2$) was purchased from Sinopharm Chemical Reagent Co., Ltd. 1640 medium, McCoy's 5 A medium, F12 medium, DMEM medium and fetal bovine serum (FBS) were provided by Gibco BRL (Invitrogen).

**Synthesis of ceria nanoparticles (CNPs).** Cerium(III) acetate hydrate (0.43 g, 1 mmol) and oleylamine (3.25 g, 12 mmol) were added into xylene (15 mL), which was stirred at room temperature until its color turned semitransparent brown. The resulting solution was heated to 90 °C under an argon environment. Subsequently, 1 mL deionized water was injected and the mixed solution was aged at 90 °C for 3 h with magnetic stirring (800 revolutions per minute, IKA, C-MAG-HS4, Germa). Acetone was used to precipitate the CNPs and the collected CNPs were dispersed in 10 mL chloroform for further use.

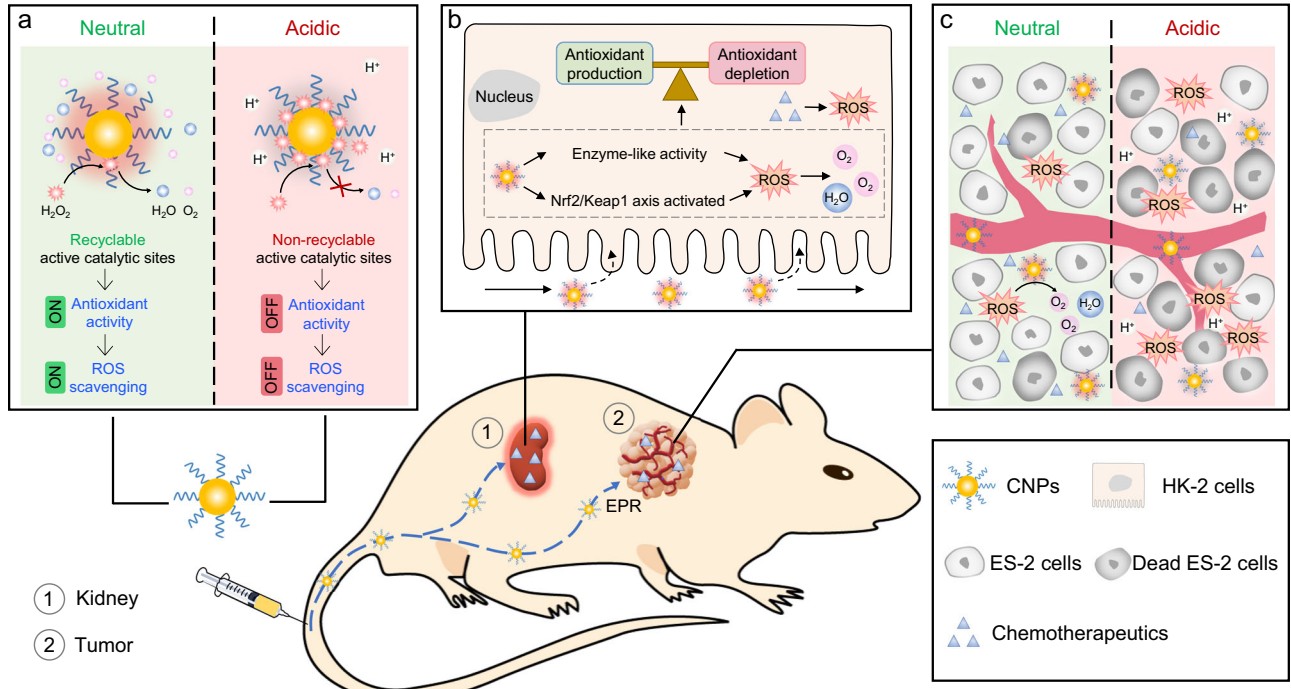

**Fig. 7 Schematic illustration of catalytic activity tunable CNPs that context-dependently regulate ROS in vivo. a** CNPs switch their antioxidant activity in a pH-dependent manner. Under neutral conditions, CNPs surface-absorbed $H_2O_2$ can be decomposed, re-exposing the active catalytic sites of CNPs for the next cycle of catalysis. However, the decomposition of $H_2O_2$ can be inhibited by excessive $H^+$, and the surface-absorbed $H_2O_2$, in turn, disrupts the re-exposure of active catalytic sites, and thus blocks the antioxidant cycles. **b** In the kidney of tumor-bearing mice, CNPs scavenge the excessive chemotherapeutics-induced ROS by decomposing $H_2O_2$ and activating the *Nrf2/Keap1* signal pathway, restoring the balance between antioxidant production and antioxidant depletion for satisfactory kidney protection. **c** Once CNPs accumulate in tumor tissues, the acidic tumor microenvironment suppresses their ROS scavenging capability, thus without causing interference with the chemotherapeutic efficiency.

**Synthesis of DSPE-PEG$_{2K}$ modified CNPs**. To impart the colloidal stability of the CNPs in aqueous solution, 20 mg of DSPE-PEG$_{2K}$ and 5 mL of 1 mg/mL CNPs in chloroform were mixed and stirred for 10 min at room temperature. Then, the mixture was evaporated by a rotary evaporator at 60 °C for 3 h. After that, 5 mL of distilled water was added to re-disperse the dried mixture and filter out the free DSPE-PEG using an ultrafiltration tube (MW = 30,000), obtaining DSPE-PEG$_{2K}$-coated CNPs. The same procedure was applied to fabricate RITC-labeled CNPs using DSPE-PEG-RITC.

**Characterization**. Transmission electron microscopy (TEM) images of CNPs and the collected urine samples were taken using HITACHI HT7700 (Tokyo, Japan) at a voltage of 120 kV. High-resolution transmission electron microscopy (HRTEM) image was taken with an FEI Tecnai F20 (FEI, USA) at a voltage of 200 kV. TEM images of kidney tissues were taken using HITACHI H-7650 (Tokyo, Japan) at a voltage of 80 kV. The thermogravimetric analysis (TGA) of DSPE-PEG$_{2K}$ modified or unmodified CNPs was analyzed by a METTLER TOLEDO TGA/DSC1 thermogravimetric analyzer system. The Fourier transform infrared (FT-IR) spectra of unmodified CNPs, physical mixture of DSPE and PEG-COOH, DSPE-PEG, and CNPs were analyzed by NicoLET iS50FT-IR. The hydrodynamic size and zeta potentials of DSPE-PEG$_{2k}$ modified CNPs were detected using a Zetasizer Nano ZS90 (Malvern Instruments, Worcestershire, UK). The hydrodynamic size of DSPE-PEG$_{2k}$ modified CNPs in the presence of $H_2O_2$ under different pH conditions was detected using a Zetasizer Nano ZS90 (Malvern Instruments, Worcestershire, UK). The X-ray photoelectron spectra (XPS) were obtained via a Thermo Scientific ESCALAB 250 Xi XPS system. The X-ray powder diffraction (XRD) patterns were detected on a Rigaku D/Max-2550 PC instrument (Rigaku, Japan). Raman spectra were obtained by using an inVia Reflex Raman spectrometer (Renishaw, UK). The concentration of Ce was quantified using the Inductively coupled plasma-Mass Spectrometry (ICP-MS, PerkinElmer NexION 300X).

**The SOD-like activity of CNPs**. The SOD-like activity of CNPs at different concentrations (50 μM, 100 μM, 150 μM, and 400 μM) was measured by a Total Superoxide Dismutase Assay Kit with WST-8 (Beyotime, China) under different pH conditions.

**The catalase-like activity of CNPs**. Catalase-like activity assays of CNPs under different pH conditions were carried out at room temperature and the generated oxygen was measured using a specific oxygen electrode on Multi-Parameter Analyzer (JPSJ-606L, Leici China). Specifically, materials were added into 8.0 mL

buffer solution (0.1 M PBS buffer, pH 7.4, pH 6.6, or pH 6.0) in the order of 0.8 mL 30% $H_2O_2$ solution and 50 μM CNPs. The generated $O_2$ (mg/L) was measured at different reaction times. In addition, CNPs were incubated with $H_2O_2$ solution under the acidic conditions for 15 min and then washed with deionized water for 3 times. The pre-treated CNPs (50 μM) and 0.8 mL 30% $H_2O_2$ were added into 8.0 mL buffer solution (0.1 M PBS buffer, pH 7.4). The generated $O_2$ (mg/L) was measured at different reaction times. The kinetic assays of CNPs with $H_2O_2$ substrate were performed by adding different amounts (25, 100, 200 μL) of 30% $H_2O_2$ solution into buffer solution (0.1 M PBS buffer, pH 7.4 or pH 6.6), and the final total volume of solution is 8.0 mL. The Michaelis–Menten constant was calculated according to the Michaelis–Menten saturation curve by GraphPad Prism 8.0 (GraphPad Software).

**Cell culture**. HK-2 cells (Human Kidney Tubular Epithelial Cells), L02 cells (Hepatocytes), ES-2 and OVCAR8 cells (Human Ovarian Carcinoma Cells), HepG2 cells (Hepatocellular Carcinoma Cells), and A549 cells (Lung Carcinoma Cells) were purchased from American Type Culture Collection (ATCC, Manassas, VA). HK-2 and OVCAR8 cells grown in 1640 medium (10% FBS and 1% penicillin/streptomycin), ES-2 cells grown in McCoy's 5A medium, L02 and HepG2 cells grown in DMEM medium, A549 cells grown in F-12 medium were at 37 °C under a humidified atmosphere with 5% $CO_2$.

**Cell viability assay**. HK-2, ES-2, L02, HepG2, A549, and OVCAR8 cells were seeded into 96-well plates at 5000 cells/well for 24 h. The culture solution was adjusted to pH 6.0 or pH 6.6 or pH 7.4 with hydrochloric acid and sodium hydroxide. Then, 5 μM paclitaxel (Taxol) or 10 μM DDP and different concentrations of CNPs (3.13, 6.25, 12.5, 25, and 50 μM) were added into the 96-well plate for the incubation of another 24 h. Specifically, CNPs were dissolved in deionized water and DDP was dissolved in DMSO. The solution of CNPs and DDP were added into a basic medium containing 10% FBS, which was then incubated with cells. The solvent used as a vehicle is the basic medium containing the same volume of deionized water and DMSO as that of CNPs and DDP solution, respectively.

HK-2 cells were seeded into a 96-well plate at 5000 cells/well for 24 h. Then, 50 μM $H_2O_2$ and different concentrations of CNPs (0.78, 1.56, 3.13, 6.25, 12.5, 25, and 50 μM) were incubated with cells, and co-incubated in the medium for another 24 h. The solvent used as a vehicle is the basic medium containing the same volume of deionized water as that of CNPs and $H_2O_2$ solution, respectively.

HK-2 and ES-2 cells were seeded into 96-well plates at 5000 cells/well for 24 h. Then, 10 μM cisplatin and (or) different concentrations of inactive CNPs (iCNPs) (3.13, 6.25, 12.5, 25, and 50 μM) were incubated with cells, and co-incubated in the medium for another 24 h. The solvent used as a vehicle is the basic medium containing the same volume of deionized water as that of iCNPs and DDP solution, respectively.

HK-2 cells were seeded into 96-well plates at 5000 cells/well for 24 h. Then cells were treated with 10 μM DDP, 50 μM NAC, 50 μM CNPs, 10 μM DDP plus 50 μM CNPs, as well as 10 μM DDP plus 50 μM NAC, and co-incubated for another 24 h at pH 6.6. The solvent used as a vehicle is the basic medium containing the same volume of deionized water and DMSO as that of CNPs or NAC and DDP solution, respectively.

A colorimetric SRB assay was performed to examine the cell viability. In detail, SRB (4 mg/mL) was added into 96-well plate after various treatments, then they were incubated for 30 min. Discard the liquid in a 96-well plate and wash it with 1% acetic acid more than 5 times. A microplate reader (Thermo, Fisher Scientific, Waltham, MA) was used to measure the absorbance at 515 nm. Each sample was repeated in triplicate.

**Endocytic pathway of CNPs.** To study the endocytic pathway involved in CNPs internalization, three specific endocytic inhibitors were used: (1) chlorpromazine, an inhibitor of clathrin-mediated endocytosis; (2) amiloride, an inhibitor of micropinocytosis; (3) methyl-β-cyclodextrin (MβCD), an inhibitor of caveolin-mediated endocytosis. HK-2 cells were preincubated in serum-free RPMI 1640 medium with chlorpromazine (30 μM, 30 min), amiloride (100 μM, 30 min), or MβCD (5 mM, 30 min), respectively. The medium was then changed to a fresh serum-free medium containing the inhibitors plus CNPs (50 μM) and further incubated for 7 h at 37 °C. The cells were then washed with 1× PBS and imaged by using a confocal laser scanning microscope (Leica TCS SP8, Germany). The Image J was used for quantitative analysis.

**AKI model establishment.** All animal use and studies were performed in compliance with all relevant ethical regulations. The procedures conducted on animals were approved by the Institutional Animal Care and Use Committee (IACUC) of Zhejiang University. ICR mice (male and female, 6–8 weeks) were obtained from Beijing Vital River Laboratory Animal Technology Co., Ltd. All mice were housed in a specific pathogen-free environment at 21 ± 1 °C and 60 ± 5% humidity, with a 12 h light-dark cycle. (1) To establish a DDP-induced acute kidney injury model, ICR mice received a single intraperitoneal injection of cisplatin at 15 mg/kg. (2) To establish a cyclophosphamide (CP)-induced acute kidney injury model, ICR mice received 100 mg/kg cyclophosphamide by intraperitoneal injection, twice in total for two consecutive days. (3) To establish an IR-induced AKI model, ICR mice were anesthetized with an intraperitoneal injection of 1% pentobarbital sodium and were subjected to bilateral renal artery occlusion with non-traumatic vascular clips for 40 min before reperfusion. In the sham group, mice underwent the same operation without renal artery clamping.

**Treatment of AKI mice.** (1) The DDP-induced AKI mice were intravenously injected with different dosages of CNPs (0.5 or 1.5 mg/kg) or iCNPs (0.5 or 1.5 mg/kg). Control group animals were intraperitoneally injected with an equal volume of saline. All mice were sacrificed at 72 h after cisplatin injection, and their blood and kidneys were collected for evaluation of renal function and tissue damage. Furthermore, organs such as the heart, liver, spleen, lungs, and kidneys were also harvested. (2) The CP-induced AKI mice were intravenously injected with saline or 1.5 mg/kg CNPs once simultaneously with the first CP injection. All mice were sacrificed at 48 h after intravenous injection of CNPs (or at 24 h after the second injection of CP). (3) The ICR mice were intravenously injected with CNPs (1.5 mg/kg) or saline 3 h prior to the establishment of the IR-induced AKI model. The blood and kidneys of mice were collected and evaluated for kidney function at 24 h after IR.

**Pharmacokinetics and biodistribution of CNPs.** To evaluate the blood circulation half-life of CNPs, ICR mice (Male, $n = 3$) with established cisplatin-induced AKI model were intravenously injected with CNPs (1.5 mg/kg). At different time points post-injection (1 min, 10 min, 1 h, 4 h, 8 h, and 24 h), 15 μL of plasma samples were collected from the mouse tail. The concentrations of CNPs in collected plasma samples were quantified by using ICP-MS. A two-compartment pharmacokinetic model was utilized to calculate the pharmacokinetic parameters of CNPs. Furthermore, the urine of mice was collected at 2 h after intravenous injection of CNPs for TEM analysis.

To evaluate the biodistribution of CNPs in the renal cortex, AKI mice ($n = 3$) were intravenously injected with CNPs (1.5 mg/kg). The mice were sacrificed to harvest kidneys at 1 h, 2 h, 8 h, 24 h, 48 h, and 72 h after injection, respectively. The renal cortex cut from the kidney was weighted and dissolved in nitric acid for ICP-MS measurement.

Three mice were sacrificed at different time points after CNPs injection (10 min, 1 h, 72 h), and a small piece of the renal cortex was collected and fixed with the 2.5% glutaraldehyde for 24 h, then post-fixed with 1% osmium tetroxide and prepared for TEM analysis.

**Blood biochemical analysis for kidney function.** Blood urea nitrogen (BUN) and creatinine (Cre) levels were tested using a fully automatic blood biochemical analyzer (Roche, Cobas c311, Switzerland).

**Hematoxylin and eosin (H&E) staining.** The collected kidney tissues were fixed in 4% paraformaldehyde (PFA), embedded in paraffin wax, and cut into 3 μm-thick tissue sections. Tubular damage was assessed by lumen dilatation, epithelial necrosis, cast formation, and loss of brush border. Randomly selecting five fields of view for scoring at 400×, and histopathological changes were scored semi-quantitatively on a 0 to 4 scale in a blinded manner based on the percentage of damaged tubules (0, no damage; 1, <25% damage; 2, 25–50% damage; 3, 50–75% damage; 4, >75% damage).

**RNA extraction and qRT-PCR.** To study the protective effect of CNPs on the kidneys, 15 mg/kg DDP and 0.5 mg/kg or 1.5 mg/kg CNPs were administered, respectively. After 72 h, the mice were sacrificed and the total RNA was extracted from the kidneys of each group to detect *KIM-1*, *HO-1*, *NOX2*, *Nrf2*, *Keap,1*, and *DJ-1* mRNA levels.

HK-2 cells were transfected with SiNrf2 mixed with Lipofectamine 2000 transfection reagent. The medium (Opti-MEM) was replaced at 6 h post-transfection. After 24 h, the cells were treated with 10 μM DDP and 50 μM CNPs. After another 24 h, HK-2 cells were collected by Trizol regent to extract RNA to measure the expressions of *HO-1* and *NOX2* mRNA.

Total RNA was isolated from kidney tissues using Trizol reagent (Invitrogen, Carlsbad, California, USA), and cDNA was prepared using a cDNA preparation supermix (TransGen Biotech, Beijing, China). Quantitative RT-PCR was carried out using SYBR Green Supermix (Bio-Rad, Hercules, California, USA) with 20 μL reaction mixtures. Primer names and sequences for qRT-PCR are listed in Supplementary Table 2. PCR reactions were performed on the Quant Studio 6 Flex Real-Time PCR System (Applied Biosystems, Carlsbad, California, USA) with the following program: step 1, 95 °C for 3 min to activate the Taq polymerase; step 2, 95 °C for 3 s to denaturize DNA; step 3, 60 °C for 31 s for annealing/extension (39 cycles for steps 2 and 3). The relative mRNA levels were quantified by the $2^{-\Delta\Delta Ct}$ method and all data were normalized to GAPDH (the internal control).

**The concentration of CNPs in kidneys and tumors.** ICR mice were intraperitoneally administered with cisplatin (15 mg/kg) and intravenously administered with CNPs (0.5 or 1.5 mg/kg) simultaneously, and the kidneys were harvested after 1 h, 24 h, 48 h, 72 h, respectively. BALB/c nude mice were intraperitoneally administered with cisplatin (3 mg/kg) and intravenously administered CNPs (1.5 mg/kg) twice one week, and the tumor tissues were harvested at 10 days after injection. Thereafter, the quantifiable amounts of kidneys and tumors were mixed with aqua regia and incubated at 60 °C for 12 h. After dilution and filtration, the Ce concentrations were quantified by ICP-MS analysis.

**Western blot analysis.** Immunoblotting analysis of proteins in HK-2 cells and tissue lysates was performed by homogenizing frozen tissues in the lysis buffer containing 50 mM Tris-HCl, 150 mM sodium chloride, 2 mM ethylene diamine tetraacetic acid (EDTA), 2 mM ethylene glycol tetraacetic acid (EGTA), 1% Triton-X 100, and a protease inhibitor (pH 7.4). The supernatants of the mixture were obtained by centrifugation at 4 °C (30 min, 16,000 × *g*), and their total protein concentrations were further determined using the BCA Protein Assay kit (Yeasen Biotech Co., Ltd., Hong Kong, China). Protein extracts were separated on sodium dodecyl sulfate-polyacrylamide gel electrophoresis (SDS-PAGE, 8-12% gels) and blotted onto PVDF membranes (Merck Millipore Ltd., Massachusetts, USA). After blocking with 5% fat-free milk, the membranes were incubated at 4 °C overnight with the following primary antibodies: anti-Nrf2 (1:1000, ab62352), anti-Nrf2 (1:1000, ab137550), anti-Keap1 (1:1000, ab119403), anti-HO-1 (1:1000, ab68477) and anti-cleaved caspase-3 (1:1000, ab49822) from Abcam (Cambridge, UK); anti-cleaved PARP (1:1000, ET1608-10, human) from HuaAn biotechnology Co., Ltd. (Hangzhou, China), anti-cleaved PARP (1:1000, 9544 s, mouse) from Cell Signaling Technology (USA); anti-DJ-1 (1:1000, sc-55572) and anti-Actin (1:1000, sc-1615) from Santa Cruz Biotechnology (Dallas, Texas, USA); and anti-GAPDH (1:1000, db106) from Diag Biotechnology Co., Ltd. (Hangzhou, China). The bound antibodies were detected using horseradish peroxidase (HRP)-conjugated IgG (Multi Sciences) and visualized with enhanced chemiluminescence (ECL) detection reagents (FDbio Science Biotech Co., Ltd., Hangzhou, China). GAPDH was used as a loading control.

**Flow cytometry analysis of cell apoptosis.** HK-2 cells were cultured in 6-well plates ($2 \times 10^5$ per well) for 24 h. Afterward, the culture mediums were, respectively, treated with vehicle (DMSO), DDP (10 μM), as well as DDP plus CNPs (50 μM), and incubated for another 24 h. Cells were collected and incubated with a FITC Annexin V Apoptosis Detection Kit (BD Pharmingen™, USA) for 15 min at room temperature, and then resuspended with 400 μL binding buffer. The intensity of fluorescence was measured by a flow cytometer (BD Biosciences, FACSuite™, USA) within 1 h. The gate strategies were shown in Supplementary Fig. 35.

**TUNEL staining**. Paraffin sections were first dewaxed in xylene for 5–10 min, then replaced with fresh xylene, and dewaxed again for 5–10 min. The resulting sections were treated with anhydrous ethanol for 5 min, 90% ethanol for 2 min, 70% ethanol for 2 min, distilled water for 2 min, in proper order. The deparaffinized sections of the kidney cortex were successively incubated with 20 μg/mL DNase-free proteinase K (20 min, room temperature) and TdT reaction mix (60 min, 37 °C) in the dark. Then, they were washed three times by 1× PBS and counterstained with 4',6-Diamidino-2-phenylindole (DAPI). The obtained sections were observed by a fluorescence microscope (Leica, DMI3000B, Germany).

**Cellular ROS detection**. HK-2 cells were cultured in 6-well confocal plates ($2 \times 10^5$ cells per well). The cells were treated with 10 μM cisplatin and 50 μM CNPs, then co-incubated for 24 h. Cellular ROS was detected using a fluorescence probe (2,7-dichlorodihydrofluorescein diacetate, Beyotime Biotechnology, China) under the fluorescence microscope. In addition, ROS were also detected by the FITC channel for fluorescence intensity using flow cytometry (BD Biosciences, FACSuite™, USA).

**MDA and SOD assays**. To assess the oxidative stress level in HK-2, ES-2 cells, kidney, and tumor tissues, the levels of SOD and MDA were tested with commercially available kits (Beyotime Biotechnology, China). Briefly, HK-2, ES-2 cells, kidney, and tumor tissues were homogenized in ice-cold 0.1 M phosphate buffer (pH 7.4), then the homogenates were filtered and centrifuged using a refrigerated centrifuge at 4 °C (20 min, $16,000 \times g$). The obtained supernatants were used to determine the SOD enzyme activity and the lipid peroxidation level by measuring MDA content. The SOD enzyme activity was expressed as a unit of activity per milligram of protein and the MDA content was expressed as micromole per gram of protein.

***Nrf2* knockdown by small interfering RNA (siRNA)**. To silence the gene expression of *Nrf2*, *Nrf2*-specific siRNA (Si*Nrf2*) and control siRNA (SiNC) were obtained from Genepharma (Shanghai, China). HK-2 cells were seeded into 6-well plates ($2 \times 10^5$ cells per well) overnight and then transfected with siRNA (Si*Nrf2* or SiNC) mixed with Lipofectamine 2000 transfection reagent. The medium (Opti-MEM) was replaced at 6 h post-transfection. After 24 h, the cells were treated with DDP and CNPs. Si*Nrf2* sequences were enlisted in Table S1.

**Subcutaneous xenograft model of ovarian cancer**. BALB/c nude mice (Female, 6-8 weeks) were purchased from Shanghai SLAC Laboratory Animal Co., Ltd. The evaluation of anti-tumor activity in vivo was performed using ES-2 cells, a kind of human ovarian cancer cell line. $5 \times 10^6 - 1 \times 10^7$ ES-2 cells were inoculated subcutaneously into the left flank of BALB/c nude mice. After stabilization of tumor traits (after 2–3 generations), the tumor blocks of the mice were dissected and placed in a glass dish containing sterile saline. The blood vessels on the surface of the tumor were peeled off. Then the bulk tumor was minced into pieces of approximately $1 \times 1 \times 1$ mm$^3$, which were transplanted into the left skin of BALB/c nude mice with a trocar. The whole process was carried out within 1 h under sterile conditions.

When the tumors grew to an average volume of 60–70 mm$^3$ at 10 days after tumor transplantation, the tumor-bearing mice were randomly divided into ten groups (6 mice per group) for further study. Six of ten groups of mice were fed with sterile water, and treated with saline, DDP, CNPs, NAC, DDP plus CNPs, or DDP plus NAC. The remaining four groups of mice were fed with 200 mM soda water, and treated with saline, DDP, CNPs, or CNPs plus DDP. DDP was intraperitoneally injected at the dosage of 15 mg/kg and CNPs were intravenously injected at the dosage of 1.5 mg/kg twice a week. NAC was given by gavage at the dosage of 200 mg/kg. The body weight of mice and the tumor volume were measured every other day. The tumor volumes were calculated as V = L/2 × W$^2$ (V, volume; L, length of the tumor; W, the width of tumor).

**Statistical analysis**. All of the data were presented as the mean ± SEM. from a minimum of three independent experiments. Data analysis was performed using Microsoft Excel version 16.16.23, Image J (version 1.8.0), and GraphPad Prism Software Version 8.0 (GraphPad Prism, San Diego, California, USA). Statistical parameters including statistical analysis, statistical significance, and n value were indicated in the figure legends. For statistical comparison, we performed two-tailed Student's t-test, one-way ANOVA Tukey's post-hoc analysis for multiple comparisons and log-rank (Mantel-Cox) test. A value of $p < 0.05$ was considered significant (represented as $*p < 0.05$, $**p < 0.01$, $***p < 0.001$, $^\#p < 0.05$, $^{\#\#}p < 0.01$, $^{\#\#\#}p < 0.001$ or not significant (n.s.)).

**Reporting summary**. Further information on research design is available in the Nature Research Reporting Summary linked to this article.

## Data availability
The main data supporting the findings of this study are available within this Article, it's Supplementary Information and Source Data. Extra data are available from the corresponding author upon reasonable request. Source data are provided with this paper.

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

## Acknowledgements

We acknowledge financial support by National Key Research and Development Program of China (2016YFA0203600), the National Natural Science Foundation of China (31822019, 32071374, 91859116, 81872878, 81761148029), the One Belt and One Road International Cooperation Project from Key Research and Development Program of Zhejiang Province (2019C04024), the Zhejiang Provincial Natural Science Foundation of China (LGF19C100002, LGF19H310002, LR21H310001), the Fundamental Research Funds for the Central Universities (2019XZZX004-15, 2020FZZX001-05), the CAS Interdisciplinary Innovation Team (JCTD-2020-08).

## Author contributions

D.L. and Q.W. conceived and designed the study. H.S., F.L., and C.F. performed the experiments with assistance from A.X., J.W., F.X., and J.R.; H.S. and F.L. synthesized and characterized the materials; C.F., H.S. performed the cell and animal experiments. F.L., H.L., X.G., A.X., J.W., and B.Y. contributed to the discussion. D.L., Q.W., H.S., C.F., F.X., H.L., and J.L. analyzed the data and wrote the manuscript. D.L., Q.W., and F.L. provided project supervision. All the authors discussed the results and approved the final version of the manuscript.

## Competing interests

The authors declare no competing interests.
