## [Peer Review File · Nature Communications]

Reviewers' Comments:

Reviewer #1:

Remarks to the Author:

This manuscript reported an enzymatic activity tunable ceria nanoparticles (CNPs) that exhibits context-dependent ROS modulation capacity to protect against chemotherapy-induced AKI without interference with the potency of chemotherapeutic agents. It is very interesting to exploit pH-dependent ROS scavenging ability of CNPs to overcome the difficulty in reconciliation of the ROS depletion and production between AKI and cancer therapy. The manuscript is well written with solid data. I believe this study is highly important for the translational nanomedicine in onco-nephrology, which I recommend to publish after the author addressing the following comments.

1. Previous studies reported that ceria nanoparticles exhibited multi-enzymatic activity including CTA-mimic activity and SOD-mimic activity. In this article, the authors confirmed the CAT-mimic activity is pH-dependent, but how the SOD-mimic activity of CNPs exhibit under different pH conditions?
2. The kinetic curves of O₂ generation from the decomposition of H₂O₂ at different concentrations in the presence of CNPs should be presented for fully verification of CAT-mimic activity of CNPs under different pH conditions.
3. The authors modified DSPE-PEG2K on the surface of CNPs, but there is no direct evidence to prove the surface modification. The colloid stability of the CNPs should be confirmed by using DLS and zeta potential.
4. The authors demonstrated that CNPs can scavenge ROS in DDP-treated HK-2 cells to restore the cell viability, but the level of ROS in ES-2 cells under acidic conditions after CNPs treatments were not presented, and the relationship between ROS and tumor cell death should be further explained in vitro.
5. CNPs showed different effects on the potency of chemotherapeutic agents in tumor sites with different pH values. The ROS-related results in vivo should be presented to explain this difference, for example, the SOD activity and the MDA level in tumor tissues.
6. It has been clear in this study that CNPs protect HK-2 at normal pH and does not promote or inhibit ES-2 proliferation at acidic pH. Have they overexpressed the Nrf2 gene on acidic tumor cells (ES-2 cells) in vitro and observe the effect on ES-2 tumor cells? Have they observed the effect of Nrf2 gene on acidic ES-2 tumor cells?

Reviewer #2:

Remarks to the Author:

Weng and colleagues report on the use of tunable ceria nanoparticles as a therapeutic strategy in cisplatin-induced acute kidney injury without interfering with the chemotherapeutic effects of reactive oxygen species on cancer cells. The authors synthesized and characterized novel ceria nanoparticles (CNP) that exhibit antioxidant properties under different pH conditions - specifically they were capable of decomposing hydrogen peroxide in neutral environment but inert in acidic conditions. They show that CNPs were able to protect both in vitro and in vivo in cisplatin-induced nephrotoxicity through a Nrf2-dependent mechanism. Furthermore, using tumor bearing mice, they show that CNPs did not affect the anti-tumor effects of cisplatin unless the intratumoral pH was artificially raised.

Comments:

The studies in this paper represent novel and important observations that have potential clinical relevance. The authors should address the following comments to further strengthen their work.

1. The animal numbers for the in vivo studies is rather limited (n=6) and only male mice were used. The sample size should be increased to at least 12-15 mice/group. Are similar findings observed in female mice?

2. Most models of acute kidney injury (AKI) involve the generation of reactive oxygen species. Are the findings from this work applicable to other models such as ischemia-reperfusion induced AKI?
3. Prior studies have shown that cisplatin induces heme oxygenase-1 (HO-1) mRNA and protein in the kidney (e.g. PMID: 8569092). In Figure 4g, the authors show that cisplatin reduces HO-1 expression in the renal cortex. How the authors explain this discrepancy?

Reviewer #3:

Remarks to the Author:

Summary: The authors have reported about the ceria nanoparticles (CNPs) that possess the variable catalytic activity depending on the environmental pH. These CNP particles can be used as a nanomedicine to recover the acute kidney injury which incurred during chemotherapy. CNPs act as small molecular antioxidants and ROS scavenging at the neutral pH but acidic pH CNPs become inert due to the excessive H⁺ that disrupts the re-exposure of active catalytic sites, allowing a buildup of chemotherapy-mediated ROS generation to kill cancer cells. The authors have performed both the in-vitro and in vivo studies as a proof of concept. The authors reported that CNPs have prevented the chemotherapeutic agents induced apoptosis of renal cells in AKI by detoxing H₂O₂, as well as activating the Nrf2/ Keap1 signaling pathway to regulate ROS-related genes for the restoration of redox homeostasis in the renal cells.

Comments:

1. An enzyme is a word usually defines a biomolecule that performs a catalytic activity. I strongly suggest the authors to change the word "enzymatic activity tunable" to "catalytic activity tunable" CNPs.
2. The authors have determined the size of the CNPs in chloroform, CNPs after DSPE-PEGylation in water via TEM and DLS analysis. Please Include the size of CNPs obtained in presence of H₂O₂ at pH 7.4 and 6.0, it will give a clear picture if there is any change in the particle.
3. Page no. 74, authors mentioned that the DSPE-PEGylation CNPs was used to impart the colloidal stability of the CNPs in aqueous solution. Describe, why you didn't use the DSPE-PEGylation CNPs particles for the current study. Besides, to measure the free radical scavenging activity of CNPs and DSPE-PEGylation CNPs using SOD assay and make a comparative statement of CNPs and DSPE-PEGylation CNPs that prevent chemotherapy-induced AKI and cancer.
4. For assessing the cell viability what solvent was used as a vehicle for the study, clearly specify in the materials and method.
5. ES-2 cells are capable of forming tumor why the effect of CNPs over the cell viability (Fig 2 a and b and supplementary data) shows a similar pattern of Hk-2 cells when medium pH is being changed? Like at pH 6.0 the cell viability is low and at pH 7.4 the cell viability increases as molar volume of CNP increases, explain Why?
6. Why did author choose the ES-2 cells and Hk-2 cells? Did you see a similar response using another cell type? because, normal cells are around the tumor site.
7. Whether authors performed the western blot for the expression of protein in the presence of CNPs and without DDP. If yes include the result in the figure 4 (a) if not please justify why it was not done.
8. For studying the gene expression authors have not clearly explained the design of experiment in the materials and method.
9. Figure 2: Requesting the author to change the color of the asterisk symbol in the image 2 (g) for the better visibility.
10. Figure 5: in the image d the scale bars for the days are not clearly visible but as far as it seems like scales has started from day 0-12. In the case of vehicle, NAC, DDP+CNP and CNP from the graph in the image b the tumor volume seems to be increasing but in the image d it looks vice versa as of the tumor size being reduced from day 0-12 can authors explain it, Why? Similar kind of observance is even found between the images of e and g.
11. Line 477: authors please write the full form for ICP-MS.
12. Line 495: represent the unit for temperature 37?

13. Line 510: the unit of time measured was represented as "min"
14. Line 551: represent seconds in abbreviated form
15. Line 594: specify the units in abbreviated form for the minutes, authors please follow the same form of unit expression through the paper.
16. Line 619: samples were centrifuged at what speed (4)? Can you please specify it?
17. The author needs to cite suitable references that CNPs already have phase transformation in aqueous media, high antioxidative properties, wound healing effect and anti-inflammatory properties.
Journal of Materials Research 34.3 (2019): 465-473.
Small. 2009 Dec;5(24):2848-56.,
Chemical Society Reviews 39 (11), 4422-4432
Biomaterials 33 (31), 7746-7755
Artificial cells, nanomedicine, and biotechnology 46.sup3 (2018): S956-S963.

Reviewer #4:

Remarks to the Author:

This manuscript described development of a nanoparticle that actively catalyzes ROS in non-acidic environments. This is a clever approach to mitigate some chemotherapy induced nephrotoxicity, or acute kidney disease (AKD), which can be a serious problem.

MAJOR CONCERNS

1. Experiments appear to be performed in one of two cell lines (HK-2 or ES-2) in different lines of investigation and there is little or no justification. Further, the use of only one oxidizing chemotherapeutic is a significant weakness. These two weaknesses argue against the generality of this phenomenon.
2. The experiment shown in figure 5j appears to indicate that cisplatin 3 mg/kg q2d can kill non tumor-bearing nude mice within 10 days and the CNPs can protect against this (as can NAC). However, the data in figures 5b and 5e show the anti-tumor effect of 3 mg/kg q3d cisplatin, without any noted killing of mice. If these mice were sick from cisplatin toxicity that could explain the lack of tumor growth.
3. Figure 2a and b are confusing. It appears that HK-2 cells were treated at pH 7.4 and ES-2 cells at 6.0, and then the differences ascribed to the differences in pH? This cannot be so. At the very least this experiment has to be performed in both cell lines (and at more physiological pH values).
4. Characterization of activity at pH 6.0 is probably not relevant to tumor pH, which is typically cited as being in the range of 6.6-6.7. How do these nparticles behave at 6.6?

MORE MINOR CONCERNS

5. The protective effects of CNPs against cisplatin nephrotoxicity in figures 2 and 3d, and 3e are impressive. However, controls using catalase-deficient CNPs must be included.
6. Fig 1 seems to indicate that these nparticles self-adhere in aqueous solution. Were there attempts to keep these disperse? While 3-7 nm is great for tissue penetration, this may be hampered if these exist as clusters.
7. The role of ROS in the etiology of chemotherapy induced AKD is far from clear and could be better developed. In the landmark NEJM review by Rosner and Perazella (ref 3), ROS were not mentioned once. In ref 9, NAC pretreatment mitigated the rise in IL-6 in Nfr2 -/- mice injured with ischemia. What does this have to do with chemotherapy induced toxicity? Hence, this may be a solution in search of a problem. Further, if NAC can prevent renal toxicity, why is it not being used in the clinic, as it is approved?

**Reviewer #1 (Remarks to the Author):**

This manuscript reported an enzymatic activity tunable ceria nanoparticles (CNPs) that exhibits
context-dependent ROS modulation capacity to protect against chemotherapy-induced AKI
without interference with the potency of chemotherapeutic agents. It is very interesting to
exploit pH-dependent ROS scavenging ability of CNPs to overcome the difficulty in
reconciliation of the ROS depletion and production between AKI and cancer therapy. The
manuscript is well written with solid data. I believe this study is highly important for the
translational nanomedicine in onco-nephrology, which I recommend to publish after the author
addressing the following comments.

**Response:** *Thank you very much for your encouraging comments. Based on your suggestions,*
*we have made point-to-point responses. The manuscript was revised to incorporate additional*
*data, and the revised sentences are marked with yellow background. We believe that your*
*comments have significantly improved the quality of our manuscript.*

1. Previous studies reported that ceria nanoparticles exhibited multi-enzymatic activity
including CTA-mimic activity and SOD-mimic activity. In this article, the authors confirmed
the CAT-mimic activity is pH-dependent, but how the SOD-mimic activity of CNPs exhibit
under different pH conditions?

**Response:** *Thank you for your valuable comments. Per your suggestions, the SOD-like activity*
*of CNPs was detected under different pH conditions. The results revealed that CNPs possess*
*SOD-like activity at both pH 7.4 and pH 6.6 but with much better performance at pH 6.6*
*(Supplementary Fig. 4), which can catalyze the dismutation of superoxide radicals ($\cdot O_2^-$) into*
*H_2O_2 .*

**Our modification to the manuscript:** *The results were added as Supplementary Fig. 4 in the*
*revised supporting information. In addition, the following sentences and methods were added*
*on page 4, 11 and 30 in the revised manuscript, respectively.*

- • Supplementary Fig. 4

**Supplementary Figure 4. a,b, SOD-like activity of CNPs at pH 7.4 (a) and pH 6.6 (b). n=3,**
**data represents as means ±S.D.**

• Page 4
“CNP s demonstrated superoxide dismutase (SOD)-like activity under neutral and acidic
condition (Supplementary Fig. 4), which catalyzes the dismutation of superoxide radicals ($\cdot\text{O}_2^-$)
into H_2O_2 ³¹.”

**References:**

31. Ward, M. B., et al. Superoxide dismutase activity enabled by a redox-active ligand rather
than metal. *Nat. Chem.* **10**, 1207–1212 (2018).

• Page 11

“CNP s exhibit SOD-like activity under both neutral and acidic conditions to catalyze the
dismutation of superoxide radicals ($\cdot\text{O}_2^-$) into H_2O_2 , which is another kind of toxic ROS.
Notably, CNP s are highly active for decomposing H_2O_2 under neutral condition while become
inert under acidic condition.”

• Page 30

**Superoxide dismutase (SOD)-like activity of CNP s.**

The SOD-like activity of CNP s at different concentrations (50 μM , 100 μM , 150 μM and 400
μM) was measured by a Total Superoxide Dismutase Assay Kit with WST-8 (Beyotime, China)
under different pH conditions.

2. The kinetic curves of O_2 generation from the decomposition of H_2O_2 at different
concentrations in the presence of CNP s should be presented for fully verification of CAT-mimic
activity of CNP s under different pH conditions.

**Response:** *Thank you for your kind suggestion. Per your suggestion, we analyzed the amount*
*of O_2 generated from different concentrations of H_2O_2 after the CNP s addition under different*
*pH conditions. The results revealed that the Michaelis–Menten constant (K_M) and maximum*
*reaction rate (V_{\max}) of as-prepared CNP s with H_2O_2 as the substrate is 0.05894 M and 0.44*
*mg $\text{L}^{-1} \text{min}^{-1}$ at pH 7.4, and 0.07901 M and 0.21 mg $\text{L}^{-1} \text{min}^{-1}$ at pH 6.6, respectively*
*(Supplementary Table 1).*

**Our modification to the manuscript:** *The results were added as Supplementary Table 1 in*
*the revised supporting information. In addition, the following sentences and methods were*
*added on page 4 and 30 in the revised manuscript, respectively.*

• Supplementary Table 1

Catalyst	pH	[E] (mg mL^{-1})	Substrate	K_M (M)	V_{\max} ($\text{mg L}^{-1} \text{min}^{-1}$)
CNP s	7.4	3.6×10^{-2}	H_2O_2	0.05894	0.44
CNP s	6.6	3.6×10^{-2}	H_2O_2	0.07901	0.21

**Table S1.** The Michaelis–Menten constant (K_M) and maximum reaction rate (V_{\max}) of as-
prepared CNP s with H_2O_2 as the substrate for CAT-like catalysis under different pH conditions.

• Page 4

“Moreover, the CNP s-induced H_2O_2 decomposition follows Michaelis–Menten kinetics. As

shown in Supplementary Table 1, the K_M of CNPs to H_2O_2 under neutral condition is 1.3 times
lower than that of CNPs under acidic condition, while the V_{max} of CNPs under neutral condition
shows 2.1-folds increase relative to CNPs under acidic condition.”

• Page 30

**Catalase-like activity of CNPs.**

... The kinetic assays of CNPs with H_2O_2 substrate were performed by adding different
amounts (25, 100, 200 μ L) of 30% H_2O_2 solution into buffer solution (0.1 M PBS buffer, pH
7.4 or pH 6.6), and the final total volume of solution is 8.0 mL. The Michaelis–Menten constant
was calculated according to the Michaelis–Menten saturation curve by GraphPad Prism 5.0
(GraphPad Software).

3. The authors modified DSPE-PEG_{2K} on the surface of CNPs, but there is no direct evidence
to prove the surface modification. The colloid stability of the CNPs should be confirmed by
using DLS and zeta potential.

**Response:** Thank you very much for your valuable comments. To verify the modification of
DSPE-PEG_{2K}, thermogravimetric analysis (TGA) and Fourier transform infrared (FT-IR)
spectra analysis were performed. The significant weight loss in TGA curves indicates that
DSPE-PEG_{2K} molecules are successfully modified onto the surface of CNPs <Ref. Nano Lett.
2018, 18, 1196-1204>. Moreover, the attachment of DSPE-PEG_{2K} to CNPs is further verified
according to the peaks at 1116 cm^{-1} (C–O–C stretching) in the FT-IR spectra <Ref. Int. J.
Nanomedicine 2018, 13, 3965-3973>.

Per your suggestion, the size and zeta potential of CNPs were detected using a Zetasizer
Nano ZS90. No obvious size or zeta potential change was observed over a week, which
demonstrates their high colloidal stability.

**Our modification to the manuscript:** The results were added as Supplementary Figs. 1 and
2 in the revised supporting information. In addition, the following sentences and methods were
added on page 4 and 29 in the revised manuscript.

• Supplementary Fig. 1

**Supplementary Figure 1. a,b, TGA curves (a) and Fourier transform infrared (FT-IR) spectra**
**(b) of hydrophobic CNPs and PEGylated CNPs.**

• Supplementary Fig. 2

**Supplementary Figure 2. a,b, The sizes (a) and zeta potentials (b) of CNPs over a week.**

• Page 4

“The hydrophobic CNPs with a diameter of ~ 3 nm (Fig. 1b) were synthesized via a modified
reverse micelle method²³, and 1,2-distearoyl-sn-glycero-3-phosphoethanolamine-N-[methoxy
(polyethylene glycol)-2,000] (DSPE-PEG_{2K}) was successfully surface modified
(Supplementary Fig. 1), imparting the colloidal stability of the CNPs in aqueous solution (with
a hydrodynamic diameter of ~ 9.7 nm) for long blood circulation²⁴⁻²⁶ (Fig. 1c and
Supplementary Fig. 2).”

• Page 29

**Characterization.**

“... The thermogravimetric analysis (TGA) of DSPE-PEG_{2K} modified or unmodified CNPs
was analyzed by a METTLER TOLEDO TGA/DSC1 thermogravimetric analyzer system. The
Fourier transform infrared (FT-IR) spectra of DSPE-PEG_{2K} modified or unmodified CNPs was
analyzed by NicoLET iS50FT-IR. The hydrodynamic size and zeta potentials of DSPE-PEG_{2K}
modified CNPs were detected using a Zetasizer Nano ZS90 (Malvern Instruments,
Worcestershire, UK).”

4. The authors demonstrated that CNPs can scavenge ROS in DDP-treated HK-2 cells to restore
the cell viability, but the level of ROS in ES-2 cells under acidic conditions after CNPs
treatments were not presented, and the relationship between ROS and tumor cell death should
be further explained *in vitro*.

**Response:** Thank you very much for your valuable comments. ROS production is known to
accompany the inhibition of SOD activity <Ref. Free Radical Bol. Med. 2017, 108, 418-432>.
Consequently, we analyzed the SOD activity to investigate the ROS generation in ES-2 cells
after different treatments. The results demonstrated that, in accordance with the low cell
viability, DDP treatment inhibits SOD activity, indicating the generation of ROS in ES-2 cells
(Supplementary Fig. 22). Interestingly, the SOD activity can be nearly recovered to basal level
after the co-treatment with NAC, but not CNPs since their anti-oxidant activity is switched off
under acidic condition (Fig. 1d). These results show that CNPs have no effect on the ROS

induced by DDP under acidic conditions, so that the viability of ES-2 cells could not be
restored.

**Our modification to the manuscript:** The results were added as Supplementary Fig. 22 in the
revised supporting information. In addition, the following sentences and methods were added
on page 10 and 37 in the revised manuscript.

• Supplementary Fig. 22

**Supplementary Figure 22.** SOD activity in ES-2 cells treated with DDP, NAC, CNPs,
DDP+NAC and DDP+CNPs under acidic condition.

• Page 10

“In accordance with *in vitro* results (Supplementary Figs. 9 and 22), the *in vivo* anti-tumour
efficiency of DDP is found to be suppressed by NAC due to the quenched tumoricidal ROS, as
confirmed by the enhanced SOD activity and reduced MDA level in the tumour tissues
(Supplementary Fig. 23a,b).”

• Page 37

**MDA and SOD assays.**

To assess the oxidative stress level in HK-2, ES-2 cells, kidney and tumour tissues, the levels
of superoxide dismutase (SOD) and malondialdehyde (MDA) were tested with commercially
available kits according to the manufacturer’s instructions (Beyotime Biotechnology, China).
Briefly, HK-2, ES-2 cells, kidney and tumour tissues were homogenized in ice-cold 0.1 M
phosphate buffer (pH 7.4), then the homogenates were filtered and centrifuged using a
refrigerated centrifuge at 4 °C (20 min, 13,000 r.p.m.). The obtained supernatants were used to
determine the SOD enzyme activity and the lipid peroxidation level by measuring MDA
content. The SOD enzyme activity was expressed as a unit of activity per milligram of protein
and the MDA content was expressed as micromole per gram of protein.

5. CNPs showed different effects on the potency of chemotherapeutic agents in tumor sites with
different pH values. The ROS-related results *in vivo* should be presented to explained this
difference, for example, the SOD activity and the MDA level in tumor tissues.

**Response:** Thank you for your kind suggestion. Per your suggestion, we analyzed the SOD
activity and MDA level to study the ROS production in tumour tissue from mice after different
treatments. As shown in Supplementary Fig. 23a,b, the SOD activity was suppressed and MDA

level was increased in tumour tissues of DDP-treated mice, indicating the excessive generation
 of ROS. Notably, CNPs showed no interference with SOD activity and MDA level in tumour
 tissues of DDP-treated mice. In accordance with their inhibited ROS scavenging ability under
 acidic condition *in vitro* (Fig. 2b), these results indicate that CNPs show no effects on the DDP-
 induced ROS in acidic tumour microenvironment, thus exhibiting no interference with the anti-
 tumour efficacy of DDP. Interestingly, CNPs showed similar effects to NAC on the efficacy of
 DDP once the pH of tumour microenvironment was artificially increased because the ROS
 scavenging effect of CNPs is high under neutral condition (Supplementary Fig. 25a,b).

 **Our modification to the manuscript:** The results were added as Supplementary Figs. 23 and
 25 in the revised supporting information. In addition, the following sentences and methods
 were added on page 10 and 37 in the revised manuscript.

- • Supplementary Fig. 23

 **Supplementary Figure 23.** The SOD activity (a) and MDA level (b) of tumour tissues from
 BALB/c mice fed with sterile water after different treatments.

 • Supplementary Fig. 25

 **Supplementary Figure 25.** The SOD activity (a) and MDA level (b) of tumour tissues from
 BALB/c mice fed with soda water after different treatments.

• Page 10

“In accordance with *in vitro* results (Supplementary Figs. 9 and 22), the *in vivo* anti-tumour
efficiency of DDP is found to be suppressed by NAC due to the quenched tumoricidal ROS, as
confirmed by the enhanced SOD activity and reduced MDA level in the tumour tissues
(Supplementary Fig. 23a,b). However, it is not affected by CNPs (Fig. 6b,c,d and
Supplementary Fig. 24a,b) whose ROS scavenging activities are turned off once exposed to the
acidic tumour microenvironment, showing no obvious change in SOD activity and MDA level
(Supplementary Fig. 23a,b). Intriguingly, CNPs can reverse the imbalance of ROS-related
elements induced by DDP (Supplementary Fig. 25a,b), and thus weakened the anti-tumour
efficiency of DDP when the intratumoral pH is artificially raised⁵⁸ (Fig. 6e,f,g).”

• Page 37

**MDA and SOD assays.**

To assess the oxidative stress level in HK-2, ES-2 cells, kidney and tumour tissues, the levels
of superoxide dismutase (SOD) and malondialdehyde (MDA) were tested with commercially
available kits according to the manufacturer’s instructions (Beyotime Biotechnology, China).
Briefly, HK-2, ES-2 cells, kidney and tumour tissues were homogenized in ice-cold 0.1 M
phosphate buffer (pH 7.4), then the homogenates were filtered and centrifuged using a
refrigerated centrifuge at 4 °C (20 min, 13,000 rpm.). The obtained supernatants were used to
determine the SOD enzyme activity and the lipid peroxidation level by measuring MDA
content. The SOD enzyme activity was expressed as a unit of activity per milligram of protein
and the MDA content was expressed as micromole per gram of protein.

6. It has been clear in this study that CNPs protect HK-2 at normal pH and does not promote
or inhibit ES-2 proliferation at acidic pH. Have they overexpressed the *Nrf2* gene on acidic
tumor cells (ES-2 cells) in vitro and observe the effect on ES-2 tumor cells? Have they observed
the effect of *Nrf2* gene on acidic ES-2 tumor cells?

**Response:** *Thank you for your kind suggestion. According to your advice, the Nrf2 was*
*overexpressed in ES-2 cells (Supplementary Fig. 30a, raw full-length gels and blots are*
*attached below), and the cell viability was examined after different treatments under acidic*
*condition to investigate the role of overexpressed Nrf2 under the effects of CNPs on cell*
*proliferation. The result showed that overexpression of Nrf2 genes has protective effects on the*
*viability of ES-2 cells under acidic conditions (Supplementary Fig. 30b).*

• Supplementary Fig. 30

 **Supplementary Figure 30. a**, Western blot analysis of Nrf2 protein expression in ES-2 cells.
 **b**, The viability of ES-2 cells with or without Nrf2 overexpression after different treatments
 under acidic condition.

**Reviewer #2 (Remarks to the Author):**

Weng and colleagues report on the use of tunable ceria nanoparticles as a therapeutic strategy
in cisplatin-induced acute kidney injury without interfering with the chemotherapeutic effects
of reactive oxygen species on cancer cells. The authors synthesized and characterized novel
ceria nanoparticles (CNP) that exhibit antioxidant properties under different pH conditions -
specifically they were capable of decomposing hydrogen peroxide in neutral environment but
inert in acidic conditions. They show that CNPs were able to protect both in vitro and in vivo
in cisplatin-induced nephrotoxicity through a Nrf2-dependent mechanism. Furthermore, using
tumor bearing mice, they show that CNPs did not affect the anti-tumor effects of cisplatin
unless the intratumoral pH was artificially raised.

**Response:** *Thank you very much for your comments. Based on your kind suggestions, we*
*have made point-to-point responses. The manuscript was revised to incorporate additional*
*data, and the revised sentences are marked with yellow background.*

1. The animal numbers for the *in vivo* studies is rather limited (n=6) and only male mice were
used. The sample size should be increased to at least 12-15 mice/group. Are similar findings
observed in female mice?

**Response:** *Thank you for your comments. Per your suggestion, we have increased the number*
*of ICR mice to 16 for each group in in vivo studies, including 8 male mice and 8 female mice.*
*All the original results are well reproduced with an increased sample size (Fig.3b-f). Notably,*
*similar findings were observed in female mice as that in male mice.*

**Our modification to the manuscript:** *The results were added as Fig. 3 in the revised*
*manuscript. In addition, the following sentences and methods were added on page 7, 32 and*
*33 in the revised manuscript.*

**Fig. 3 CNPs protect against chemotherapy-induced AKI *in vivo*.** **a**, Schematic
 representation of treatment schedule and therapy assessments for AKI mice. **n=16, including 8**
 **male mice and 8 female mice in each group.** **b, c**, Serum BUN (**b**) and Cre (**c**) levels of mice
 **treated with vehicle, DDP, DDP and CNPs or iCNPs (inactive CNPs), respectively (n =**
 **16/group).** **d**, Relative mRNA expression of *KIM-1* in the renal cortex from each group. (**n =**
 **16/group).** **e**, The tubular injury score was calculated according to the percentage of damaged
 tubules as reported: 0, no damage; 1, <25% damage; 2, 25-50% damage; 3, 50-75% damage;

4, >75% damage. A pathologist evaluated 5 randomly selected fields per section of the mouse
kidneys at a magnification of $\times 400$ in a blind manner. Data are presented as means \pm standard
error of mean (SEM) ($n = 16/\text{group}$). **f**, Representative H&E sections of the kidney from **male**
**and female mice in** each group ($n = 16/\text{group}$). Arrows indicate tubules with necrosis, epithelial
anoikis cavitation, or loss of brush border. **Triangles denote the formation of casts in tubes.**
Scale bar: 100 μm . *** $p < 0.001$ vs. vehicle group and ### $p < 0.001$ vs. DDP group.

• Page 7

“As shown in Fig. **3a**, ICR mice (**8 male mice and 8 female mice**) were intraperitoneally
injected (*i.p.*) with a single dose of DPP (15 mg/kg) to induce AKI, accompanied by intravenous
injection (*i.v.*) of saline, CNPs (0.5 or 1.5 mg/kg) **or iCNPs** (0.5 or 1.5 mg/kg), respectively.”

“Also, the serum blood urea nitrogen (BUN) and creatinine (Cre), the classic indicators of
clinical renal dysfunction³⁹, were drastically increased in mice treated with DDP alone, but
prominently decreased in **both CNPs co-treated male and female mice (Fig. 3b,c).**”

“**In contrast, the iCNPs show no renal protection effects in DDP-treated mice.**”

• Page 32

**AKI model establishment.**

“All animal experiments were approved by the Institutional Animal Care and Use Committee
(IACUC) of Zhejiang University. ICR mice (Male **and female**, 6-8 weeks) were obtained from
Beijing Vital River Laboratory Animal Technology Co., Ltd. All mice were housed in a specific
pathogen-free environment at $21 \pm 1^\circ\text{C}$ and $60 \pm 5\%$ humidity, with a 12 h light-dark cycle. **(1)**
To establish **DDP-induced** acute kidney injury model, **ICR** mice received a single
intraperitoneal injection of cisplatin at 15 mg/kg. ...”

• Page 33

**Treatment of AKI mice.**

“(1) The DDP-induced AKI mice were intravenously injected with different dosages of CNPs
(0.5 or 1.5 mg/kg) **or iCNPs (0.5 or 1.5 mg/kg).** Control group animals were intraperitoneally
injected with an equal volume of saline. All mice were sacrificed at 72 h after cisplatin
injection, and their blood and kidneys were collected for evaluation of renal function and
tissue damage. Furthermore, organs such as the heart, liver, spleen, lungs and kidneys were
also harvested.”

2. Most models of acute kidney injury (AKI) involve the generation of reactive oxygen species.
Are the findings from this work applicable to other models such as ischemia-reperfusion
induced AKI?

**Response:** *Thank you for the valuable comment. To address whether our findings are*
*applicable to other models, we evaluated the effect of CNPs on ischemia reperfusion (IR)-*
*induced AKI, and find that, in this scenario, CNPs are also effective in ameliorating the*
*symptoms (Supplementary Fig. 28a-d), suggesting the generality of our findings.*

**Our modification to the manuscript:** *The results were added as Supplementary Fig. 28 in the*

revised supporting information. In addition, the following sentences and methods were added
on page 12, 32 and 33 in the revised manuscript.

• Supplementary Fig. 28

Supplementary Figure 28. a,b, Serum BUN (a) and Cre (b) levels in each group (n = 5/group).

**c**, The tubular injury score was calculated according to the percentage of damaged tubules as
reported: 0, no damage; 1, <25% damage; 2, 25-50% damage; 3, 50-75% damage; 4, >75%
damage. A pathologist evaluated 5 randomly selected fields per section of the mouse kidneys
at a magnification of $\times 400$ in a blind manner. Data are presented as means \pm standard error of
mean (SEM). **d**, Representative H&E sections of the kidney from each group (n = 5/group).
Arrows indicate tubules with necrosis, epithelial anoikis cavitation, or loss of brush border.
Triangles denote the formation of casts in tubes. The lower panel are the magnified regions
from the upper panel. Scale bar: 200 μm (up) or 100 μm (down), n = 5; ***p < 0.001 vs. vehicle
group and ###p < 0.001 vs. DDP group.

• Page 12

“After systemic administration, the CNPs possess antioxidant activity for kidney protection in
chemotherapeutic agents-induced AKI (Fig. 3). Moreover, the protective effect of CNPs is also
observed in ischemia-reperfusion (IR)-induced AKI, which involves excessive ROS generation
(Supplementary Fig. 28).”

• Page 32-33

**AKI model establishment.**

“... (3) To establish IR-induced AKI model, ICR mice were anesthetized with an
intraperitoneal injection of 1% pentobarbital sodium and were subjected to bilateral renal
artery occlusion with non-traumatic vascular clips for 40 min before reperfusion. In the sham
group, mice underwent the same operation without renal artery clamping.”

• Page 33

**Treatment of AKI mice.**

“... (3) the ICR mice were intravenously injected with CNPs (1.5 mg/kg) or saline 3 h prior to
the establishment of the IR-induced AKI model. The blood and kidneys of mice were collected
and evaluated for kidney function at 24 h after IR.”

3. Prior studies have shown that cisplatin induces heme oxygenase-1 (HO-1) mRNA and
protein in in the kidney (e.g. PMID: 8569092). In Figure 4g, the authors show that cisplatin
reduces HO-1 expression in the renal cortex. How the authors explain this discrepancy?

**Response:** *Thank you for your valuable comment. In the article (Ref. Kidney Int. 1995, 48,*
*1298-1307, PMID: 8569092), the level of heme oxygenase-1 (HO-1) mRNA and protein were*
*increased at 6 h and 12 h after DDP treatment; while our results show that the level of HO-1*
*protein and mRNA were reduced at 72 h after DDP treatment. To explain this discrepancy, we*
*examined the expressions of HO-1 protein and mRNA in renal cortex of mice at different time*
*points after DDP treatment (6 mg/kg or 15 mg/kg).*

*As shown in the Supplementary Fig. 18a-d, the expression of HO-1 protein and mRNA were*
*increased at 6 h and 12 h after DDP treatment, which is consistent with the previous report*
*<Ref. Kidney Int. 1995, 48, 1298-1307, PMID: 8569092>; however, it begins to decrease at*
*24 h after DDP treatment (15 mg/kg), and becomes even lower than basal level at 72 h post-*
*injection, probably due to the persistently high level of ROS (Supplementary Fig. 11) and*

accumulated oxidative damage that eventually decrease the production antioxidant proteins
 including HO-1 <Ref. Am. J. Med. Sci. 2007, 334, 115-124; Ren. Fail. 2018, 40, 371-378>.

 **Our modification to the manuscript:** The results were added as Supplementary Fig. 18 in the
 revised supporting information. In addition, the following sentence was added on page 8-9 in
 the revised manuscript.

• Supplementary Fig. 18

 **Supplementary Figure 18. a,b**, The expression of HO-1 in renal cortex of mice after treatment
 with 6 mg/kg (a) and 15 mg/kg (b) DDP. **c,d**, Relative mRNA expression of HO-1 in renal
 cortex of mice after treatment with 6 mg/kg DDP (c) and 15 mg/kg (d). n=3. Data are presented
 as mean ± SEM; *p < 0.05, **p < 0.01 and ***p < 0.001 vs. vehicle group.

 • Page 8-9
 “In accordance with the previous study⁴⁹, the qRT-PCR analysis demonstrates the upregulation
 of HO-1 expression at 6 h and 12 h after the injection of DDP (15 mg/kg). However, the HO-1
 expression begins to decrease at 24 h, and becomes even lower than basal level at 72 h post-
 injection, which probably results from the oxidant injury induced by excessive ROS
 accumulation^{50,51} (Supplementary Fig. 18). Additionally, the NOX2 expression is found to be
 upregulated at 72 h post-injection, which coincides with the high cellular ROS level⁵².
 Dramatically, the reduced HO-1 and elevated NOX2 expression are substantially reserved to
 basal level upon CNPs treatment (Fig. 4j,k).”

 **References:**
 49. Agarwal, A., et al. Induction of heme oxygenase in toxic renal injury: A protective role in
 cisplatin nephrotoxicity in the rat. *Kidney Int.* **48**, 1298-1307 (1995).
 50. Behiry, S., et al. Effect of combination sildenafil and gemfibrozil on cisplatin-induced
 nephrotoxicity; role of heme oxygenase-1. *Ren. Fail.* **40**, 371-378 (2018).

- 51. Sahin, K., et al. Epigallocatechin-3-gallate activates Nrf2/HO-1 signaling pathway in
cisplatin-induced nephrotoxicity in rats. *Life Sci.* **87**, 240-245 (2010).
- 52. Bedard, K. & Krause, K. H. The NOX family of ROS-generating NADPH oxidases:
physiology and pathophysiology. *Physiol. Rev.* **87**, 245–313 (2007).

**Reviewer #3 (Remarks to the Author):**

Summary: The authors have reported about the ceria nanoparticles (CNPs) that possess the
variable catalytical activity depending on the environmental pH. These CNP particles can be
used as a nanomedicine to recover the acute kidney injury which incurred during chemotherapy.
CNPs act as small molecular antioxidants and ROS scavenging at the neutral pH but acidic pH
CNPs become inert due to the excessive H⁺ that disrupts the re-exposure of active catalytic
sites, allowing a buildup of chemotherapy-mediated ROS generation to kill cancer cells. The
authors have performed both the in-vitro and in vivo studies as a proof of concept. The authors
reported that CNPs have prevented the chemotherapeutic agents induced apoptosis of renal
cells in AKI by detoxing H₂O₂, as well as activating the Nrf2/ Keap1 signaling pathway to
regulate ROS-related genes for the restoration of redox homeostasis in the renal cells.

**Response:** *Thank you very much for your encouraging comments. Based on your kind*
*suggestions, we have made point-to-point responses. The manuscript was revised to*
*incorporate additional data, and the revised sentences are marked with yellow background.*

1. An enzyme is a word usually defines a biomolecule that performs a catalytic activity. I
strongly suggest the authors to change the word “enzymatic activity tunable” to “catalytic
activity tunable” CNPs.

**Response:** *Thank you for your kind suggestion. We have changed the description “enzymatic*
*activity tunable” to “catalytic activity tunable” throughout our manuscript.*

2. The authors have determined the size of the CNPs in chloroform, CNPs after DSPE-
PEGylation in water via TEM and DLS analysis. Please Include the size of CNPs obtained in
presence of H₂O₂ at pH 7.4 and 6.0, it will give a clear picture if there is any change in the
particle.

**Response:** *Thank you very much for your suggestion. The size of CNPs was determined using*
*a Zetasizer Nano ZS90 in the presence of H₂O₂ under different pH conditions. The results*
*showed that, before adding H₂O₂, the average sizes of CNPs at pH 6.0, pH 6.6 and pH 7.4*
*are ~15.2 nm, ~15.4 nm and ~15.7 nm, respectively. After the incubation with H₂O₂, no obvious*
*size change of CNPs was observed under each pH condition, indicating the excellent stability*
*of CNPs (Supplementary Fig. 6).*

**Our modification to the manuscript:** *The results were added as Supplementary Fig. 6 in the*
*revised supporting information. In addition, the following sentences and methods were added*
*on page 5 and 29 in the revised manuscript.*

• Supplementary Fig. 6

**Supplementary Figure 6.** The hydrodynamic size of CNPs in the presence of H₂O₂ at pH 6.0,
pH 6.6 and pH 7.4 at different time points (0, 10, 30, 60 min).

• Page 5

“Also, there is no obvious change in the size of CNPs after the incubation with H₂O₂ under
different pH conditions (Supplementary Fig. 6).”

• Page 29

**Characterization.**

“...The hydrodynamic size of DSPE-PEG_{2k} modified CNPs in the presence of H₂O₂ under
different pH conditions were detected using a Zetasizer Nano ZS90 (Malvern Instruments,
Worcestershire, UK).”

3. Page No. 74, authors mentioned that the DSPE-PEGylation CNPs was used to impart the
colloidal stability of the CNPs in aqueous solution. Describe, why you didn't use the DSPE-
PEGylation CNPs particles for the current study. Besides, to measure the free radical
scavenging activity of CNPs and DSPE-PEGylation CNPs using SOD assay and make a
comparative statement of CNPs and DSPE-PEGylation CNPs that prevent chemotherapy-
induced AKI and cancer.

**Response:** Thank you for your comment. We apologize for the confusion. The DSPE-PEG_{2K}
modified ceria nanoparticles (CNPs) were used for all the in vitro and in vivo experiments in
this study, given that without PEGylation, the CNPs are hydrophobic and not applicable to
aqueous systems. We have made clear the difference between CNPs and PEGylated CNPs in
the revised manuscript, and modified the description of “CNPs” to “hydrophobic CNPs” on
Page 4.

Per your suggestion, we have analyzed the free radical scavenging ability of CNPs via the
SOD assay kit. The results showed that CNPs exhibited SOD-like activity, which catalyzes ·O₂⁻
to H₂O₂ (Supplementary Fig. 4).

**Our modification to the manuscript:** The results were added as Supplementary Fig. 4 in the

revised supporting information. In addition, the following sentences and methods were added
on page 4 and 30 in the revised manuscript.

• Supplementary Fig. 4

**Supplementary Figure 4. a,b, SOD-like activity of CNPs at pH 7.4 (a) and pH 6.6 (b).**

• Page 4

“The hydrophobic CNPs with a diameter of ~ 3 nm (Fig. 1b) were synthesized via a modified
reverse micelle method²³, and 1,2-distearoyl-sn-glycero-3-phosphoethanolamine-N-[methoxy
(polyethylene glycol)-2,000] (DSPE-PEG_{2K}) was successfully surface modified
(Supplementary Fig. 1), imparting the colloidal stability of the CNPs in aqueous solution (with
a hydrodynamic diameter of ~ 9.7 nm) for long blood circulation²⁴⁻²⁶ (Fig. 1c and
Supplementary Fig. 2).”

“CNPs demonstrated superoxide dismutase (SOD)-like activity under neutral and acidic
condition (Supplementary Fig. 4), which catalyzes the dismutation of superoxide radicals ($\cdot\text{O}_2^-$)
into H_2O_2 ³¹.”

**References:**

31. Ward, M. B., et al. Superoxide dismutase activity enabled by a redox-active ligand rather
than metal. *Nat. Chem.* **10**, 1207–1212 (2018).

• Page 30

**Superoxide dismutase (SOD)-like activity of CNPs.**

The SOD-like activity of CNPs at different concentrations (50 μM, 100 μM, 150 μM and 400
μM) was measured by a Total Superoxide Dismutase Assay Kit with WST-8 (Beyotime, China)
under different pH conditions.

4. For assessing the cell viability what solvent was used as a vehicle for the study, clearly
specify in the materials and method.

**Response:** Thank you for your kind suggestion. The solvents used as vehicles were DMSO and
deionized water for DDP and CNPs, respectively. We have specified them in the revised
manuscript.

**Our modification to the manuscript:** *The following methods were added on page 31-32 in*
*the revised manuscript.*

• Page 31-32

**Cell viability assay.**

HK-2, ES-2, L02, HepG2, A549 and OVCAR8 cells were seeded into 96-well plates at 5,000
cells/well for 24 h. The culture solution was adjusted to pH 6.0 or pH 6.6 or pH 7.4 with
hydrochloric acid and sodium hydroxide. Then, 5 μ M paclitaxel (Taxol) or 10 μ M DDP and
different concentrations of CNPs (3.13, 6.25, 12.5, 25 and 50 μ M) were added into the 96-well
plate for the incubation of another 24 h. Specifically, CNPs were dissolved in deionized water
and DDP was dissolved in DMSO. The solution of CNPs and DDP were added into basic
medium containing 10% FBS, which were then incubated with cells. The solvent used as a
vehicle is the basic medium containing same volume of deionized water and DMSO as that of
CNPs and DDP solution, respectively.

HK-2 cells were seeded into 96-well plate at 5,000 cells/well for 24 h. Then, 50 μ M H₂O₂
and different concentrations of CNPs (0.78, 1.56, 3.13, 6.25, 12.5, 25 and 50 μ M) were
incubated with cells, and co-incubated in the medium for another 24 h. The solvent used as a
vehicle is the basic medium containing same volume of deionized water as that of CNPs and
H₂O₂ solution, respectively.

HK-2 and ES-2 cells were seeded into 96-well plates at 5,000 cells/well for 24 h. Then,
10 μ M cisplatin and (or) different concentrations of CAT-deficient CNPs (iCNPs) (3.13, 6.25,
12.5, 25 and 50 μ M) were incubated with cells, and co-incubated in the medium for another 24
517 h. The solvent used as a vehicle is the basic medium containing same volume of deionized
water as that of CNPs and DDP solution, respectively.

HK-2 cells were seeded into 96-well plates at 5,000 cells/well for 24 h. Then cells were
treated with 10 μ M DDP, 50 μ M NAC, 50 μ M CNPs, 10 μ M DDP plus 50 μ M CNPs as well
as 10 μ M DDP plus 50 μ M NAC, and co-incubated for another 24 h at pH 6.6. The solvent
used as a vehicle is the basic medium containing same volume of deionized water and DMSO
as that of CNPs and DDP solution, respectively.

A colorimetric SRB assay was performed to examine the cell viability. In detail, SRB (4
525 mg/ml) was added into 94-well plate after various treatments, then they were incubated for 30
526 min. Discard the liquid in 94-well plate and wash it with 1% acetic acid more than 5 times. A
527 microplate reader (Thermo, Fisher Scientific, Waltham, MA) was used to measure the
528 absorbance at 515 nm. Each sample was repeated in triplicate.

5. ES-2 cells are capable of forming tumor why the effect of CNPs over the cell viability (Fig
2a and b and supplementary data) shows a similar pattern of HK-2 cells when medium pH is
being changed? Like at pH 6.0 the cell viability is low and at pH 7.4 the cell viability increases
as molar volume of CNP increases, explain Why?

**Response:** *Thank you for your comment. DDP has been reported to induce the high level of*
*ROS, which can induce the apoptosis of tumour cells including ES-2 cells <Apoptosis 2007,*
*12, 1733–1742>. Moreover, high level of ROS was observed in HK-2 cells after DDP treatment,*
*as well as the sever cell apoptosis (Fig. 4a). These indicate that the imbalance of ROS plays*
*an important role both in tumour and normal cells. The synthesized CNPs actively decomposed*

H_2O_2 at pH 7.4 but are completely inert at pH 6.0 (Fig. 1d). Therefore, the DDP-induced
excessive lethal ROS in both HK-2 cells and ES-2 cells can be quenched upon CNPs treatment
at pH 7.4, which contributes to the restoration of cell viability. However, at pH 6.0, CNPs
could not prevent the DDP-induced cell death by detoxifying ROS since their antioxidant
activity is switched off; therefore, no obvious enhanced viability of ES-2 and HK-2 cells was
observed. Taken together, under neutral condition, CNPs can quench DDP-induced ROS to
rescue the viability of ES-2 cells and HK-2 cells; while under acidic condition, CNPs switch
off the ROS scavenging ability and thus cannot rescue the cell viability.

6. Why did author choose the ES-2 cells and Hk-2 cells? Did you see a similar response using
another cell type? because, normal cells are around the tumor site.

**Response:** Thank you for your valuable comments. DDP has been commonly utilized as anti-
tumour agent for treating various solid tumours <Pharmacol. Res. 2016, 106, 27–36>,
especially as a first-line drug for treating ovarian cancer <Ref. J. Clin. Oncol. 2018, 36, 2585-
2592; J. Clin. Oncol. 2016, 34, 2881-2887>. Therefore, we used ovarian carcinoma cell ES-2
to establish the tumour model. The HK-2 cells are representative renal epithelial cells that are
typical targets of DDP-induced nephrotoxicity, and have been extensively used by previous
studies on AKI <Ref. J. Clin. Invest. 2019, 129, 5033-5049; EBioMedicine. 2018, 36, 266-
280>. We have justified the usage of these two cell lines in the revised manuscript.

Per your suggestion, in addition to HK-2 and ES-2 cells, we further investigated more cell
types including another ovarian carcinoma (OVCAR8) cell line, hepatocellular carcinoma
(HepG2) cell line, lung carcinoma (A549) cell line and normal hepatocytes (L02). Additionally,
we have included the cell viability experiments after different treatments of cells at pH 6.6,
which is more relevant to the pH value of tumour extracellular microenvironment <Ref. Nature
Rev. Cancer 2011, 11, 671–677; Nat. Nanotech. 2016, 11, 724-730; Cancer Res. 2013, 73,
1524-1535; Biomaterials, 2019, 219, 119393>. These results revealed that CNPs could protect
ES-2 and HK-2 cells under neutral condition in a dose-dependent manner (pH 7.4), but not
under acidic condition (pH 6.0 and pH 6.6) (Fig.2a,b and Supplementary Fig. 8). Likewise, the
similar responses were observed in other tumour cell lines including OVCAR8, HepG2 and
A549 and normal cell lines L02 (Fig. 2c,d,e and Supplementary Figs. 8 and 11). Notably, CNPs
rescue the viability of L02 cells under neutral condition while maintaining the potency of DDP
to HepG2 cells under acidic condition, suggesting that CNPs can reduce the side effects of
chemotherapeutic agents to normal cells around tumour cells.

These results indicate the generality of CNPs' capacity that context-dependently protects
against AKI and maintain the lethal potency of chemotherapeutics to tumour in vitro.

**Our modification to the manuscript:** The results were added as Fig. 2, Supplementary Figs.
10 and 11 in the revised manuscript and supporting information. In addition, the following
sentences and methods were added on page 6, 12 and 30-32 in the revised manuscript.

• Fig. 2

**Fig. 2 CNPs display context-dependent catalytic activity *in vitro*.** **a**, The survival rate of
 HK-2 cells upon treatments with 10 μM DDP and different concentrations of CNPs at pH 7.4
 and pH 6.6. **b-e**, The survival rate of ES-2 cells (**b**), HepG2 cells (**c**), A549 cells (**d**) and
 OVCAR8 cells (**e**) upon treatments with 10 μM DDP and different concentrations of CNPs at
 pH 7.4 and pH 6.6. **f**, The survival rate of HK-2 cells upon treatments with 5 μM Taxol and
 different concentrations of CNPs at pH 7.4 and pH 6.6. **g**, The survival rate of EK-2 cells upon
 treatments with 5 μM Taxol and different concentrations of CNPs at pH 7.4 and pH 6.6.

• Supplementary Fig. 8

**Supplementary Figure 8. a**, The survival rate of HK-2 cells upon treatments with 10 μM DDP
 and different concentrations of CNPs at pH 6.0. **b**, The survival rate of L02 cells upon
 treatments with 10 μM DDP and different concentrations of CNPs at pH 6.0. **c-f**, The survival
 rate of A549 cells (c), HepG2 cells (d), OVCAR8 cells (e) and ES-2 cells (f) upon treatments
 with 10 μM DDP and different concentrations of CNPs at pH 6.0. **g**, The survival rate of HK-
 2 cells upon treatments with 5 μM Taxol and different concentrations of CNPs at pH 6.0. **h**,
 The survival rate of ES-2 cells upon treatments with 5 μM Taxol and different concentrations
 of CNPs at pH 6.0.

• Supplementary Fig. 11

**Supplementary Figure 11.** The survival rate of hepatocytes (L02) after treatments with 10 μM
 DDP and different concentrations of CNPs at pH 7.4 and pH 6.6.

• Page 6

[revised manuscript text omitted]

7. Whether authors performed the western blot for the expression of protein in the presence of
CNPs and without DDP. If yes include the result in the figure 4 (a) if not please justify why it
was not done.

**Response:** *Thank you for your suggestion. Accordingly, we have examined the protein*
*expression in the presence of CNPs alone. As shown in Fig. 5a (previous Fig. 4a), CNPs alone*
*only slightly increased the expression of Nrf2 and Keap1, and did not affect DJ-1 expression.*

**Our modification to the manuscript:** *The results were added as Fig. 5a in the revised*
*manuscript. In addition, the following sentences were added on page 6 in the revised*
*manuscript.*

**Fig. 5 CNPs counteract DDP-induced oxidative stress via activating *Nrf2/Keap1* signaling**
 **pathway.** **a**, Western blot analysis of the *Nrf2*, *Keap1* and *DJ-1* levels in HK-2 cells after
 **respective treatments with vehicle, CNPs, DDP and CNPs plus DDP.** **b**, **c**, **d**, Relative mRNA
 expression of *Nrf2* (**b**), *Keap1* (**c**) and *DJ-1* (**d**) in the renal cortex from each group. **e**,
 Schematic diagram of the experimental setup to evaluate the role of *Nrf2* by small interfering
 RNA (SiRNA) *in vitro* using western blot and qRT-PCR analysis. **f**, Western blot analysis of
 *Nrf2*, clv-PARP and clv-caspase-3 levels in Si*Nrf2*-transfected HK-2 cells. **g**, **h**, Relative
 mRNA expression of *HO-1* (**g**) and *NOX2* (**h**) in the respective groups. **i**, Schematic
 representation of the mechanism of CNPs in reducing ROS level. CNPs decompose DDP-
 induced H₂O₂, a predominant cellular ROS, into H₂O and O₂. Meanwhile, the minimal residual
 amounts of ROS activate the *Nrf2/Keap1* signaling pathway. Specifically, *Nrf2* moves into the
 nucleus and subsequently binds to the antioxidant response elements (ARE), leading to the
 upregulation of antioxidant gene (*HO-1*) and downregulation of pro-oxidant gene (*NOX2*).
 These genes regulation can further detoxify ROS. Each experiment was repeated three times
 independently. Data are presented as mean ± SEM; **p* < 0.05, ***p* < 0.01 and ****p* < 0.001 vs.
 vehicle group; #*p* < 0.05 and ##*p* < 0.01 vs. DDP group.

• Page 9

“There is no obvious change in the expression of these proteins in HK-2 cells treated with

CNPs alone (Fig. 5a).”

8. For studying the gene expression authors have not clearly explained the design of experiment
in the materials and method.

**Response:** *Thank you for your kind suggestion. We have provided detailed explanation of the*
*design of the gene expression experiment in the revised “Methods” section.*

**Our modification to the manuscript:** *The following methods were added on page 34 in the*
*revised manuscript.*

• Page 34

**RNA extraction and qRT-PCR.**

“To study the protective effect of CNPs on the kidneys, 15 mg/kg DDP and 0.5 mg/kg or 1.5
711 mg/kg CNPs were administered, respectively. After 72 h, the mice were sacrificed and the total
712 RNA were extracted from the kidneys of each group to detect *KIM-1*, *HO-1*, *NOX2*, *Nrf2*,
*Keap-1* and *DJ-1* mRNA levels.

HK-2 cells were transfected with Si*Nrf2* mixed with Lipofectamine 2000 transfection
reagent according to the manufacturer’s protocols. The medium (Opti-MEM) was replaced at
6 h post-transfection. After 24 h, the cells were treated with 10 μ M DDP and different
concentrations of CNPs (50 μ M). After another 24 h, HK-2 cells were collected by Trizol reagent
to extract RNA to measure the expressions of *HO-1* and *NOX2* mRNA.

Total RNA was isolated from kidney tissues using Trizol reagent (Invitrogen, Carlsbad,
California, USA) in accordance with the manufacturer’s protocol, and cDNA was prepared
using a cDNA preparation supermix (TransGen Biotech, Beijing, China). Quantitative RT-PCR
was carried out using SYBR Green Supermix (Bio-Rad, Hercules, California, USA) with 20
μ L reaction mixtures (primers are enlisted in Table S2). PCR reactions were performed on the
Quant Studio 6 Flex Real-Time PCR System (Applied Biosystems, Carlsbad, California, USA)
with the following program: step 1, 95 $^{\circ}$ C for 3 min to activate the Taq polymerase; step 2,
95 $^{\circ}$ C for 3 seconds to denaturize DNA; step 3, 60 $^{\circ}$ C for 31 seconds for annealing/extension
(39 cycles for steps 2 and 3). The relative mRNA levels were quantified by the $2^{-\Delta\Delta C_t}$ method
and all data were normalized to GAPDH (the internal control).”

9. Figure 2: Requesting the author to change the color of the asterisk symbol in the image 2 (g)
for the better visibility.

**Response:** *Thank you for your kind suggestion. We have changed the color and shape of the*
*symbol in the previous image 2 (g) (now Fig. 3e in the revised manuscript) for better*
*visibility. In the revised manuscript, the black triangles denote the formation of casts in tubes.*

**Our modification to the manuscript:** *We have replaced the yellow asterisk with black*
*triangles to denote casts formation in tubes in the image 2 (g) (now Fig. 3f in the revised*
*manuscript) for better visibility.*

**Fig. 3 CNPs protect against chemotherapy-induced AKI *in vivo*.** **a**, Schematic
 representation of treatment schedule and therapy assessments for AKI mice. **n** = 16, including
 8 male mice and 8 female mice in each group. **b**, **c**, Serum BUN (**b**) and Cre (**c**) levels of mice
 treated with vehicle, DDP, DDP and CNPs or iCNPs (inactive CNPs), respectively (**n** =
 16/group). **d**, Relative mRNA expression of *KIM-1* in the renal cortex from each group. (**n** =
 16/group). **e**, The tubular injury score was calculated according to the percentage of damaged
 tubules as reported: 0, no damage; 1, <25% damage; 2, 25-50% damage; 3, 50-75% damage;
 4, >75% damage. A pathologist evaluated 5 randomly selected fields per section of the mouse

kidneys at a magnification of $\times 400$ in a blind manner. Data are presented as means \pm standard
error of mean (SEM) ($n = 16/\text{group}$). **f**, Representative H&E sections of the kidney from **male**
**and female mice in** each group ($n = 16/\text{group}$). Arrows indicate tubules with necrosis, epithelial
anoikis cavitation, or loss of brush border. **Triangles denote the formation of casts in tubes.**
Scale bar: 100 μm . *** $p < 0.001$ vs. vehicle group and ### $p < 0.001$ vs. DDP group.

10. Figure 5: in the image d the scale bars for the days are not clearly visible but as far as it
seems like scales has started from day 0-12. In the case of vehicle, NAC, DDP+CNP and CNP
from the graph in the image by the tumor volume seems to be increasing but in the image d it
looks vice versa as of the tumor size being reduced from day 0-12 can authors explain it, Why?
Similar kind of observance is even found between the images of e and g.

**Response:** *Thank you for the comment. We feel sorry for the unclear graphic display. Actually,*
*in Fig. 5d and g (now Fig. 6d and g in the revised manuscript), the digital photographs of*
*tumour masses were all recorded on day 10, and the different tumours in the same row were*
*collected from different mice in the same group. The bottom of the previous picture is a ruler*
*used for informing the actual tumour sizes but not for the days following treatment. To make*
*our graphics clearer to readers, we have replaced the style of the scale bars in the revised*
*manuscript.*

**Our modification to the manuscript:** *We have replaced the rule with a scale bar for better*
*visibility in Fig. 6d,g.*

 **Fig. 6 CNPs improve the overall chemotherapeutic outcome *in vivo*.** **a**, Schematic diagram
 of ES-2 tumour xenograft establishment, treatment schedule and therapy assessments. Groups
 of nude mice were treated with DDP (*i.p.*, 3 mg/kg, twice one week, total for twice) and CNPs
 (*i.v.*, 1.5 mg/kg, twice one week, total for twice) or NAC (oral administration, 200 mg/kg, daily)
 respectively. Saline was used as a negative control. Soda water was used to raise the
 intratumoral pH of mice. **b**, Relative tumour volume of nude mice fed with sterile water in the

respective groups (n=5). **c**, Tumour inhibition rate of nude mice fed with sterile water in each
group. **d**, Images of the dissected tumours from groups of nude mice fed with sterile water after
treatment. **Scale bar: 10 mm.** **e**, Relative tumour volume of nude mice fed with soda water (200
mM, begun on the first day of dividing groups) in the respective groups. **f**, Tumour inhibition
rate of mice fed with soda water in each group. **g**, Images of the dissected tumours from groups
of nude mice fed with soda water after treatment. **Scale bar: 10 mm.** **h, i**, Serum BUN (**h**) and
Cre (**i**) levels of nude mice were measured on the last day. **j**, Survival probability of **tumour-**
**bearing** nude mice treated with DDP (*i.p.*, 3 mg/kg, twice one week, total for three times) and
(or) CNPs (*i.v.*, 1.5 mg/kg, twice one week, total for three times) throughout the observation
period (n=11/group). Data are presented as mean \pm SEM; *p < 0.05, ***p < 0.001 vs. vehicle
group; #p < 0.05, ##p < 0.01, ###p < 0.001 vs. DDP group.

11. Line 477: authors please write the full form for ICP-MS.

**Response:** *Thank you for your kind suggestion. We have added the full form of ICP-MS in the*
*revised manuscript.*

**Our modification to the manuscript:** *We have added the full form of ICP-MS on the page 29*
*in the revised manuscript.*

- • Page 29

The concentration of Ce was quantified using the **Inductively coupled plasma-Mass**
**Spectrometry (ICP-MS)**, PerkinElmer NexION 300X).

12. Line 495: represent the unit for temperature 37?

**Response:** *Thank you for your kind suggestion. We have added the temperature unit in the*
*manuscript.*

**Our modification to the manuscript:** *We have added the temperature unit on the page 31 in*
*the manuscript.*

- • Page 31

**Cell culture.**

...ES-2 cells grown in McCoy's 5A medium, **L02 and HepG2 cells grown in DMEM medium,**
**A549 cells grown in F-12 medium were at 37 °C** under a humidified atmosphere with 5% CO₂.

13. Line 510: the unit of time measured was represented as “min”

**Response:** *Thank you for pointing this out. We have corrected this issue throughout the revised*
*manuscript.*

**Our modification to the manuscript:**

- • Page 28

“To impart the colloidal stability of the CNPs in aqueous solution, 20 mg of DSPE-PEG_{2K} and
5 mL of 1 mg mL⁻¹ CNPs in chloroform were mixed and stirred for 10 **min** at room temperature.”

- • Page 30

“Additionally, CNPs were incubated with H₂O₂ solution under the acid condition for 15 **min**,
and then washed by deionized water for 3 times.”

• Page 35

“The supernatants of the mixture were obtained by centrifugation at 4 °C (30 min, 13,000
r.p.m.),”

• Page 36

Paraffin sections were first dewaxed in xylene for 5-10 min, then replaced with fresh xylene
and dewaxed again for 5-10 min. The resulting sections were treated with anhydrous ethanol
for 5 min, 90% ethanol for 2 min, 70% ethanol for 2 min, distilled water for 2 min, in proper
order. The deparaffinized sections of the kidney cortex were successively incubated with 20
833 µg/mL DNase-free proteinase K (20 min, room temperature) and TdT reaction mix (60 min,
37 °C) in the dark.

14. Line 551: represent seconds in abbreviated form

**Response:** *Thank you for your kind suggestion. We have changed “seconds” to its abbreviated*
*form “s” on page 34 in the revised manuscript.*

**Our modification to the manuscript:**

• Page 34

“step 2, 95 °C for 3 s to denaturize DNA; step 3, 60 °C for 31 s for annealing/extension (39
cycles for steps 2 and 3).”

15. Line 594: specify the units in abbreviated form for the minutes, authors please follow the
same form of unit expression throughout the paper.

**Response:** *Thank you for your kind suggestion. We have checked and unified the form of unit*
*throughout the paper.*

**Our modification to the manuscript:**

• Page 28

“To impart the colloidal stability of the CNPs in aqueous solution, 20 mg of DSPE-PEG_{2K} and
5 mL of 1 mg mL⁻¹ CNPs in chloroform were mixed and stirred for 10 min at room temperature.”

• Page 30

“Additionally, CNPs were incubated with H₂O₂ solution under the acid condition for 15 min,
and then washed by deionized water for 3 times.”

• Page 35

“The supernatants of the mixture were obtained by centrifugation at 4 °C (30 min, 13,000
r.p.m.),”

• Page 36

Paraffin sections were first dewaxed in xylene for 5-10 min, then replaced with fresh xylene
and dewaxed again for 5-10 min. The resulting sections were treated with anhydrous ethanol
for 5 min, 90% ethanol for 2 min, 70% ethanol for 2 min, distilled water for 2 min, in proper

order. The deparaffinized sections of the kidney cortex were successively incubated with 20
$\mu\text{g/mL}$ DNase-free proteinase K (20 min, room temperature) and TdT reaction mix (60 min,
37 °C) in the dark.

16. Line 619: samples were centrifuged at what speed (4)? Can you please specify it?

**Response:** *Thank you very much. Per your kind suggestion, we have specified the speed in the*
*revised manuscript.*

**Our modification to the manuscript:** *We have specified the speed on the page 34 in the*
*revised manuscript.*

• Page 37

“then the homogenates were filtered and centrifuged using a refrigerated centrifuge at 4 °C. (20
879 min, 13, 000 rpm.)”

17. The author needs to cite suitable references that CNPs already have phase transformation
in aqueous media, high antioxidative properties, wound healing effect and anti-inflammatory
properties.

Journal of Materials Research 34.3 (2019): 465-473.

Small. 2009 Dec;5(24):2848-56.,

Chemical Society Reviews 39 (11), 4422-4432

Biomaterials 33 (31), 7746-7755

Artificial cells, nanomedicine, and biotechnology 46.sup3 (2018): S956-S963.

**Response:** *Thank you very much. Based on your kind suggestion, we have cited these suitable*
*references in the revised manuscript.*

**Our modification to the manuscript:** *The following sentences and references were added on*
*page 4, 11 in the revised manuscript.*

• Page 4

“X-ray photoelectron spectroscopy (XPS) analysis indicates that Ce^{3+} and Ce^{4+} co-exist on the
surface of CNPs, providing the chemical basis for the catalytic activities²⁷⁻³⁰”

• Page 11

“Recently, ceria nanoparticles with antioxidant activities have been widely studied in
biomedical applications^{61,62}, such as pro-angiogenesis⁶³ and anti-inflammatory⁶⁴ therapies.”

**References:**

30. Kuchibhatla, S. V.N.T., et al. An unexpected phase transformation of ceria nanoparticles in
aqueous media. *J. Mater. Res.* **34**, 465-473 (2019).

61. Karakoti, A., Singh, S., Dowding, J. M., Seal, S. & Self, W. T. Redox-active radical
scavenging nanomaterials. *Chem. Soc. Rev.* **39**, 4422-4432 (2010).

62. Singh, S., et al. Cerium oxide nanoparticles at the nano-bio interface: size-dependent
cellular uptake. *Artif. Cells, Nanomed., Biotechnol.* **46**, S956–S963 (2018).

- 63. Das, S., et al. The induction of angiogenesis by cerium oxide nanoparticles through the
modulation of oxygen in intracellular environments. *Biomaterials* **33**, 7746-7755 (2012).
- 64. Hirst, S. M., et al. Anti-inflammatory properties of cerium oxide nanoparticles. *Small* **5**,
2848-2856 (2009).

**Reviewer #4 (Remarks to the Author):**

This manuscript described development of a nanoparticle that actively catalyzes ROS in non-
acidic environments. This is a clever approach to mitigate some chemotherapy induced
nephrotoxicity, or acute kidney disease (AKD), which can be a serious problem.

**Response:** *Thank you very much for your encouraging comments. Based on your kind*
*suggestions, we have made point-to-point responses. The manuscript was revised to*
*incorporate additional data, and the revised sentences are marked with yellow background.*

**MAJOR CONCERNS**

1. Experiments appear to be performed in one of two cell lines (HK-2 or ES-2) in different
lines of investigation and there is little or no justification. Further, the use of only one oxidizing
chemotherapeutic is a significant weakness. These two weaknesses argue against the generality
of this phenomenon.

**Response:** *Thank you very much for your constructive comments.*

*DDP has been commonly utilized as anti-tumour agent for various solid tumours <Nat.*
*Genet. 2009, 41, 1345–1349; Pharmacol. Res. 2016, 106, 27-36>, especially as a first-line*
*drug for treating ovarian cancer <Ref. J. Clin. Oncol. 2018, 36, 2585-2592; J. Clin. Oncol.*
*2016, 34, 2881-2887>. Therefore, we used ovarian carcinoma cell ES-2 to establish the tumour*
*model for this study. The HK-2 cells are representative renal epithelial cells that are typical*
*targets of DDP-induced nephrotoxicity, and have been extensively used by previous studies on*
*AKI <Ref. J. Clin. Invest. 2019, 129, 5033-5049; EBioMedicine. 2018, 36, 266-280>.*
*According to your comment, we have now justified the usage of these two cell lines in different*
*lines of investigation in the revised manuscript.*

*Furthermore, to examine the generality of our work, we employed several additional cell*
*lines, including ovarian carcinoma OVCAR8, hepatocellular carcinoma HepG2, and lung*
*carcinoma A549, along with normal hepatocytes L02. We find that the phenomena observed in*
*HK-2 and/or ES-2 cells are well reproduced in these cell lines (Fig. 2a-e, Supplementary Figs.*
*8 and 11).*

*We also used additional oxidizing chemotherapeutics in the revised manuscript, including*
*cyclophosphamide (CP) and paclitaxel (Taxol), to investigate whether the protective effects of*
*CNPs are confined to DDP-induced AKI or are generalizable. Results show that CNPs can*
*also improve the viability of Taxol-treated cells, in a pH-dependent manner as observed in*
*DDP-treated cells (Fig. 2f,g, Supplementary Fig. 8). More importantly, CNPs can protect*
*against CP-induced AKI in vivo (Supplementary Fig. 15a-d).*

*We believe these additional data verify the generality of CNPs' pH-switchable catalytic*
*activities that can be exploited to protect against chemotherapy-induced AKI without affecting*
*the therapeutic effects of chemotherapy drugs.*

**Our modification to the manuscript:** *The results were added as Fig. 2, Supplementary Figs.*
*8, 11 and 14 in the revised manuscript and supporting information. In addition, the following*
*sentences and methods were added on page 6, 7, 12 and 30-33 in the revised manuscript.*

• Fig. 2

**Fig. 2 CNPs display context-dependent catalytic activity *in vitro*.** a, The survival rate of
 HK-2 cells upon treatments with 10 μM DDP and different concentrations of CNPs at pH 7.4
 and pH 6.6. b-e, The survival rate of ES-2 cells (b), HepG2 cells (c), A549 cells (d) and
 OVCAR8 cells (e) upon treatments with 10 μM DDP and different concentrations of CNPs at
 pH 7.4 and pH 6.6. f, The survival rate of HK-2 cells upon treatments with 5 μM Taxol and
 different concentrations of CNPs at pH 7.4 and pH 6.6. g, The survival rate of EK-2 cells upon
 treatments with 5 μM Taxol and different concentrations of CNPs at pH 7.4 and pH 6.6.

• Supplementary Fig. 8

**Supplementary Figure 8. a**, The survival rate of HK-2 cells upon treatments with 10 µM DDP
 and different concentrations of CNPs at pH 6.0. **b**, The survival rate of L02 cells upon
 treatments with 10 µM DDP and different concentrations of CNPs at pH 6.0. **c-f**, The survival
 rate of A549 cells (**c**), HepG2 cells (**d**), OVCAR8 cells (**e**) and ES-2 cells (**f**) upon treatments
 with 10 µM DDP and different concentrations of CNPs at pH 6.0. **g**, The survival rate of HK-
 2 cells upon treatments with 5 µM Taxol and different concentrations of CNPs at pH 6.0. **h**,
 The survival rate of ES-2 cells upon treatments with 5 µM Taxol and different concentra-
 tions of CNPs at pH 6.0.

• Supplementary Fig. 11

**Supplementary Figure 11. The survival rate of hepatocytes (L02) after treatments with 10 µM**
 **DDP and different concentrations of CNPs at pH 7.4 and pH 6.6.**

**Supplementary Figure 14. CNPs protect against cyclophosphamide (CP)-induced AKI *in vivo*.**

**a**, Schematic representation of treatment schedule and therapy assessments for

cyclophosphamide (CP)-induced AKI mice. **b**, **c**, Serum BUN (**b**) and Cre (**c**) levels in each

group (n = 5/group). **d**, Representative H&E sections of the kidney from each group (n =

5/group). Arrows indicate tubules with necrosis, epithelial anoikis cavitation, or loss of brush

border. Triangles denote the formation of casts in tubes. The lower panel are the magnified
regions from the upper panel. The lower panel are the magnified regions from the upper panel.
Scale bar: 200 μm (up) or 100 μm (down), n = 5.

- Page 6

[revised manuscript text omitted]

2. The experiment shown in figure 5j appears to indicate that cisplatin 3 mg/kg q2d can kill
non tumor-bearing nude mice within 10 days and the CNPs can protect against this (as can
NAC). However, the data in figures 5b and 5e show the anti-tumor effect of 3 mg/kg q3d
cisplatin, without any noted killing of mice. If these mice were sick from cisplatin toxicity that
could explain the lack of tumor growth.

**Response:** *Thank you for your comment. First of all, we would like to make it clear that the*
*mice used in Fig. 5j (now Fig.6j in the revised manuscript) were tumour-bearing mice. Sorry*
*for the confusing, we made it clear in the legend of Fig.6j in the revised manuscript.*

*As you have noted, cisplatin at 3 mg/kg q2d (three times a week) can induce severe*
*nephrotoxicity, leading to tumour-bearing mice death (Fig. 6j), which makes it difficult for us*
*to investigate anti-tumour effects of cisplatin in the presence of CNPs. Nevertheless, we found*
*that the reduction of cisplatin administration frequency to 3 mg/kg q3d (twice a week) did not*
*cause significant toxicity, as indicated by the unaffected body weight of DDP-treated mice*
*(Supplementary Figs. 20 and 21). Importantly, at this dosage and administration frequency,*
*cisplatin showed evident tumour suppression effects (Fig. 6b and e). This allowed us to further*
*examine whether CNPs would affect the anti-tumour effects of cisplatin without causing severe*
*sickness of mice that, as you suggested, might perplex the interpretation of the results.*
*Moreover, CNPs have been proven to improve the survival of AKI mice by reducing the*
*cisplatin-induced nephrotoxicity (Fig. 6j), and notably, the mice co-treated with cisplatin and*
*CNPs show similar tumour inhibition rate to mice treated cisplatin alone (Fig. 6c). These*
*suggest that 3 mg/kg q3d cisplatin inhibits the tumour growth by its intrinsic toxicity to tumour*
*rather than undermining the overall health of mice.*

*We have discussed this important point in the revised manuscript.*

**Our modification to the manuscript:** *The results were added as Supplementary Figs. 20-*
*21, 23 and 25 in the revised supporting information. In addition, the following sentences and*
*methods were added on page 10 and 36 in the revised manuscript.*

Fig. 6 CNPs improve the overall chemotherapeutic outcome *in vivo*. **a**, Schematic diagram of ES-2 tumour xenograft establishment, treatment schedule and therapy assessments. Groups of nude mice were treated with DDP (*i.p.*, 3 mg/kg, twice one week, total for twice) and CNPs (*i.v.*, 1.5 mg/kg, twice one week, total for twice) or NAC (oral administration, 400 mg/kg, daily) respectively. Saline was used as a negative control. Soda water was used to raise the intratumoral pH of mice. **b**, Relative tumour volume of nude mice fed with sterile water in the respective groups (n=5). **c**, Tumour inhibition rate of nude mice fed with sterile water in each

1111 group. **d**, Images of the dissected tumours from groups of nude mice fed with sterile water after
 1112 treatment. **Scale bar: 10 mm.** **e**, Relative tumour volume of nude mice fed with soda water (200
 mM, begun on the first day of dividing groups) in the respective groups. **f**, Tumour inhibition
 rate of mice fed with soda water in each group. **g**, Images of the dissected tumours from groups
 of nude mice fed with soda water after treatment. **Scale bar: 10 mm.** **h, i**, Serum BUN (**h**) and
 Cre (**i**) levels of nude mice were measured on the last day. **j**, Survival probability of **tumour-**
 **bearing** nude mice treated with DDP (*i.p.*, 3 mg/kg, twice one week, total for three times) and
 (or) CNPs (*i.v.*, 1.5 mg/kg, twice one week, total for three times) throughout the observation
 period (n=11/group). Data are presented as mean \pm SEM; *p < 0.05, ***p < 0.001 vs. vehicle
 group; #p < 0.05, ##p < 0.01, ###p < 0.001 vs. DDP group.

 • Supplementary Fig. 20

**Supplementary Figure 20.** Body weight of tumour-bearing BALB/c nude mice fed with sterile
 water in day 0 and day 10 after treatment with DDP (*i.p.*, 3 mg/kg, twice a week, total for
 twice), NAC (oral administration, 400 mg/kg, daily) and (or) CNPs (*i.v.*, 1.5 mg/kg, twice a
 1127 week, total for twice).

- • Supplementary Fig. 21

**Supplementary Figure 21.** Body weight of tumour-bearing BALB/c nude mice fed with soda
 water in day 0 and day 10 after treatment with DDP (*i.p.*, 3 mg/kg, twice a week, total for twice),
 NAC (oral administration, 400 mg/kg, daily) and (or) CNPs (*i.v.*, 1.5 mg/kg, twice a week, total
 for twice).

• Supplementary Fig. 23

**Supplementary Figure 23.** The SOD activity (a) and MDA level (b) of tumour tissues from
BALB/c mice fed with sterile water after treatments.

• Supplementary Fig. 25

**Supplementary Figure 25.** The SOD activity (a) and MDA level (b) of tumour tissues from
BALB/c mice fed with soda water after treatment with NAC or CNPs.

• Page 10

“DDP (3 mg/kg, twice a week) was intraperitoneally administered into ES-2 tumour xenograft
nude mice, followed by CNPs or NAC treatment (Fig. 6a). The body weights of mice treated
with DDP show negligible change as compared to that of vehicle group (Supplementary Figs.
20 and 21), indicating that DDP at this dosage and administration frequency did not cause
significant toxicity. In accordance with *in vitro* results (Supplementary Figs. 9 and 22), the *in*
*vivo* anti-tumour efficiency of DDP is found to be suppressed by NAC due to the quenched
tumoricidal ROS, as confirmed by the enhanced SOD activity and reduced MDA level in the
tumour tissues (Supplementary Fig. 23a,b). However, it is not affected by CNPs (Fig. 6b,c,d,
Supplementary Fig. 24) whose ROS scavenging activities are turned off once exposed to the
acidic tumour microenvironment, showing no obvious change of SOD activity and MDA level

(Supplementary Fig. 23a,b). Intriguingly, CNPs can reverse the imbalance of ROS-related
elements induced by DDP (Supplementary Fig. 25a,b), and thus weakened the anti-tumour
efficiency of DPP when the intratumoral pH is artificially raised⁵⁸ (Fig. 6e,f,g). These *in vivo*
results clearly show that, compared with NAC, CNPs can regulate the ROS scavenging activity
in a context-dependent manner, thereby exerting no effect on the potency of chemotherapeutic
agents in the acidic tumour microenvironment.”

• Page 37

**MDA and SOD assays.**

“To assess the oxidative stress level in HK-2, ES-2 cells, kidney tissues and tumour tissues, the
levels of superoxide dismutase (SOD) and malondialdehyde (MDA) were tested with
commercially available kits according to the manufacturer’s instructions (Beyotime
Biotechnology, China). Briefly, HK-2, ES-2 cells, kidney tissues and tumour tissues were
homogenized in ice-cold 0.1 M phosphate buffer (pH 7.4), then the homogenates were filtered
and centrifuged using a refrigerated centrifuge at 4 °C (20 min, 13,000 rpm.). The obtained
supernatants were used to determine the SOD enzyme activity and the lipid peroxidation level
by measuring MDA content. The SOD enzyme activity was expressed as a unit of activity per
milligram of protein and the MDA content was expressed as micromole per gram of protein.”

3. Figure 2a and b are confusing. It appears that HK-2 cells were treated at pH 7.4 and ES-2
cells at 6.0, and then the differences ascribed to the differences in pH? This cannot be so. At
the very least this experiment has to be performed in both cell lines (and at more physiological
pH values).

**Response:** *Thank you for your comment. We agree with the reviewer that these experiments*
*should be performed in both HK-2 and ES-2 cell lines and under more physiological pH values,*
*which will strength our conclusion. Accordingly, the viability of these two kinds of cell lines*
*upon CNPs treatment was further examined at tumour relevant pH 6.6, which showed a similar*
*trend as observed at pH 6.0.*

**Our modification to the manuscript:** *The results were added as Fig. 2a,b and Supplementary*
*Fig. 8 in the revised manuscript and supporting information. In addition, the following*
*sentences were added on page 6 and 31 in the revised manuscript.*

• Fig. 2

**Fig. 2 CNPs display context-dependent catalytic activity *in vitro*.** **a**, The survival rate of
 HK-2 cells upon treatments with 10 μM DDP and different concentrations of CNPs at pH 7.4
 and pH 6.6. **b-e**, The survival rate of ES-2 cells (**b**), HepG2 cells (**c**), A549 cells (**d**) and
 OVCAR8 cells (**e**) upon treatments with 10 μM DDP and different concentrations of CNPs at
 pH 7.4 and pH 6.6. **f**, The survival rate of HK-2 cells upon treatments with 5 μM Taxol and
 different concentrations of CNPs at pH 7.4 and pH 6.6. **g**, The survival rate of EK-2 cells upon
 treatments with 5 μM Taxol and different concentrations of CNPs at pH 7.4 and pH 6.6.

• Supplementary Fig. 8

**Supplementary Figure 8. a,** The survival rate of HK-2 cells upon treatments with 10 μM DDP
 and different concentrations of CNPs at pH 6.0. **b,** The survival rate of L02 cells upon
 treatments with 10 μM DDP and different concentrations of CNPs at pH 6.0. **c-f,** The survival
 rate of A549 cells (**c**), HepG2 cells (**d**), OVCAR8 cells (**e**) and ES-2 cells (**f**) upon treatments
 with 10 μM DDP and different concentrations of CNPs at pH 6.0. **g,** The survival rate of HK-
 2 cells upon treatments with 5 μM Taxol and different concentrations of CNPs at pH 6.0. **h,**
 The survival rate of ES-2 cells upon treatments with 5 μM Taxol and different concentra-
 tions of CNPs at pH 6.0.

• Page 6

“Interestingly, the DDP-induced cytotoxicity is significantly reduced by CNPs at pH 7.4 but
 not affected at pH 6.6 and pH 6.0 (Fig 2a,b, Supplementary Fig. 8)”

• Page 31

**Cell viability assay.**

“HK-2, ES-2, L02, HepG2, A549 and OVCAR8 cells were seeded into 96-well plates at 5,000
 cells/well for 24 h. The culture solution was adjusted to pH 6.0 or pH 6.6 or pH 7.4 with
 hydrochloric acid and sodium hydroxide. Then, 5 μM paclitaxel (Taxol) or 10 μM DDP and
 different concentrations of CNPs (3.13, 6.25, 12.5, 25 and 50 μM) were added into the cells for
 the incubation of another 24 h. ...”

4. Characterization of activity at pH 6.0 is probably not relevant to tumor pH, which is typically
cited as being in the range of 6.6-6.7. How do these nanoparticles behave at 6.6?

**Response:** *Thank you for your helpful comment. We agree with the reviewer that pH 6.6 is*
*more relevant to tumour pH, accordingly, we have systemically investigated how CNPs behave*
*at pH 6.6. As shown in Fig. 1d, Fig. 2 and Supplementary Fig. 8, there are similar*
*performances of CNPs at pH 6.6 and pH 6.0, which does not affect the original conclusion*

**Our modification to the manuscript:** *The results were added as Fig. 1d, Fig. 2 and*
*Supplementary Fig. 11 in the revised manuscript and supporting information. In addition, the*
*following methods were added on page 30 in the revised manuscript.*

Fig. 1 Design and characterization of catalytic activity tunable CNPs that protect against chemotherapy-induced AKI. **a**, Schematic illustration of catalytic activity tunable CNPs with context-dependent cytoprotective activities for AKI prevention during chemotherapy. High-concentration of chemotherapeutics in the kidneys induced AKI by producing excessive ROS. In the renal cortex, the administered ultrafine CNPs were switched “on”, and could counteract toxic ROS to prevent AKI. In the tumour acidic microenvironment, CNPs would be switched “off” by the high-level H^+ , and exert no effect on intratumoral ROS, maintaining the efficacy of chemotherapy. **b**, TEM image of ultrafine CNPs in chloroform, scale bar: 50 nm. Insert: high-resolution TEM image of CNPs, scale bar: 2 nm. **c**, TEM image of ultrafine CNPs after DSPE-PEGylation in water, scale bar: 50 nm. Insert: dynamic light scattering (DLS) of DSPE-PEGylated CNPs. **d**, The oxygen (O_2) production of CNPs under different pH conditions (pH 7.4, pH 6.6 and pH 6.0) during the reaction with H_2O_2 . **e**, Raman spectra of CNPs at different time points in each cycle of reaction with H_2O_2 under different pH conditions (pH 7.4 and pH 6.0). **f**, Schematic illustration of the context-dependent catalase-like activity of CNPs under different pH conditions (pH 7.4 and pH 6.0). V_o , oxygen vacancy.

• Fig. 2

**Fig. 2 CNPs display context-dependent catalytic activity *in vitro*.** a, The survival rate of
 HK-2 cells upon treatments with 10 μM DDP and different concentrations of CNPs at pH 7.4
 and pH 6.6. b-e, The survival rate of ES-2 cells (b), HepG2 cells (c), A549 cells (d) and
 OVCAR8 cells (e) upon treatments with 10 μM DDP and different concentrations of CNPs at
 pH 7.4 and pH 6.6. f, The survival rate of HK-2 cells upon treatments with 5 μM Taxol and
 different concentrations of CNPs at pH 7.4 and pH 6.6. g, The survival rate of ES-2 cells upon
 treatments with 5 μM Taxol and different concentrations of CNPs at pH 7.4 and pH 6.6.

• Supplementary Fig. 8

**Supplementary Figure 8. a**, The survival rate of HK-2 cells upon treatments with 10 μM DDP
 and different concentrations of CNPs at pH 6.0. **b**, The survival rate of L02 cells upon
 treatments with 10 μM DDP and different concentrations of CNPs at pH 6.0. **c-f**, The survival
 rate of A549 cells (**c**), HepG2 cells (**d**), OVCAR8 cells (**e**) and ES-2 cells (**f**) upon treatments
 with 10 μM DDP and different concentrations of CNPs at pH 6.0. **g**, The survival rate of HK-
 2 cells upon treatments with 5 μM Taxol and different concentrations of CNPs at pH 6.0. **h**,
 The survival rate of ES-2 cells upon treatments with 5 μM Taxol and different concentra-
 tions of CNPs at pH 6.0.

• Page 30

**Catalase-like activity of CNPs.**

“Catalase-like activity assays of CNPs under different pH conditions were carried out at room
 temperature and the generated oxygen was measured using a specific oxygen electrode on
 Multi-Parameter Analyzer (JPSJ-606L, Leici China). Specifically, materials were added into
 8.0 mL buffer solution (0.1 M PBS buffer, pH 7.4, pH 6.6 or pH 6.0) in the order of 0.8 ml 30%
 H₂O₂ solution and 50 μM CNPs. The generated O₂ (mg·L⁻¹) was measured at different reaction
 1279 times. ...”

**MORE MINOR CONCERNS**

5. The protective effects of CNPs against cisplatin nephrotoxicity in figures 2 and 3d, and 3e
 are impressive. However, controls using catalase-deficient CNPs must be included.

**Response:** *Thank you for your valuable suggestion. Per your suggestion, controls using*
*catalase-deficient CNPs (inactive CNPs, iCNPs) are investigated and the results are included*
*in Fig. 2 and Fig. 3d,e (now Fig.3 and Fig. 4d,e). iCNPs were obtained by pretreatment with*
*H₂O₂ and H⁺ overnight at room temperature (Supplementary Fig. 7). As shown in Fig. 3 and*
*Supplementary Fig. 15, iCNPs could not protect against cisplatin-induced nephrotoxicity*
*either in vitro or in vivo.*

**Our modification to the manuscript:** *The results were added as Fig. 3 and Supplementary*
*Figs. 10 and 15 in the revised manuscript and supporting information. In addition, the*
*following sentences and the methods were added on page 5-8, 12, 30 and 32 in the revised*
*manuscript, respectively.*

**Fig. 3 CNPs protect against chemotherapy-induced AKI *in vivo*.** a, Schematic
 representation of treatment schedule and therapy assessments for AKI mice. n = 16, including
 8 male mice and 8 female mice in each group. b, c, Serum BUN (b) and Cre (c) levels of mice
 treated with vehicle, DDP, DDP and CNPs or iCNPs (inactive CNPs), respectively (n =
 16/group). d, Relative mRNA expression of *KIM-1* in the renal cortex from each group. (n =
 16/group). e, The tubular injury score was calculated according to the percentage of damaged
 tubules as reported: 0, no damage; 1, <25% damage; 2, 25-50% damage; 3, 50-75% damage;
 4, >75% damage. A pathologist evaluated 5 randomly selected fields per section of the mouse

kidneys at a magnification of $\times 400$ in a blind manner. Data are presented as means \pm standard
 error of mean (SEM) ($n = 16/\text{group}$). **f**, Representative H&E sections of the kidney from **male**
 **and female mice in** each group ($n = 16/\text{group}$). Arrows indicate tubules with necrosis, epithelial
 anoikis cavitation, or loss of brush border. **Triangles denote the formation of casts in tubes.**
 Scale bar: 100 μm . ******* $p < 0.001$ vs. vehicle group and **###** $p < 0.001$ vs. DDP group.

• Supplementary Fig. 10

**Supplementary Figure 10. a,b**, The survival rate of **HK-2** cells (**a**) and ES-2 cells (**b**) after
 treatments with 10 μM DDP and different concentrations of inactive CNPs at pH 7.4. The
 inactive CNPs were obtained by pre-treatment with H_2O_2 under acidic condition.

• Supplementary Fig. 15

**Supplementary Figure 15. a**, Representative TUNEL staining images in kidneys from mice
 treated with CAT-deficient CNPs (iCNPs). Scale bar: 100 μm . Red dotted line indicates the
 boundary between the kidney edge and background, and the white dotted line represents the
 boundary between the renal cortex and the medulla. **b**, Quantification of TUNEL positive cells
 in the respective groups. Data are presented as mean \pm SEM; $n = 6/\text{group}$; ******* $p < 0.05$ vs.
 vehicle group.

• Page 5

“Notably, **inactive CNPs (iCNPs, H_2O_2 and H^+ pretreated CNPs)** lose their anti-oxidant activity,
 **which** is difficult to reverse (Supplementary Fig. 7).”

• Page 6
“Furthermore, the iCNPs show no cytoprotective effects on HK-2 cells and ES-2 cells under
neutral condition since their anti-oxidant activity is irreversibly lost upon pre-treatment with
H₂O₂ under acidic condition (Supplementary Fig. 10).”

• Page 7
“We further evaluated the kidney protective effect of CNPs *in vivo*. As shown in Fig. 3a, ICR
mice (8 male mice and 8 female mice) were intraperitoneally injected (*i.p.*) with a single dose
of DPP (15 mg/kg) to induce AKI, accompanied by intravenous injection (*i.v.*) of saline, CNPs
(0.5 or 1.5 mg/kg) or iCNPs (0.5 or 1.5 mg/kg), respectively.”

“In contrast, the iCNPs show no renal protection effects in DDP-treated mice.”

• Page 8
“In contrast, iCNPs show no anti-apoptotic effect *in vivo* (Supplementary Fig. 15).”

• Page 12
“In contrast, iCNPs with deficient ROS scavenging ability has no protective effects against
DDP-induced AKI *in vitro* and *in vivo* (Fig. 3, Supplementary Figs. 10 and 15), indicating the
indispensable role of CNPs’ antioxidant activity for AKI protection.”

• Page 31-32
**Cell viability assay.**
“HK-2 and ES-2 cells were seeded into 96-well plates at 5,000 cells/well for 24 h. Then, 10
μM cisplatin and (or) different concentrations of CAT-deficient CNPs (iCNPs) (3.13, 6.25, 12.5,
25 and 50 μM) were added into the cells, and co-incubated in the medium for another 24 h.
The solvent used as a vehicle is the basic medium containing same volume of deionized water
as that of CNPs and DDP solution, respectively. ...”

• Page 33
**Treatment of AKI mice.**
“(1) The DDP-induced AKI mice were intravenously injected with different dosages of CNPs
(0.5 or 1.5 mg/kg) or iCNPs (0.5 or 1.5 mg/kg). Control group animals were intraperitoneally
injected with an equal volume of saline. All mice were sacrificed at 72 h after cisplatin injection,
and their blood and kidneys were collected for evaluation of renal function and tissue damage.
Furthermore, organs such as the heart, liver, spleen, lungs and kidneys were also harvested. ...”

6. Fig 1 seems to indicate that these nanoparticles self-adhere in aqueous solution. Were there
attempts to keep these dispersed? While 3-7 nm is great for tissue penetration, this may be
hampered if these exist as clusters.

**Response:** Thank you for your valuable comments. In this study, DSPE-mPEG_{2k} modification
was used to keep CNPs dispersed in aqueous solution < Angew. Chem. Int. Ed. 2012, 51, 11039
1372 –11043. DOI: 10.1002/anie.201203780>. We believe the partial aggregation in Fig. 1c was
1373 introduced during TEM sample preparation that involve solvent volatilization and nanoparticle

*cle concentration. We consider the DLS results in Fig. 1c and the stability analysis in*
*Supplementary Fig. 2a may better reflect the dispersibility of CNPs modified with DSPE-*
*mPEG_{2k}, which have a stable hydrodynamic diameter of ~10 nm without self-adhering in*
*aqueous solutions for over one week. We agree that this good dispersibility of CNPs can*
*facilitate their tissue penetration.*

**Our modification to the manuscript:** *The results were added as Supplementary Fig. 2 in the*
*revised supporting information. In addition, the following methods were added on page 29 in*
*the revised manuscript.*

- • Supplementary Fig. 2

**Supplementary Figure 2. a,b, The sizes (a) and zeta potentials (b) of CNPs over a week.**

- • Page 29

**Characterization.**

“...The hydrodynamic size **and zeta potentials** of DSPE-PEG_{2k} modified CNPs were detected
using a Zetasizer Nano ZS90 (Malvern Instruments, Worcestershire, UK).”

7. The role of ROS in the etiology of chemotherapy induced AKD is far from clear and could
be better developed. In the landmark NEJM review by Rosner and Perazella (ref 3), ROS were
not mentioned once. In ref 9, NAC pretreatment mitigated the rise in IL-6 in *Nfr2^{-/-}* mice injured
with ischemia. What does this have to do with chemotherapy induced toxicity? Hence, this may
be a solution in search of a problem. Further, if NAC can prevent renal toxicity, why is it not
being used in the clinic, as it is approved?

**Response:** *Thank you very much for your comment. We agree that the role of ROS in the*
*etiology of chemotherapy induced AKD need to be better described. Therefore, we have*
*expanded the introduction on the pathological role of ROS in chemotherapy-induced AKD and*
*cited more relevant literatures for the readers to refer <Ref. Nat. Mater. 2019, 18, 1133-1143;*
*Nat. Rev. Nephrol. 2018, 14, 217-230; Nat. Rev. Nephrol. 2018, 14, 378-393; N. Engl. J. Med.*
*2017, 376, 1770-1781>. Notably, the NEJM review by Rosner and Perazella mentioned that*
*“These drugs can induce direct cellular toxicity as a result of their transport through tubular*
*cells, induction of mitochondrial injury, oxidative stress, and activation of apoptotic signaling*
*pathways within cells.” As we know, mitochondrial injury and oxidative stress are tightly*

related to the ROS production, which indicates the significance of ROS in the chemotherapy-
 induced AKD. For better clarity, we have corrected the ref.9 to a more relevant reference
 regarding the role of ROS in the etiology of chemotherapy induced toxicity. (Brit. J. Pharmacol.
 2008, 153, 1364-1372).

In fact, NAC has been used clinically for the prevention of contrast agent-induced
 nephropathy <Ref. N. Engl. J. Med. 2000, 343, 180–184; J. Am. Coll. Cardiol. 2002, 40, 1383–
 1388>. However, the antioxidant activity of NAC is non-selective, which may compromise
 the anti-tumour efficacy of oxidizing chemotherapeutics at tumour sites <J. Pharmacol. Exp.
 Ther. 2005, 312, 424-431>, thereby hindering their potential application for chemotherapy-
 induced AKI prevention in clinic. Considering the CNPs' context-dependent cytoprotective
 effects against chemotherapeutics in renal cells and tumour cells, we anticipate CNPs would
 be promising candidates for the prevention of chemotherapy-induced AKI.

**Our modification to the manuscript:** The results were added as Supplementary Figs. 22 and
 23 in the revised supporting information. In addition, the following sentences and methods
 were added on page 3, 10 and 36 in the revised manuscript.

- • Supplementary Fig. 22

**Supplementary Figure 22.** SOD activity in ES-2 cells treated with DDP, NAC, CNPs,
 DDP+NAC and DDP+CNPs under acidic condition.

- • Supplementary Fig. 23

**Supplementary Figure 23.** The SOD activity (a) and MDA level (b) of tumour tissues from

BALB/c mice fed with sterile water after treatments.

• Page 3

"... N-acetyl cysteine (NAC), can be utilized as an alternative to protect against chemotherapy-
induced AKI by reducing oxidative damage⁹."

**Reference:**

9. Chen, N., Aleksa, K., Woodland, C., Rieder, M. & Koren, G. N-Acetylcysteine prevents
ifosfamide-induced nephrotoxicity in rats. *Brit. J. Pharmacol.* **153**, 1364-1372 (2008).

• Page 10

"In accordance with *in vitro* results (Supplementary Figs. 9 and 22), the *in vivo* anti-tumour
efficiency of DDP is found to be suppressed by NAC due to the quenched tumoricidal ROS, as
confirmed by the increased SOD activity and reduced MDA level in the tumour tissues
(Supplementary Fig. 23a,b)."

"These *in vivo* results clearly show that, compared with NAC, CNPs can regulate the ROS
scavenging activity in a context-dependent manner, thereby exerting no effect on the potency
of chemotherapeutic agents in the acidic tumour microenvironment."

• Page 37

**MDA and SOD assays.**

"To assess the oxidative stress level in HK-2, ES-2 cells, kidney tissues and tumour tissues, the
levels of superoxide dismutase (SOD) and malondialdehyde (MDA) were tested with
commercially available kits according to the manufacturer's instructions (Beyotime
Biotechnology, China). Briefly, HK-2, ES-2 cells, kidney tissues and tumour tissues were
homogenized in ice-cold 0.1 M phosphate buffer (pH 7.4), then the homogenates were filtered
and centrifuged using a refrigerated centrifuge at 4 °C (20 min, 13,000 rpm.). The obtained
supernatants were used to determine the SOD enzyme activity and the lipid peroxidation level
by measuring MDA content. The SOD enzyme activity was expressed as a unit of activity per
milligram of protein and the MDA content was expressed as micromole per gram of protein."

Reviewers' Comments:

Reviewer #1:

Remarks to the Author:

The authors have well addressed all my concerns, and it can be published at present format.

Reviewer #2:

Remarks to the Author:

the authors have most satisfactorily addressed all the comments from the initial review.

Reviewer #4:

Remarks to the Author:

The Authors have done an excellent job of responding to my (Reviewer 4) and other reviewers' concerns. I have one lagging issue:

The response to Reviewer 4, comment 2, is still confusing.

First they state that "cisplatin at 3 mg/kg q2d (three times a week) can induce severe nephrotoxicity" and that "reduction of cisplatin administration frequency to 3 mg/kg q3d (twice a week) did not cause significant toxicity", but then they state "CNPs have been proven to improve the survival of AKI mice by reducing the cisplatin-induced nephrotoxicity". How can this last statement be correct if there is no "significant toxicity"? This was not clarified in the marked up manuscript version.

Robert J. Gillies

Reviewer #5:

Remarks to the Author:

Overall, authors were assertive and properly addressed the queries raised by the previous reviewers. The originality of the manuscript is supported by the obtained results. One important aspect from the characterization point of view is on the properties of modified CNPs. The attachment of DSPE-PEG2K to CNPs was proposed to be confirmed by FTIR. It is mentioned that peaks at 1116 cm^{-1} (C–O–C stretching) in the FT-IR spectra demonstrate the new link. First, FTIR is characterized by bands, not peaks. Second, the control of the physical mixture of DSPE and PEG2K is missing.

It would be expected, also, a significant change on zeta potential, as the surface PEGylation is known to provide neutral charges to nanosystems. Further discussion is needed.

Another conceptual issue is related with the proof of concept. CNPs are expected to act at the renal level, but it is not obvious how those nanoparticles are being actively targeting the kidney cells and retain there. PK values of the nanoparticles are required.

REVIEWER COMMENTS

Reviewer #1 (Remarks to the Author):

The authors have well addressed all my concerns, and it can be published at present format.

Response: *We thank the reviewer for supporting the publication of our manuscript.*

Reviewer #2 (Remarks to the Author):

the authors have most satisfactorily addressed all the comments from the initial review.

Response: *We thank the reviewer for the positive comments.*

Reviewer #4 (Remarks to the Author):

The Authors have done an excellent job of responding to my (Reviewer 4) and other reviewers' concerns. I have one lagging issue:

The response to Reviewer 4, comment 2, is still confusing.

First they state that “cisplatin at 3 mg/kg q2d (three times a week) can induce severe nephrotoxicity” and that “reduction of cisplatin administration frequency to 3 mg/kg q3d (twice a week) did not cause significant toxicity”, but then they state “CNP_s have been proven to improve the survival of AKI mice by reducing the cisplatin-induced nephrotoxicity”. How can this last statement be correct if there is no “significant toxicity”? This was not clarified in the marked up manuscript version.

Robert J. Gillies

Response: *We thank the reviewer for the positive comments. In fact, our results showed that cisplatin at 3 mg/kg q3d (twice a week) could induce nephrotoxicity without leading to mouse death, which allows us to investigate the anti-tumour effects of cisplatin in the presence of CNP_s since no mice died during the whole treatment. Furthermore, to fully evaluate the renal protective effects of CNP_s during higher intensity chemotherapy, we treated mice with cisplatin at 3 mg/kg q2d (three times a week). Indeed, without CNP_s treatment, cisplatin at 3 mg/kg q2d (three times a week) could induce severe nephrotoxicity, leading to the death of mice. However, the CNP_s treatment can significantly increase the survival rate of tumour-bearing mice receiving more frequent administration of cisplatin, attributing to their renal protective effects. To avoid possible confusing, we have modified the related sentences in the revised manuscript for clear expression.*

Our modification to the manuscript: *The following sentences were added on page 10 and 11 in the revised manuscript.*

- Page 10

“.....indicating that DDP at this dosage and administration frequency **did not cause severe toxicity that can threaten the survival of tumour-bearing mice.**”

- Page 11

“Finally, we investigated the effect of CNPs on the survival of tumour-bearing mice receiving **more frequent administration of DDP (3 mg/kg, three times a week). None of the mice with high frequency DDP treatment can survive more than 10 days due to the severe side effects**⁶².”

Reviewer #5 (Remarks to the Author):

Overall, authors were assertive and properly addressed the queries raised by the previous reviewers. The originality of the manuscript is supported by the obtained results.

One important aspect from the characterization point of view is on the properties of modified CNPs. The attachment of DSPE-PEG2K to CNPs was proposed to be confirmed by FTIR. It is mentioned that peaks at 1116 cm⁻¹ (C–O–C stretching) in the FT-IR spectra demonstrate the new link. First, FTIR is characterized by bands, not peaks. Second, the control of the physical mixture of DSPE and PEG2K is missing. It would be expected, also, a significant change on zeta potential, as the surface PEGylation is known to provide neutral charges to nanosystems. Further discussion is needed. Another conceptual issue is related with the proof of concept. CNPs are expected to act at the renal level, but it is not obvious how those nanoparticles are being actively targeting the kidney cells and retain there. PK values of the nanoparticles are required.

Response: *Thank you for your encouraging comments, and we truly appreciate your suggestions which helped us to improve our manuscript. Firstly, we have changed the description of “peaks” to “bands” in terms of FT-IR analysis. Then, per your suggestion, we have added the FT-IR spectra analysis of the physical mixture of DSPE and PEG-COOH as a control. We found that the characteristic band at 1550-1510 cm⁻¹ (Amide II, C-N stretching plus N-H bending) appeared in the FT-IR spectra of commercially available DSPE-PEG in comparison to the mixture of DSPE and PEG-COOH, indicating the covalent binding between DSPE and PEG-COOH <Ref. J. Mol.*

Struct. 2016, 1121, 86-92>. Certainly, the FT-IR spectra of DSPE-PEG and the physical mixture of DSPE and PEG-COOH both exhibit characteristic bands of PEG at 1116 cm^{-1} (C–O–C stretching). After surface modification, the unattached DSPE-PEG in solution was filtered out using ultrafiltration tubes (MW=30,000). The spectra of the resulted DSPE-PEG coated ceria nanoparticles (CNPs) exhibit characteristic bands both at $1550\text{-}1510\text{ cm}^{-1}$ and 1116 cm^{-1} , demonstrating the successful attachment of intact DSPE-PEG on hydrophobic CNPs.

Since the hydrophobic ceria nanoparticles easily undergo aggregation in aqueous solution, it is infeasible to directly measure the zeta potential before DSPE-PEG modification <Ref. *Biomaterials* 2007, 28, 4991-4999>. We agree with the reviewer that PEGylation can provide neutral charges to nano-systems <Ref. *RSC Adv.* 2013, 3, 6085; *J. Control. Release* 2006, 114, 343-347>, and the surface charge of CNPs was measured using a Zetasizer Nano ZS90. The results showed that CNPs are negatively charged (surface zeta potential of $\sim -20.4\text{ mV}$) attributed by the negative phosphate group of DSPE-PEG <Ref. *Angew. Chem.* 2018, 130, 9552–9556; *Chem-Asian J.* 2015, 10, 370-376; *Int. J. Pharmaceut.* 2016, 510, 255-262>.

Per your suggestion, we have further studied the pharmacokinetic (PK) profiles of CNPs in AKI mice. The blood circulation profiles of CNPs fit well with the classic two-compartment pharmacokinetic model, and the terminal elimination half-lives of the central component and peripheral component are calculated to be 0.47 h and 72.2 h, respectively (Supplementary Fig. 13). The renal cortex is highly vulnerable to chemotherapeutics such as DDP <Ref. *Int. J. Nanomed.* 2014, 9, 1065–1082; *Expert Opin. Drug Met.* 2018, 14, 937-950>, and CNPs are found to accumulate there and retain for a long time (Supplementary Fig. 15). Furthermore, CNPs are found in the renal tubules epithelial cell cilia, glomerular basement membrane (GBM) and renal tubules, as indicated by TEM images of renal cortex (Supplementary Fig. 16). The renal accumulation and retention of CNPs are beneficial from the small core size ($\sim 3\text{ nm}$) <Ref. *APL Mater.* 2017, 5, 053406>, deformable PEG surface-modification <Ref. *Nat. Rev. Matter.* 2018, 3, 358–374> and negative surface charge of CNPs <Ref. *ACS Nano* 2016, 10, 387–395>, as well as enhanced GBM permeability of injured kidney <Ref. *Nat. Rev. Matter.* 2018, 3, 358–374>. Moreover, TEM image of the urine collected from CNPs-injected AKI mice verifies that CNPs can be excreted through the urine (Supplementary Fig. 17). To further investigate how CNPs enter kidney cells to exert protective effects, we quantified the cellular uptake of RITC-labeled CNPs after adding methyl- β -cyclodextrin (M β CD, an inhibitor of caveolin-mediated endocytosis), chlorpromazine (an inhibitor of clathrin-mediated endocytosis) or amiloride (an inhibitor of macropinocytosis) <Ref. *Colloid. Surfaces B* 2018, 161, 10–17; *Nat.*

Commun. 2016, 7, 11284>. In consistent with previously reported PEGylated nanoparticles <Ref. *Biomaterials* 2011, 32, 3435–3446; *Colloid. Surfaces B* 2018, 161, 10–17>, the cellular uptake of CNPs was reduced by ~43%, ~16%, ~12%, respectively, in the presence of M β CD, chlorpromazine or amiloride, indicating that CNPs can be internalized into HK-2 cells by multiple pathways, among which the caveolin-mediated endocytosis played a major role in the cellular uptake of CNPs (Supplementary Fig. 18).

Our modification to the manuscript: The results were added as Supplementary Figs. 1, 13, 15, 16, 17 and 18 in the revised supporting information. In addition, the following sentences and methods were added on page 4, 6, 7, 14, 15, and 18-20 in the revised manuscript.

- Supplementary Fig. 1

Supplementary Figure 1. a,b, TGA curves of hydrophobic CNPs and CNPs (a) and Fourier transform infrared (FT-IR) spectra (b) of hydrophobic CNPs, physical mixture of DSPE and PEG-COOH (DSPE/PEG), DSPE-PEG and CNPs. Source data are provided as a Source Data file.

- Supplementary Fig. 13

Supplementary Figure 13. *In vivo* blood pharmacokinetic curves of CNPs. Source data are provided as a Source Data file.

- Supplementary Fig. 15

Supplementary Figure 15. Concentration of cerium ions in renal cortex of AKI mice at different time points (1, 2, 8, 24, 48, 72 h) after intravenous administration of CNPs (1.5 mg/kg, n = 4/group). Source data are provided as a Source Data file.

- Supplementary Fig. 16

Supplementary Figure 16. Biodistribution of CNPs in the renal cortex of AKI mice at different time points (10 min, 1 h, 72 h) after intravenous administration of CNPs (1.5 mg/kg). Yellow arrows indicate the presence of CNPs.

- Supplementary Fig. 17

Supplementary Figure 17. TEM image of the urine collected from AKI mice at 2 h after intravenous administration of CNPs (1.5 mg/kg). Scale bar: 50 nm.

- Supplementary Fig. 18

Supplementary Figure 18. a, Confocal images of HK-2 cells pre-treated with serum-free medium (vehicle), chlorpromazine (an inhibitor of clathrin-mediated endocytosis), amiloride (an inhibitor of macropinocytosis) or methyl-b-cyclodextrin (MβCD, an inhibitor of caveolin-mediated endocytosis) followed by co-incubation with 50 μM CNPs. Scale bar: 20 μm. **b,** Quantitative analysis of fluorescence intensity of RITC-labeled CNPs in HK-2 cells after different pretreatments. Source data are provided as a Source Data file.

- Page 4

“.....1,2-distearoyl-sn-glycero-3-phosphoethanolamine-N-[methoxy (polyethylene glycol)-2,000] (DSPE-PEG_{2K}) was successfully surface modified (Supplementary Fig. 1), imparting the colloidal stability of the CNPs in aqueous solution (with a hydrodynamic diameter of ~ 9.7 nm and a surface zeta potential of ~ -20.4 mV) for long blood circulation²⁴⁻²⁶ (Fig. 1c and Supplementary Fig. 2).”

- Page 6-7

“We further evaluated the pharmacokinetic and biodistribution of CNPs in AKI mice. The result of histological examination indicated that CNPs have no evident toxicity to major organs (Supplementary Fig. 12). As shown in Fig. 3a, ICR mice (8 male mice and 8 female mice) were intraperitoneally injected (*i.p.*) with a single dose of DPP (15 mg/kg) to induce AKI, accompanied by intravenous injection (*i.v.*) of saline, CNPs (0.5 or 1.5 mg/kg) or iCNPs (0.5 or 1.5 mg/kg), respectively. The blood circulation profiles of CNPs fitted well with the classic two-compartment pharmacokinetic model, in which the terminal elimination half-lives of the central component and peripheral component

were 0.47 h and 72.2 h, respectively (Supplementary Fig. 13). The enhanced accumulation of CNPs in the kidneys of AKI mice was observed as compared to the normal mice (Supplementary Fig. 14), and CNPs exhibited long retention times in renal cortex (Supplementary Fig. 15). Furthermore, CNPs were found in the renal tubules epithelial cell cilia, glomerular basement membrane (GBM) and renal tubules, as demonstrated by TEM images of renal cortex (Supplementary Fig. 16). The renal accumulation and retention of CNPs are likely due to the small core size (~ 3 nm), deformable PEG surface-modification and negative surface charge of CNPs, as well as enhanced GBM permeability of injured kidney^{40,41}. CNPs were also found in the urine of treated AKI mice, indicating that CNPs can filter through GBM to the renal tubules and to be excreted through urine (Supplementary Fig. 17). Moreover, we studied the mechanism underlying the cellular endocytosis of CNPs in kidney cells. In consistent with previously reported PEGylated nanoparticles⁴², the cellular uptake of CNPs is reduced in the presence of M β CD, chlorpromazine and amiloride, respectively, indicating that CNPs can be internalized into HK-2 cells via multiple pathways, among which the caveolin-mediated endocytosis plays a major role in the cellular uptake of CNPs⁴² (Supplementary Fig. 18).”

References:

40. Du, B., Yu, M. & Zheng, J. Transport and interactions of nanoparticles in the kidneys. *Nat. Rev. Mater.* **3**, 358-374 (2018).
41. Liang, X., et al. Short- and long-term tracking of anionic ultrasmall nanoparticles in kidney. *ACS Nano* **10**, 387-395 (2016).
42. Lee, Y. K., et al. Suppression of human arthritis synovial fibroblasts inflammation using dexamethasone-carbon nanotubes via increasing caveolin-dependent endocytosis and recovering mitochondrial membrane potential. *Int. J. Nanomed.* **12**, 5761-5779 (2017).

- Page 7

“Then, the *in vivo* renal protective effect of CNPs was tested.”

- Page 14

“Cerium(III) acetate (99.99% metals basis) was purchased from Aladdin. Oleylamine (technical grade, 70%), xylene (98.5%) and N-acetyl cysteine (NAC) were purchased from Sigma-Aldrich. Rhodamine B Isothiocyanate (RITC), chlorpromazine, amiloride and methyl- β -cyclodextrin (M β CD) were purchased from Aladdin. Cisplatin injection (5 mg ml⁻¹) and cisplatin powder were purchased from Jiangsu Haosen Pharmaceutical Co., Ltd. 1,2-distearoyl-sn-glycero-3-phosphoethanolamine-N-[methoxy (polyethylene glycol)-2,000] (DSPE-PEG_{2K}) and DSPE-PEG_{2K}-NH₂ were purchased from Shanghai AVT Pharmaceutical Technology Co., Ltd. Hydrogen peroxide (H₂O₂) was purchased from Sinopharm Chemical Reagent Co., Ltd. 1640 medium, McCoy’s 5A medium and fetal bovine serum (FBS) were provide by Gibco BRL (Invitrogen).”

- Page 15

“The same procedure was applied to fabricate RITC-labelled CNPs using DSPE-PEG-RITC.”

“Transmission electron microscopy (TEM) images of CNPs and the collected urine samples were taken using HITACHI HT7700 (Tokyo, Japan) at a voltage of 120 kV. High resolution transmission electron microscopy (HRTEM) image was taken with a FEI Tecnai F20 (FEI, USA) at a voltage of 200 kV. TEM images of kidney tissues were taken using HITACHI H-7650 (Tokyo, Japan) at a voltage of 80 kV. The thermogravimetric analysis (TGA) of DSPE-PEG_{2K} modified or unmodified CNPs was analyzed by a METTLER TOLEDO TGA/DSC1 thermogravimetric analyzer system. The Fourier transform infrared (FT-IR) spectra of unmodified CNPs, physical mixture of DSPE and PEG-COOH, DSPE-PEG and CNPs were analyzed by NicoLET iS50FT-IR.”

- Page 18-19

Endocytic pathway of CNPs.

“To study the endocytic pathway involved in CNPs internalization, three specific

endocytic inhibitors were used: (1) chlorpromazine, an inhibitor of clathrin-mediated endocytosis; (2) amiloride, an inhibitor of micropinocytosis; (3) methyl- β -cyclodextrin (M β CD), an inhibitor of caveolin-mediated endocytosis. HK-2 cells were preincubated in serum-free RPMI 1640 medium with chlorpromazine (30 μ M, 30 min), amiloride (100 μ M, 30 min) or M β CD (5 mM, 30 min), respectively. The medium was then changed to fresh serum-free medium containing the inhibitors plus CNPs (50 μ M) and further incubated for 7 h at 37 °C. The cells were then washed with 1 \times PBS and imaged by using confocal laser scanning microscope (Leica TCS SP8, Germany). The Image J was used for quantitative analysis.”

- Page 20

Pharmacokinetics and biodistribution of CNPs.

“To evaluate the blood circulation half-life of CNPs, ICR mice (Male, n = 3) with established cisplatin-induced AKI model were intravenously injected with CNPs (1.5 mg/kg). At different time points post injection (1 min, 10 min, 1, 4, 8 and 24 h), 15 μ L of plasma samples were collected from the mouse tail vein. The concentrations of CNPs in collected plasma samples were quantified by using ICP-MS. A two-compartment pharmacokinetic model was utilized to calculate the pharmacokinetics parameters of CNPs. Furthermore, the urine of mice were collected at 2 h after intravenous injection of CNPs for TEM analysis.

To evaluate the biodistribution of CNPs in the renal cortex, AKI mice (n = 3) were intravenously injected with CNPs (1.5 mg/kg). The mice were sacrificed to harvest kidneys at 1 h, 2 h, 4 h, 24 h, 48 h and 72 h after injection, respectively. The renal cortex cut from the kidney were weighted and dissolved in nitric acid for ICP-MS measurement.

Three mice were sacrificed at different time points after CNPs injection (10 min, 1 h, 72 h), and a small piece of renal cortex was collected and fixed with the 2.5% glutaraldehyde for 24 h, then post-fixed with 1% osmium tetroxide and prepared for TEM analysis.”

Reviewers' Comments:

Reviewer #5:

Remarks to the Author:

The queries were suitably replied. No further comments and the paper may be accepted.

REVIEWER COMMENTS

Reviewer #5 (Remarks to the Author):

The queries were suitably replied. No further comments and the paper may be accepted.

Response: *We thank the reviewer for supporting the publication of our manuscript.*